# Navigating the landscape of multiplayer games

Shayegan Omidshafiei [1,4 ✉], Karl Tuyls[1,4], Wojciech M. Czarnecki[2], Francisco C. Santos[3], Mark Rowland[2], Jerome Connor[2], Daniel Hennes [1], Paul Muller[1], Julien Pérolat[1], Bart De Vylder[1], Audrunas Gruslys[2] & Rémi Munos[1]

Multiplayer games have long been used as testbeds in artificial intelligence research, aptly referred to as the Drosophila of artificial intelligence. Traditionally, researchers have focused on using well-known games to build strong agents. This progress, however, can be better informed by characterizing games and their topological landscape. Tackling this latter question can facilitate understanding of agents and help determine what game an agent should target next as part of its training. Here, we show how network measures applied to response graphs of large-scale games enable the creation of a landscape of games, quantifying relationships between games of varying sizes and characteristics. We illustrate our findings in domains ranging from canonical games to complex empirical games capturing the performance of trained agents pitted against one another. Our results culminate in a demonstration leveraging this information to generate new and interesting games, including mixtures of empirical games synthesized from real world games.

[1] DeepMind, Paris, France. [2] DeepMind, London, UK. [3] INESC-ID and Instituto Superior Técnico, Universidade de Lisboa, Lisboa, Portugal. [4] These authors contributed equally: Shayegan Omidshafiei, Karl Tuyls. ✉email: somidshafiei@google.com

Games have played a prominent role as platforms for the development of learning algorithms and the measurement of progress in artificial intelligence (AI)[1–4]. Multiplayer games, in particular, have played a pivotal role in AI research and have been extensively investigated in machine learning, ranging from abstract benchmarks in game theory over popular board games such as Chess[5,6] and Go[7] (referred to as the Drosophila of AI research[8]), to realtime strategy games such as StarCraft II[9] and Dota 2[10]. Overall, AI research has primarily placed emphasis on training of strong agents; we refer to this as the Policy Problem, which entails the search for super human-level AI performance. Despite this progress, the need for a task theory, a framework for taxonomizing, characterizing, and decomposing AI tasks has become increasingly important in recent years[11,12]. Naturally, techniques for understanding the space of games are likely beneficial for the algorithmic development of future AI entities[12,13]. Understanding and decomposing the characterizing features of games can be leveraged for downstream training of agents via curriculum learning[14], which seeks to enable agents to learn increasingly-complex tasks.

A core challenge associated with designing such a task theory has been recently coined the Problem Problem, defined as "the engineering problem of generating large numbers of interesting adaptive environments to support research"[15]. Research associated with the Problem Problem has a rich history spanning over 30 years, including the aforementioned work on task theory[11,12,16], procedurally-generated videogame features[17–19], generation of games and rule-sets for General Game Playing[20–26], and procedural content generation techniques[27–36]; we refer readers to Supplementary Note 1 for detailed discussion of these and related works. An important question that underlies several of these interlinked fields is: what makes a game interesting enough for an AI agent to learn to play? Resolving this requires techniques that can characterize the topological landscape of games, which is the topic of interest in this paper. We focus, in particular, on the characterization of multiplayer games (i.e., those involving interactions of multiple agents), and henceforth use the shorthand of games to refer to this class.

The objective of this paper is to establish tools that enable discovery of a topology over games, regardless of whether they are interesting or not; we do not seek to answer the interestingness question here, although such a toolkit can be useful for subsequently considering it. Naturally, many notions of what makes a game interesting exist, from the perspectives of human-centric game design, developmental learning, curriculum learning, AI training, and so on. Our later experiments link to the recent work of Czarnecki et al.[37], which investigated properties that make a game interesting specifically from an AI training perspective, as also considered here. We follow the interestingness characterization of Czarnecki et al.[37], which defines so-called Games of Skill that are engaging for agents due to: (i) a notion of progress; (ii) availability of diverse play styles that perform similarly well. We later show how clusters of games discovered by our approach align with this notion of interestingness. An important benefit of our approach is that it applies to adversarial and cooperative games alike. Moreover, while the procedural game structure generation results we later present target zero-sum games due to the payoff parameterization chosen in those particular experiments, they readily extend to general-sum games.

How does one topologically analyze games? One can consider characterizations of a game as quantified by measures such as the number of strategies available, players involved, whether the game is symmetric, and so on. One could also order the payouts to players to taxonomize games, as done in prior works exploring $2 \times 2$ games[38–40]. For more complex games, however, such measures are crude, failing to disambiguate differences in similar

games. One may also seek to classify games from the standpoint of computational complexity. However, a game that is computationally challenging to solve may not necessarily be interesting to play. Overall, designation of a single measure characterizing games is a non-trivial task.

It seems useful, instead, to consider measures that characterize the possible strategic interactions in the game. A number of recent works have considered problems involving such interactions[41–51]. Many of these works analyze agent populations, relying on game-theoretic models capturing pairwise agent relations. Related models have considered transitivity (or lack thereof) to study games from a dynamical systems perspective[52,53]; here, a transitive game is one where strategies can be ordered in terms of strength, whereas an intransitive game may involve cyclical relationships between strategies (e.g., Rock–Paper–Scissors). Fundamentally, the topology exposed via pairwise agent interactions seems a key enabler of the powerful techniques introduced in the above works. In related literature, graph theory is well-established as a framework for topological analysis of large systems involving interacting entities[54–56]. Complexity analysis via graph-theoretic techniques has been applied to social networks[57,58], the web-graph[59,60], biological systems[61–63], econometrics[64,65], and linguistics[66]. Here, we demonstrate that the combination of graph and game theory provides useful tools for analyzing the structure of general-sum, many-player games.

The primary contribution of this work is a graph-based toolkit for analysis and comparison of games. As detailed below, the nodes in our graphs are either strategies (in abstract games) or AI agents (in empirical games, where strategies correspond to learned or appropriately-sampled player policies). The interactions between these agents, as quantified by the game's payoffs, constitute the structure of the graph under analysis. We show that this set of nodes and edges, also known as the $\alpha$-Rank response graph[49–51], yields useful insights into the structure of individual games and can be used to generate a landscape over collections of games (as in Fig. 1). We subsequently use the toolkit to analyze various games that are both played by humans or wherein AI agents have reached human-level performance, including Go, MuJoCo Soccer, and StarCraft II. Our overall analysis culminates in a demonstration of how the topological structure over games can be used to tackle the interestingness question of the Problem Problem, which seeks to automatically generate games with interesting characteristics for learning agents[15].

## Results

**Overview.** We develop a foundational graph-theoretic toolkit that facilitates analysis of canonical and large-scale games, providing insights into their related topological structure in terms of their high-level strategic interactions. The prerequisite game theory background and technical details are provided in the "Methods" section, with full discussion of related works and additional details in Supplementary Note 1.

Our results are summarized as follows. We use our toolkit to characterize a number of games, first analyzing motivating examples and canonical games with well-defined structures, then extending to larger-scale empirical games datasets. For these larger games, we rely on empirical game-theoretic analysis[67,68], where we characterize an underlying game using a sample set of policies. While the empirical game-theoretic results are subject to the policies used to generate them, we rely on a sampling scheme designed to capture a diverse variety of interactions within each game, and subsequently conduct sensitivity analysis to validate the robustness of the results. We demonstrate correlation between the complexity of the graphs associated with games and the complexity of solving the game itself. In Supplementary Note 2,

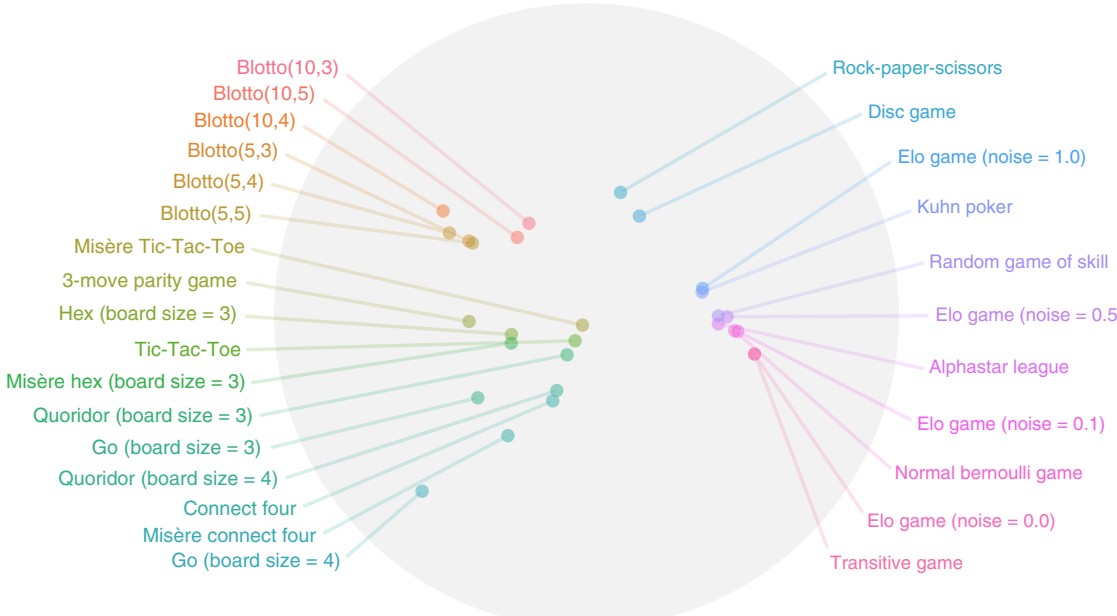

**Fig. 1 A landscape of games revealed by the proposed response graph-based workflow.** This landscape is generated by collecting features associated with the response graph of each game, and plotting the top two principal components. At a high level, games whose response graphs are characteristically similar are situated close to one another in this landscape. Notably, variations of games with related rules are well-clustered together, indicating strong similarity despite their widely-varying raw sizes. Instances of Blotto cluster together, despite their payoff table sizes ranging from 20 × 20 for Blotto(5,3) to 1000 × 1000 for Blotto(10,5). Games with strong transitive components (e.g., variations of Elo games, AlphaStar League, Random Game of Skill, and Normal Bernoulli Game) can be observed to be well separated from strongly cyclical games (Rock–Paper–Scissors and the Disc game). Closely-related real-world games (i.e., games often played by humans in the real world, such as Hex, Tic-Tac-Toe, Connect Four and each of their respective Misere counterparts) are also clustered together.

we evaluate our proposed method against baseline approaches for taxonomization of $2 \times 2$ games[38]. We finally demonstrate how this toolkit can be used to automatically generate interesting game structures that can, for example, subsequently be used to train AI agents.

**Motivating example**. Let us start with a motivating example to solidify intuitions and explain the workflow of our graph-theoretic toolkit, using classes of games with simple parametric structures in the player payoffs. Specifically, consider games of three broad classes (generated as detailed in the Supplementary Methods 1): games in which strategies have a clear transitive ordering (Fig. 2a); games in which strategies have a cyclical structure wherein all but the final strategy are transitive with respect to one another (Fig. 2d); and games with random (or no clear underlying) structure (Fig. 2g). We shall see that the core characteristics of games with shared underlying structure is recovered via the proposed analysis.

Each of these figures visualizes the payoffs corresponding to 4 instances of games of the respective class, with each game involving 10 strategies per player; more concretely, entry $M(s_i, s_j)$ of each matrix visualized in Fig. 2a, d and g quantifies the payoff received by the first player if the players, respectively, use strategies $s_i$ and $s_j$ (corresponding, respectively, to the $i$-th and $j$-th row and column of each payoff table). Despite the variance in payoffs evident in the instances of games exemplified here, each essentially shares the payoff structure exposed by re-ordering their strategies, respectively, in Fig. 2b, e and h. In other words, the visual representation of the payoffs in this latter set of figures succinctly characterizes the backbone of strategic interactions within these classes of games, despite not being immediately apparent in the individual instances visualized.

More importantly, the complexity of learning useful mixed strategies to play in each of these games is closely associated with this structural backbone. To exemplify this, consider the computational complexity of solving each of these games; for brevity, we henceforth refer to solving a game as synonymous with finding a Nash equilibrium (similar to prior works[69–72], wherein the Nash equilibrium is the solution concept of interest). Specifically, we visualize this computational complexity by using the Double Oracle algorithm[73], which has been well-established as a Nash solver in multiagent and game theory literature[47,74–76]. At a high level, Double Oracle starts from a sub-game consisting of a single randomly-selected strategy, iteratively expands the strategy space via best responses (computed by an oracle), until discovery of the Nash equilibrium of the full underlying game.

Figure 2c, f and i visualize the distribution of Double Oracle iterations needed to solve the corresponding games, under random initializations. Note, in particular, that although the underlying payoff structure of the transitive and cyclical games, respectively, visualized in Fig. 2a, d is similar, the introduction of a cycle in the latter class of games has a substantial impact on the complexity of solving them (as evident in Fig. 2f). In particular, whereas the former class of games are solved using a low (and deterministic) number of iterations, the latter class requires additional iterations due to the presence of cycles increasing the number of strategies in the support of the Nash equilibrium.

**Workflow**. Overall, the characterization of the topological structure of games is an important and nuanced problem. To address this problem, we use graph theory to build an analytical toolkit automatically summarizing the high-level strategic interactions within a game, and providing useful complexity measures

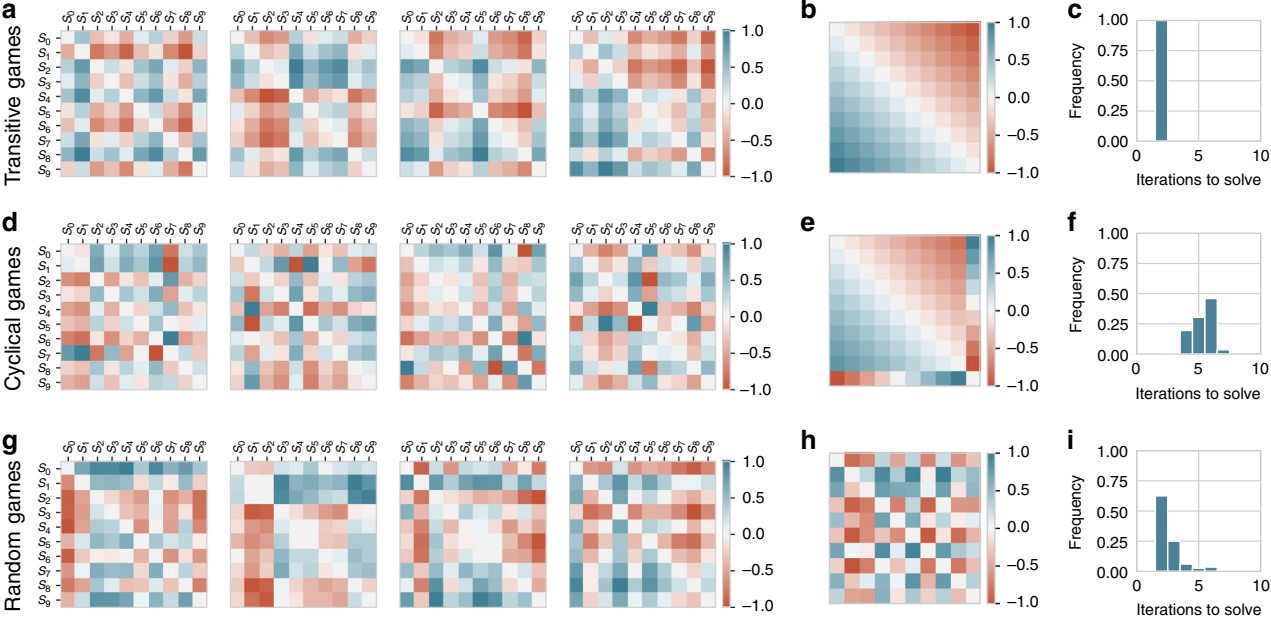

**Fig. 2 Motivating example of three classes of two-player, symmetric zero-sum games. a**, **d**, and **g**, respectively, visualize payoffs for instances of games with transitive, cyclical, and random structure. Each exemplified game consists of two players with 10 strategies each (with payoff row and column labels, $\{s_0, ..., s_9\}$, indicating the strategies). Despite the numerous payoff variations possible in each class of games illustrated, each shares the underlying payoff structure shown, respectively, in **b**, **e**, and **h**. Moreover, variations in payoffs can notably impact the difficulty of solving (i.e., finding the Nash equilibrium) of these games, as visualized in **c**, **f**, **i**.

thereof. Specifically, consider again our motivating transitive game, re-visualized using a collection of graph-based measures in Fig. 3. Each of these measures provides a different viewpoint on the underlying game, collectively characterizing it. Specifically, given the game payoffs, Fig. 3b visualizes the so-called $\alpha$-Rank response graph of the game; here, each node corresponds to a strategy (for either player, as this particular game's payoffs are symmetric). Transition probabilities between nodes are informed by a precise evolutionary model (detailed in "Methods" section and Omidshafiei et al.[50]); roughly speaking, a directed edge from one strategy to another indicates the players having a higher preference for the latter strategy, in comparison to the former. The response graph, thus, visualizes all preferential interactions between strategies in the game. Moreover, the color intensity of each node indicates its so-called $\alpha$-Rank, which measures the long-term preference of the players for that particular strategy, as dictated by the transition model mentioned above; specifically, darker colors here indicate more preferable strategies.

This representation of a game as a graph enables a variety of useful insights into its underlying structure and complexity. For instance, consider the distribution of cycles in the graph, which play an important role in multiagent evaluation and training schemes[41,50,53,77] and, as later shown, are correlated to the computational complexity of solving two-player zero-sum games (e.g., via Double Oracle). Figure 3f makes evident the lack of cycles in the particular class of transitive games; while this is clearly apparent in the underlying (fully ordered) payoff visualization of Fig. 3a, it is less so in the unordered variants visualized in Fig. 2a. Even so, the high-level relational structure between the strategies becomes more evident by conducting a spectral analysis of the underlying game response graph. Full technical details of this procedure are provided in the "Methods" section. At a high level, the so-called Laplacian spectrum (i.e., eigenvalues) of a graph, along with associated eigenvectors, captures important information regarding it (e.g., number of spanning trees, algebraic connectivity, and numerous related

properties[78]). Reprojecting the response graph by using the top eigenvectors yields the spectral response graph visualized in Fig. 3c, wherein similar strategies are placed close to one another. Moreover, one can cluster the spectral response graph, yielding the clustered response graph, which exposes three classes of strategies in Fig. 3d: a fully dominated strategy with only outgoing edges (a singleton cluster, on the bottom left of the graph), a transient cluster of strategies with both incoming and outgoing edges (top cluster), and a dominant strategy with all incoming edges (bottom right cluster). Finally, contracting the clustered graph by fusing nodes within each cluster yields the high-level characterization of transitive games shown in Fig. 3e.

We can also conduct this analysis for instances of our other motivating games, such as the cyclical game visualized in Fig. 4a. Note here the distinct differences with the earlier transitive game example; in the cyclical game, the $\alpha$-Rank distribution in the response graph (Fig. 4b) has higher entropy (indicating preference for many strategies, rather than one, due to the presence of cycles). Moreover, the spectral reprojection in Fig. 4d reveals a clear set of transitive nodes (left side of visualization) and a singleton cluster of a cycle-inducing node (right side). Contracting this response graph reveals the fundamentally cyclical nature of this game (Fig. 4f). Finally, we label each strategy (i.e., each row and column) of the original payoff table Fig. 4a based on this clustering analysis. Specifically, the color-coded labels on the far left (respectively, top) of each row (respectively, column) in Fig. 4a correspond to the clustered strategy colors in Fig. 4d. This color-coding helps clearly identify the final strategy (i.e., bottom row of the payoff table) as the outlier enforcing the cyclical relationships in the game. Note that while there is no single graphical structure that summarizes the particular class of random games visualized earlier in Fig. 2h, we include this analysis for several instances of such games in Supplementary Note 2.

Crucially, a key benefit of this analysis is that the game structure exposed is identical for all instances of the transitive and cyclical games visualized earlier in Fig. 2a, d, making it significantly easier

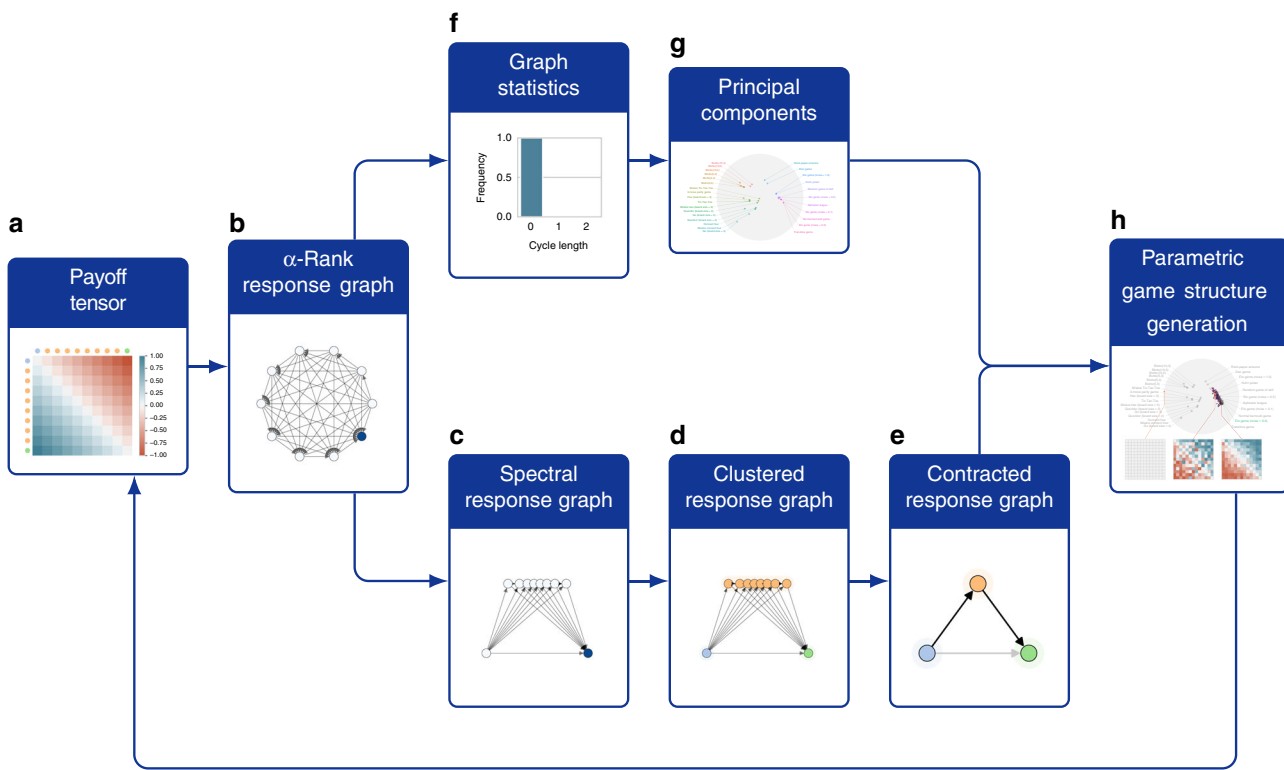

**Fig. 3 Method workflow, with accompanying transitive game results.** Given the game payoffs **a**, the so-called $\alpha$-Rank response graph of the game is visualized in **b**. In **c**, reprojecting the response graph by using the top eigenvectors of the graph Laplacian yields the spectral response graph, wherein similar strategies are placed close to one another. In **d**, taking this one step further, one can cluster the spectral response graph, yielding the clustered response graph, which exposes three classes of strategies in this particular example. In **e**, contracting the clustered graph by fusing nodes within each cluster yields the high-level characterization of transitive games. In **f**, the lack of cycles in the particular class of transitive games becomes evident. Finally, in **g** and **h**, one can extract the principal components of various response graph statistics and establish a feedback loop to a procedural game structure generation scheme to yield new games.

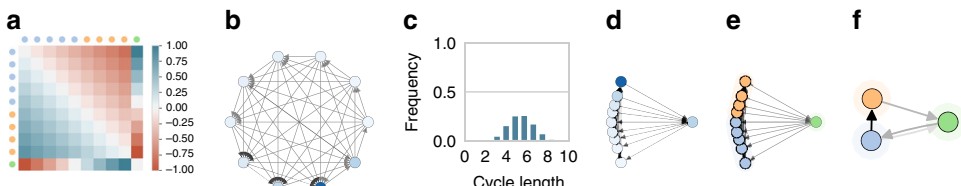

**Fig. 4 Cyclical game results. a** Game payoffs, **b** response graph, **c** cycles histogram, **d** spectral response graph, **e** clustered response graph, **f** contracted response graph.

to characterize games with related structure, in contrast to analysis of raw payoffs. Our later case studies further exemplify this, exposing related underlying structures for several classes of more complex games.

**Analysis of canonical and real-world games.** The insights afforded by our graph-theoretic approach apply to both small canonical games and larger empirical games (where strategies are synonymous with trained AI agents).

Consider the canonical Rock–Paper–Scissors game, involving a cycle among the three strategies (wherein Rock loses to Paper, which loses to Scissors, which loses to Rock). Figure 5a visualizes a variant of this game involving a copy of the first strategy, Rock, which introduces a redundant cycle and thus affects the distribution of cycles in the game. Despite this, the spectral response graph (Fig. 5d) reveals that the redundant game topologically remains the same as the original Rock–Paper–Scissors game, thus reducing to the original game under spectral clustering.

This graph-based analysis also extends to general-sum games. As an example, consider the slightly more complex game of 11–20, wherein two players each request an integer amount of money between 11 and 20 units (inclusive). Each player receives the amount requested, though a bonus of 20 units is allotted to one player if they request exactly 1 unit less than the other player. The payoffs and response graph of this game are visualized, respectively, in Fig. 5g, h, where strategies, from top-to-bottom and left-to-right in the payoff table, correspond to increasing units of money being requested. This game, first introduced by Arad and Rubinstein[79], is structurally designed to analyze so-called $k$-level reasoning, wherein a level-0 player is naive (i.e., here simply requests 20 units), and any level-$k$ player responds to an assumed level-$(k-1)$ opponent; e.g., here a level-1 player best responds to an assumed level-0 opponent, thus requesting 19 units to ensure receiving the bonus units.

The spectral response graph here (Fig. 5j) indicates a more complex mix of transitive and intransitive relations between

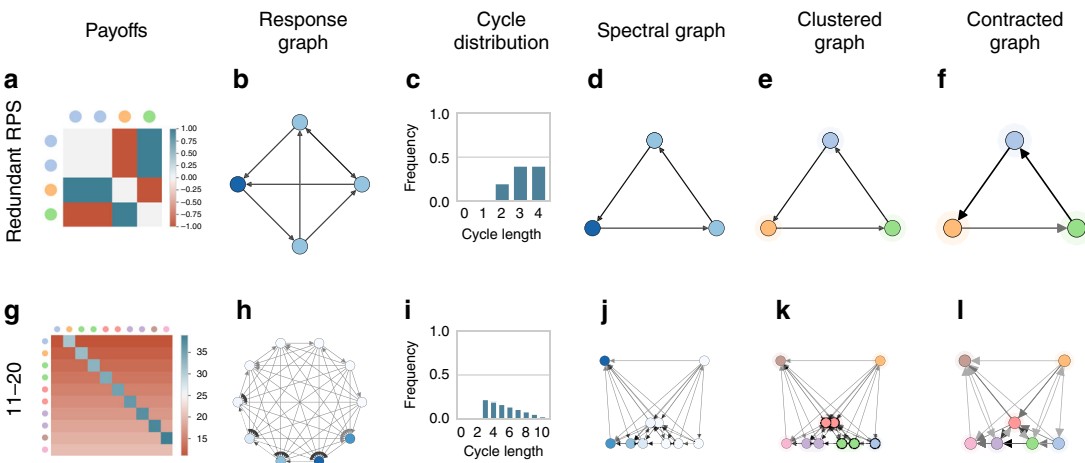

**Fig. 5 Results for redundant Rock–Paper–Scissors (RPS) and 11–20 game.** In Redundant RPS, the redundant copy of the first strategy (Rock) is clustered in the spectral response graph. In 11–20, seven clusters of strategies are revealed, exposing the cyclical nature of this game.

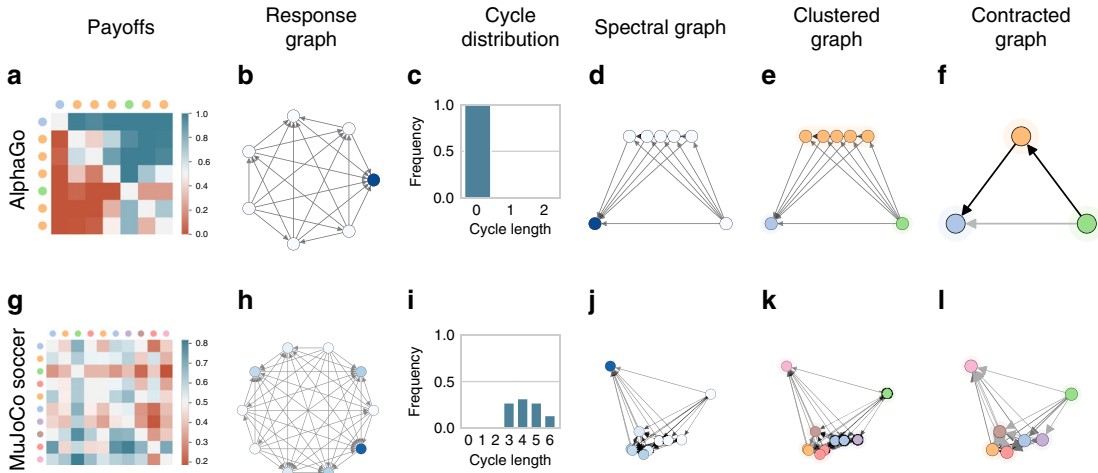

**Fig. 6 Results for empirical games of AlphaGo and MuJoCo soccer dataset.** Note that as these are empirical games, strategies here correspond to trained AI agents. In AlphaGo, the strong transitive relationship between agents is revealed via our analysis. In MuJoCo soccer, more complex relations between similarly-performing agents are revealed in the clusters produced.

strategies. Notably, the contracted response graph (Fig. 5l) reveals 7 clusters of strategies. Referring back to the rows of payoffs in Fig. 5g, relabeled to match cluster colors, demonstrates that our technique effectively pinpoints the sets of strategies that define the rules of the game: weak strategies (11 or 12 units, first two rows of the payoff table, and evident in the far-right of the clustered response graph), followed by a set of intermediate strategies with higher payoffs (clustered pairwise, near the lower-center of the clustered response graph), and finally the two key strategies that establish the cyclical relationship within the game through $k$-level reasoning (19 and 20 units, corresponding to level-0 and level-1 players, in the far-left of the clustered response graph).

This analysis extends to more complex instances of empirical games, which involve trained AI agents, as next exemplified. Consider first the game of Go, as played by 7 AlphaGo variants: $AG(r)$, $AG(p)$, $AG(v)$, $AG(rv)$, $AG(rp)$, $AG(vp)$, and $AG(rvp)$, where each variant uses the specified combination of rollouts $r$, value networks $v$, and/or policy networks $p$. We analyze the empirical game where each strategy corresponds to one of these agents, and payoffs (Fig. 6a) correspond to the win rates of these agents when paired against each other (as detailed by Silver et al.[7]

Table 9). The $\alpha$-Rank distribution indicated by the node (i.e., strategy) color intensities in Fig. 6b reveals $AG(rvp)$ as a dominant strategy, and the cycle distribution graph Fig. 6c reveals a lack of cycles here. The spectral response graph, however, goes further, revealing a fully transitive structure (Fig. 6d, e), as in the motivating transitive games discussed earlier. The spectral analysis on this particular empirical game, therefore, reveals its simple underlying transitive structure (Fig. 6f).

Consider a more interesting empirical game, wherein agents are trained to play soccer in the continuous control domain of MuJoCo, exemplified in Fig. 6 (second row). Each agent in this empirical game is generated using a distinct set of training parameters (e.g., feedforward vs. recurrent policies, reward shaping enabled and disabled, etc.), with full agent specifications and payoffs detailed by Liu et al.[80] The spectral response graph (Fig. 6k) reveals two outlier agents: a strictly dominated agent (node in the top-right), and a strong (yet not strictly dominant) agent (node in the top-left). Several agents here are clustered pairwise, revealing their closely-related interactions with respect to the other agents; such information could, for example, be used to discard or fuse such redundant agents during training to save computational costs.

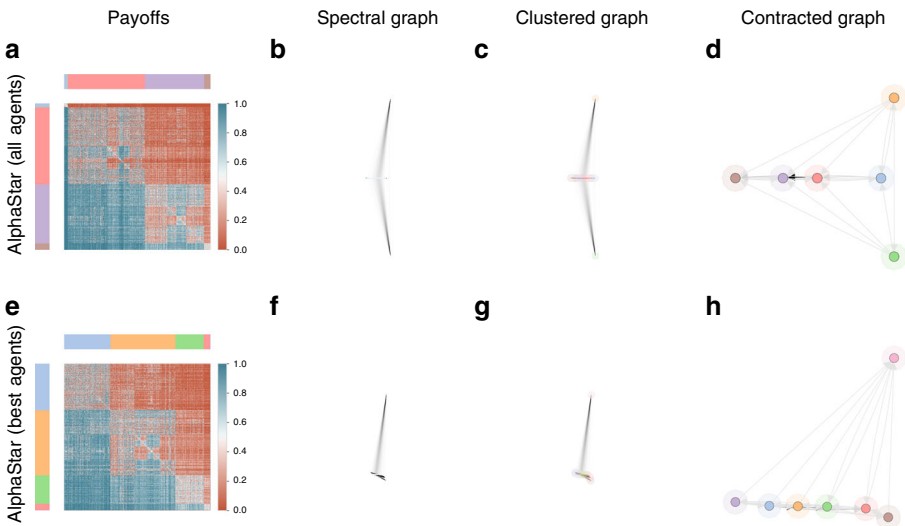

**Fig. 7 AlphaStar results, with both the full league and the league with best agents visualized.** Spectral analysis of the empirical AlphaStar League game reveals that several key subsets of closely-performing agents, illustrated in **d**. Closer inspection of the agents used to construct this empirical payoff table reveals the following insights, with agent types corresponding to those detailed in Vinyals et al.[9]: (i) the blue, orange, and green clusters are composed of agents in the initial phases of training, which are generally weakest (as observed in **d**, and also visible as the narrow band of low payoffs in the top region of the payoff table **a**); (ii) the red cluster consists primarily of various, specialized exploiter agents; (iii) the purple and brown clusters are primarily composed of the league exploiters and main agents, with the latter being generally higher strength than the former.

Consider next a significantly larger-scale empirical game, consisting of 888 StarCraft II agents from the AlphaStar Final league of Vinyals et al.[9]. StarCraft II is a notable example, involving a choice of 3 races per player and realtime gameplay, making a wide array of behaviors possible in the game itself. The empirical game considered is visualized in Fig. 7a, and is representative of a large number of agents with varying skill levels. Despite its size, spectral analysis of this empirical game reveals that several key subsets of closely-performing agents exist here, illustrated in Fig. 7d. Closer inspection of the agents used to construct this empirical payoff table reveals the following insights, with agent types corresponding to those detailed in Vinyals et al.[9]: (i) the blue, orange, and green clusters are composed of agents in the initial phases of training, which are generally weakest (as observed in Fig. 7d, and also visible as the narrow band of low payoffs in the top of Fig. 7a); (ii) the red cluster consists primarily of various, specialized exploiter agents; iii) the purple and brown clusters are primarily composed of the league exploiters and main agents, with the latter being generally higher strength than the former. To further ascertain the relationships between only the strongest agents, we remove the three clusters corresponding to the weakest agents, repeating the analysis in Fig. 7 (bottom row). Here, we observe the presence of a series of progressively stronger agents (bottom nodes in Fig. 7h), as well as a single outlier agent which quite clearly bests several of these clusters (top node of Fig. 7h).

An important caveat, as this stage, is that the agents in AlphaGo, MuJoCo soccer, and AlphaStar above were trained to maximize performance, rather than to explicitly reveal insights into their respective underlying games of Go, soccer, and StarCraft II. Thus, this analysis focused on characterizing relationships between the agents from the Policy Problem perspective, rather than the underlying games themselves, which provide insights into the interestingness of the game (Problem Problem). This latter investigation would require a significantly larger population of agents, which cover the policy space of the underlying game effectively, as exemplified next.

Naturally, characterization of the underlying game can be achieved in games small enough where all possible policies can be

explicitly compared against one another. For instance, consider Blotto($\tau, \rho$), a zero-sum two-player game wherein each player has $\tau$ tokens that they can distribute amongst $\rho$ regions[81]. In each region, each player with the most tokens wins (see Tuyls et al.[53] for additional details). In the variant we analyze here, each player receives a payoff of $+1$, 0, and $-1$ per region, respectively, won, drawn, and lost. The permutations of each player's allocated tokens, in turn, induce strong cyclical relations between the possible policies in the game. While the strategy space for this game is of size $\binom{\tau + \rho - 1}{\rho - 1}$, payoffs matrices can be fully specified for small instances, as shown for Blotto(5,3) and Blotto (10,3) in Fig. 8 (first and second row, respectively). Despite the differences in strategy space sizes in these particular instances of Blotto, the contracted response graphs in Fig. 8d, h capture the cyclical relations underlying both instances, revealing a remarkably similar structure.

For larger games, the cardinality of the pure policy space typically makes it infeasible to fully enumerate policies and construct a complete empirical payoff table, despite the pure policy space being finite in size. For example, even in games such as Tic–Tac–Toe, while the number of unique board configurations is reasonably small ($9! = 362880$), the number of pure, behaviorally unique policies is enormous ($\approx 10^{567}$, see Czarnecki et al.[37] Section J for details). Thus, coming up with a principled definition of a scheme for sampling a relevant set of policies summarizing the strategic interactions possible within large games remains an important open problem. In these instances, we rely on sampling policies in a manner that captures a set of representative policies, i.e., a set of policies of varying skill levels, which approximately capture variations of strategic interactions possible in the underlying game. The policy sampling approach we use is motivated with the above discussions and open question in mind, in that it samples a set of policies with varying skill levels, leading to a diverse set of potential transitive and intransitive interactions between them.

Specifically, we use the policy sampling procedure proposed by Czarnecki et al.[37], which also seeks a set of representative policies for a given game. The details of this procedure are provided in

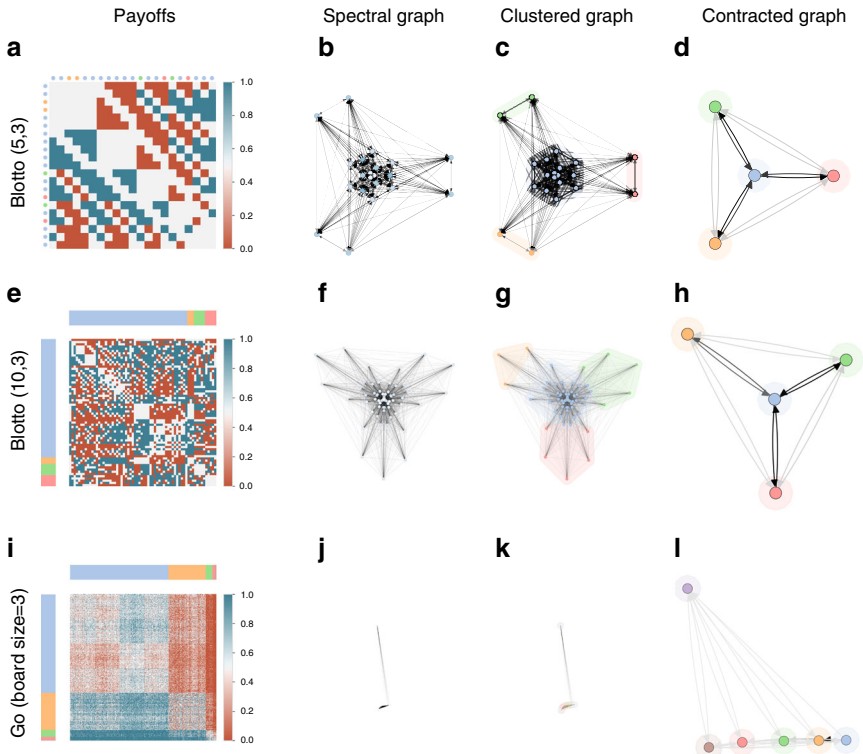

**Fig. 8 Results for Blotto(5,3), Blotto(10,3), and Go (board size=3).** Despite the significant difference in sizes, both instances of Blotto yield a remarkably similar contracted response graph. Moreover, the contracted response graph for Go is notably different from AlphaGo results, due to the latter being an empirical game constructed from trained AI agents rather than a representative set of sampled policies.

Supplementary Methods 1, and at a high level involve three phases: (i) using a combination of tree search algorithms, Alpha–Beta[82] and Monte Carlo Tree Search[83], with varying tree depth limits for the former and varying number of simulations allotted to the latter, thus yielding policies of varying transitive strengths; (ii) using a range of random seeds in each instantiation of the above algorithms, thus producing a range of policies for each level of transitive strength; (iii) repeating the same procedure with negated game payoffs, thus also covering the space of policies that actively seek to lose the original game. While this sampling procedure is a heuristic, it produces a representative set of policies with varying degrees of transitive and intransitive relations, and thus provides an approximation of the underlying game that can be feasibly analyzed.

Let us revisit the example of Go, constructing our empirical game using the above policy sampling scheme, rather than the AlphaGo agents used earlier. We analyze a variant of the game with board size $3 \times 3$, as shown in Fig. 8 (third row). Notably, the contracted response graph (Fig. 8l) reveals the presence of a strongly-cyclical structure in the underlying game, in contrast to the AlphaGo empirical game (Fig. 6). Moreover, the presence of a reasonably strong agent (visible in the top of the contracted response graph) becomes evident here, though this agent also shares cyclical relations with several sets of other agents. Overall, this analysis exemplifies the distinction between analyzing an underlying game (e.g., Go) vs. analyzing the agent training process (e.g., AlphaGo). Investigation of links between these two lines of analysis, we believe, makes for an interesting avenue for future work.

**Linking response graph and computational complexity.** A question that naturally arises is whether certain measures over response graphs are correlated with the computational complexity of solving their associated games. We investigate these potential correlations here, while noting that these results are not intended to propose that a specific definition of computational complexity (e.g., with respect to Nash) is explicitly useful for defining a topology/classification over games. In Fig. 9, we compare several response graph complexity measures against the number of iterations needed to solve a large collection of games using the Double Oracle algorithm[73]. The results here consider specifically the $\alpha$-Rank entropy, number of 3-cycles, and mean in-degree (with details in "Methods" section, and results for additional measures included in Supplementary Note 2). As in earlier experiments, solution of small-scale games is computed using payoffs over full enumeration of pure policies, whereas that of larger games is done using the empirical games over sampled policies. Each graph complexity measure reported is normalized with respect to the maximum measure possible in a graph of the same size, and the number of iterations to solve is normalized with respect to the number of strategies in the respective game. Thus, for each game, the normalized number of iterations to solve provides a measure of its relative computational complexity compared to games with the same strategy space size (for completeness, we include experiments testing the effects of normalization on these results in Supplementary Note 2).

Several trends can be observed in these results. First, the entropy of the $\alpha$-Rank distribution associated with each game correlates well with its computational complexity (see Spearman's correlation coefficient $\rho_s$ in the top-right of Fig. 9a). This matches intuition, as higher entropy $\alpha$-Rank distributions indicate a larger support over the strategy space (i.e., strong strategies, with non-zero $\alpha$-Rank mass), thus requiring additional iterations to solve. Moreover, the number of 3-cycles in the response graph also correlates well with computational complexity, again matching intuition as the intransitivities introduced by cycles typically make it more difficult to traverse the strategy space[42]. Finally, the

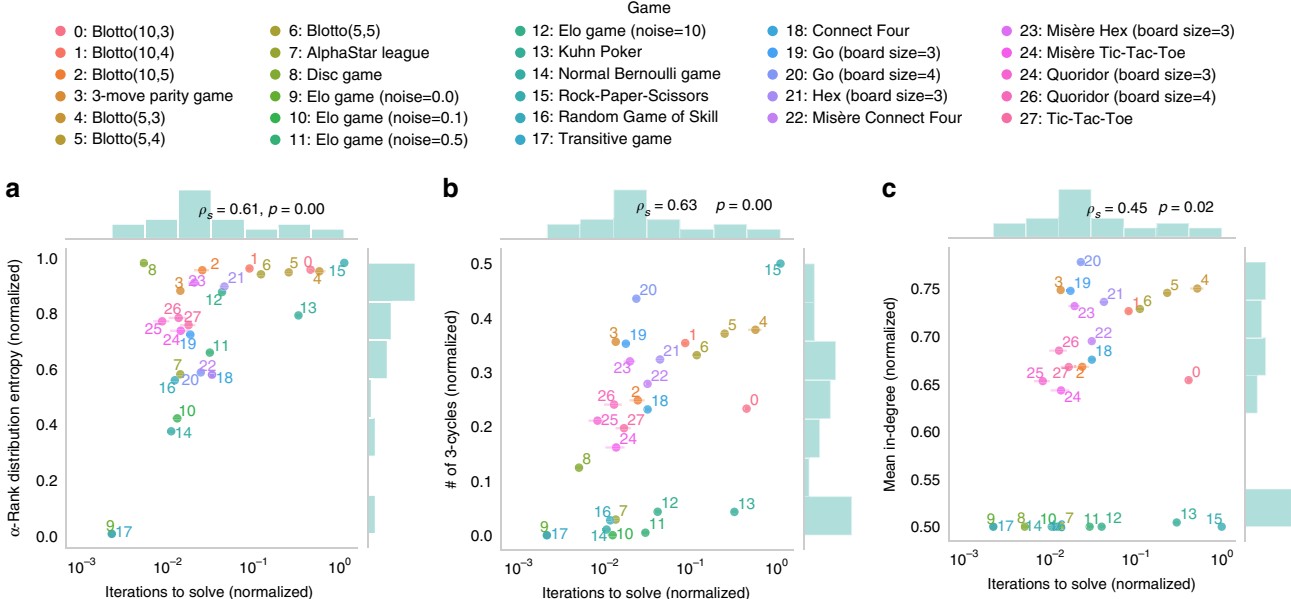

**Fig. 9 Response graph complexity vs. computational complexity of solving associated games.** Each figure plots a respective measure of graph complexity against the normalized number of iterations needed to solve the associated game via the Double Oracle algorithm (with normalization done with respect to the total number of strategies in each underlying game). The Spearman correlation coefficient, $\rho_s$, is shown in each figure (with the reported two-sided $p$-value rounded to two decimals).

mean in-degree over all response graph nodes correlates less so with computational complexity (though degree-based measures still serve a useful role in characterizing and distinguishing graphs of differing sizes[84]). Overall, these results indicate that response graph complexity provides a useful means of quantifying the computational complexity of games.

**The landscape of games.** The results, thus far, have demonstrated that graph-theoretic analysis can simplify games (via spectral clustering), uncover their topological structure (e.g., transitive structure of the AlphaGo empirical game), and yield measures correlated to the computational complexity of solving these games. Overall, it is evident that the perspective offered by graph theory yields a useful characterization of games across multiple fronts. Given this insight, we next consider whether this characterization can be used to compare a widely-diverse set of games.

To achieve this, we construct empirical payoff tables for a suite of games, using the policy sampling scheme described earlier for the larger instances (also see Supplementary Methods 1 for full details, including description of the games considered and analysis of the sensitivity of these results to the choice of empirical policies and mixtures thereof). For each game, we compute the response graphs and several associated local and global complexity measures (e.g., $\alpha$-Rank distribution entropy, number of 3-cycles, node-wise in-degrees and out-degree statistics, and several other measures detailed in the "Methods" section), which constitute a feature vector capturing properties of interest. Finally, a principal component analysis of these features yields the low-dimensional visualization of the landscape of games considered, shown in Fig. 1.

We make several key insights given this empirical landscape of games. Notably, variations of games with related rules are well-clustered together, indicating strong similarity despite the widely-varying sizes of their policy spaces and empirical games used to construct them; specifically, all considered instances of Blotto cluster together, with empirical game sizes ranging from $20 \times 20$ for Blotto(5,3) to $1000 \times 1000$ for Blotto(10,5). Moreover, games

with strong transitive components (e.g., variations of Elo games, AlphaStar League, Random Game of Skill, and Normal Bernoulli Game) are also notably separated from strongly cyclical games (Rock–Paper–Scissors, Disc game, and Blotto variations). Closely-related real-world games (i.e., games often played by humans in the real world, such as Hex, Tic-Tac-Toe, Connect Four, and each of their respective Misère counterparts wherein players seek to lose) are also well-clustered. Crucially, the strong alignment of this analysis with intuitions of the similarity of certain classes of games serves as an important validation of the graph-based analysis technique proposed in this work. In addition, the analysis and corresponding landscape of games make clear that several games of interest for AI seem well-clustered together, which also holds for less interesting games (e.g., Transitive and Elo games).

We note that the overall idea of generating such a landscape over games ties closely with prior works on taxonomization of multiplayer games[38,85]. Moreover, 2D visualization of the expressivity (i.e., style and diversity) and the overall space of of procedurally-generated games features have been also investigated in closely related work[86,87]. A recent line of related inquiry also investigates the automatic identification, and subsequent visualization of core mechanics in single-player games[88]. Overall, we believe this type of investigation can be considered a method to taxonomize which future multiplayer games may be interesting, and which ones less so to train AI agents on.

While the primary focus of this paper is to establish a means of topologically studying games (and their similarities), a natural artifact of such a methodology is that it can enable investigation of interesting and non-interesting classes of games. There exist numerous perspectives on what may make a game interesting, which varies as a function of the field of study or problem being solved. These include (overlapping) paradigms that consider interestingness from: a human-centric perspective (e.g., level of social interactivity, cognitive learning and problem solving, enjoyment, adrenaline, inherent challenge, esthetics, story-telling in the game, etc.)[89–94]; a curriculum learning perspective (e.g., games or tasks that provide enough learning signal to the

human or artificial learner)[95]; a procedural content generation or optimization perspective (in some instances with a focus on General Game Playing)[21] that use a variety of fitness measures to generate new game instances (e.g., either direct measures related to the game structure, or indirect measures such as player win-rates)[20,31,34,35,96]; and game-theoretic, multiplayer, or player-vs.-task notions (e.g., game balance, level of competition, social equality or welfare of the optimal game solution, etc.)[97–99]. Overall, it is complex (and, arguably, not very useful) to introduce a unifying definition of interestingness that covers all of the above perspectives.

Thus, we focus here on a specialized notion of interestingess from the perspective of AI training, linked to the work of Czarnecki et al.[37]. As mentioned earlier, Czarnecki et al.[37] introduce the notion of Games of Skill, which are of interest, in the sense of AI training, due to two axes of interactions between agents: a transitive axis enabling progression in terms of relative strength or skill, and a radial axis representing diverse intransitive/cyclical interactions between strategies of similar strength levels. The overall outlook provided by the above paper is that in these games of interest there exist many average-strength strategies with intransitive relations among them, whereas the level of intransitivity decreases as transitive strength moves towards an extremum (either very low, or very high strength); the topology of strategies in a Game of Skill is, thus, noted to resemble a spinning top.

Through the spinning top paradigm, Czarnecki et al.[37] identifies several real-world games (e.g., Hex, Tic–Tac–Toe, Connect–Four, etc.) as Games of Skill. Notably, the lower left cluster of games in Fig. 1 highlights precisely these games. Interestingly, while the Random Game of Skill, AlphaStar League, and Elo game (noise = 0.5) are also noted as Games of Skill in Czarnecki et al.[37], they are found in a distinct cluster in our landscape. On closer inspection of these payoffs for this trio of games, there exists a strong correlation between the number of intransitive relations and the transitive strength of strategies, in contrast to other Games of Skill such as Hex. Our landscape also seems to highlight non-interesting games. Specifically, variations of Blotto, Rock–Paper–Scissors, and the Disc Game are noted to not be Games of Skill in Czarnecki et al.[37], and are also found to be in distinct clusters in Fig. 1.

Overall, these results highlight how topological analysis of multiplayer games can be used to not only study individual games, but also identify clusters of related (potentially interesting) games. For completeness, we also conduct additional studies in Supplementary Note 2, which compare our taxonomization of 2 × 2 normal-form games (and clustering into potential classes of interest) to that of Bruns[38].

**The problem problem and procedural game structure generation.** Having now established various graph-theoretic tools for characterizing games of interest, we revisit the so-called Problem Problem, which targets automatic generation of interesting environments. Here we focus on the question of how we can leverage the topology discovered by our method to procedurally generate collections of new games (which can be subsequently analyzed, used for training, characterized as interesting or not per previous discussions, and so on).

Full details of our game generation procedure are provided in the "Methods" Section. At a high level, we establish the feedback loop visualized in Fig. 3, enabling automatic generation of games as driven by our graph-based analytical workflow. We generate the payoff structure of a game (i.e., as opposed to the raw underlying game rules, e.g., as done by Browne and Maire[20]). Thus, the generated payoffs can be considered either direct representations of normal form games, or empirical games indirectly representing underlying games with complex rules. At a high level, given a parameterization of a generated game that specifies an associated payoff tensor, we synthesize its response graph and associated measures of interest (as done when generating the earlier landscape of games). We use the multidimensional Elo para-meterization for generating payoffs, due to its inherent ability to specify complex transitive and intransitive games[41]. We then specify an objective function of interest to optimize over these graph-based measures. As only the evaluations of such graph-based measures (rather than their gradients) are typically available, we use a gradient-free approach to iteratively generate games optimizing these measures (CMA-ES[100] is used in our experiments). The overall generation procedure used can be classified as a Search-Based Procedural Content Generation technique (SBPCG)[34]. Specifically, in accordance to the taxonomy defined by Togelius et al.[34], our work uses a direct encoding representation of the game (as the generated payoffs are represented as real-valued vectors of mElo parameters), with a theory-driven direct evaluation used for quantifying the game fitness/quality (as we rely on graph theory to derive the game features of interest, then directly minimize distances of principal components of generated and target games).

Naturally, we can maximize any individual game complexity measures, or a combination thereof, directly (e.g., entropy of the $\alpha$-Rank distribution, number of 3-cycles, etc.). More interestingly, however, we can leverage our low-dimensional landscape of games to directly drive the generation of new games towards existing ones with properties of interest. Consider the instance of game generation shown in Fig. 10a, which shows an overview of the above pipeline generating a 5 × 5 game minimizing Euclidean distance (within the low-dimensional complexity landscape) to the standard 3 × 3 Rock–Paper–Scissors game. Each point on this plot corresponds to a generated game instance. The payoffs visualized, from left to right, respectively correspond to the initial procedural game parameters (which specify a game with constant payoffs), intermediate parameters, and final optimized parameters; projections of the corresponding games within the games landscape are also indicated, with the targeted game of interest (Rock–Paper–Scissors here) highlighted in green. Notably, the final optimized game exactly captures the underlying rules that specify a general-size Rock–Paper–Scissors game, in that each strategy beats as many other strategies as it loses to. In Fig. 10b, we consider a larger 13 × 13 generated game, which seeks to minimize distance to a 1000 × 1000 Elo game (which is transitive in structure, as in our earlier motivating example in Fig. 2a). Once again, the generated game captures the transitive structure associated with Elo games.

Next, we consider generation of games that exhibit properties of mixtures of several target games. For example, consider what happens if 3 × 3 Rock–Paper–Scissors were to be combined with the 1000 × 1000 Elo game above; one might expect a mixture of transitive and cyclical properties in the payoffs, though the means of generating such mixed payoffs directly is not obvious due to the inherent differences in sizes of the targeted games. Using our workflow, which conducts this optimization in the low-dimensional graph-based landscape, we demonstrate a sequence of generated games targeting exactly this mixture in Fig. 10c. Here, the game generation objective is to minimize Euclidean distance to the mixed principal components of the two target games (weighted equally). The payoffs of the final generated game exhibit exactly the properties intuited above, with predominantly positive (blue) upper-triangle of payoff entries establishing a transitive structure, and the more sporadic positive entries in the lower-triangle establishing cycles.

Naturally, this approach opens the door to an important avenue for further investigation, targeting generation of yet more

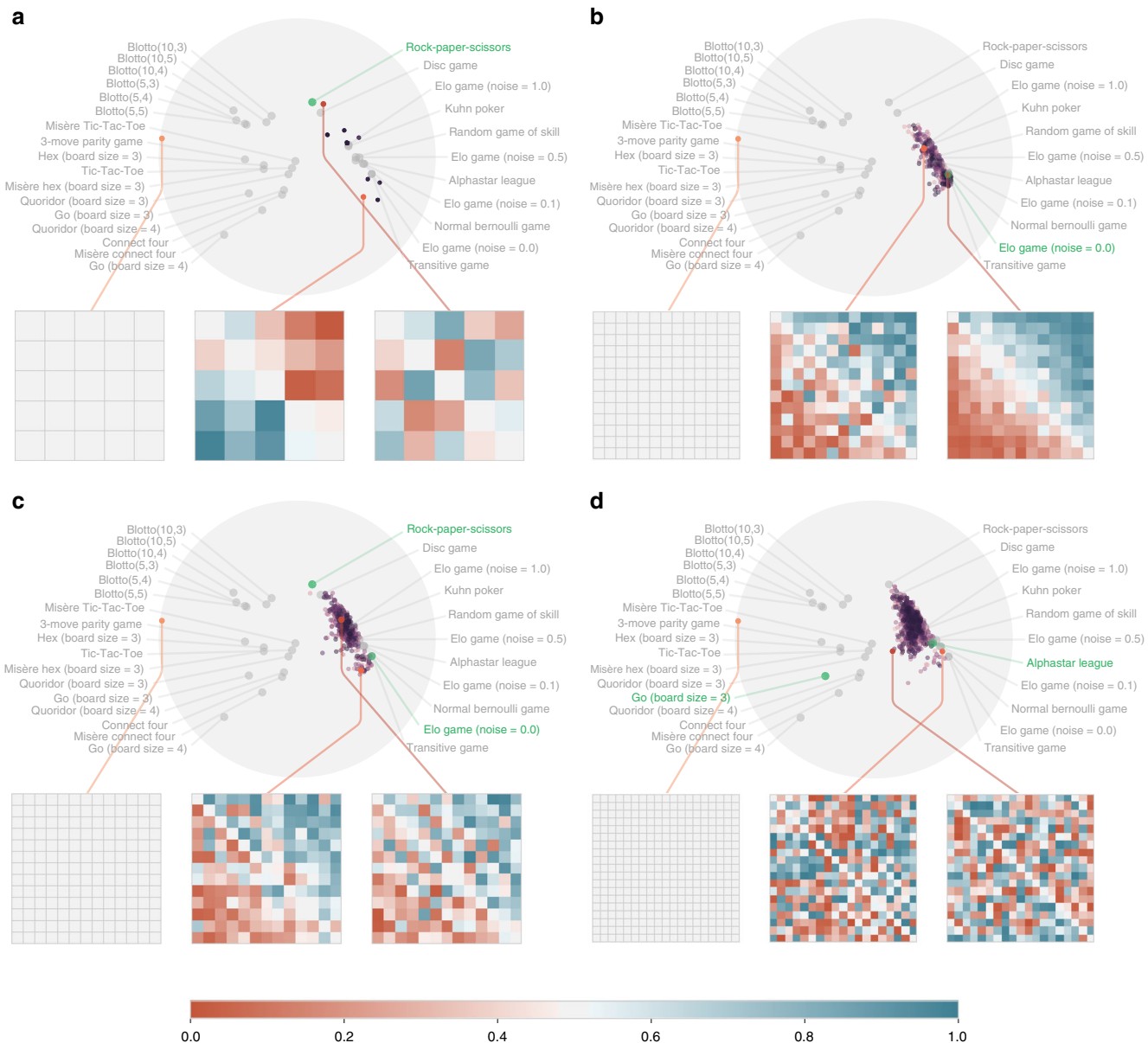

**Fig. 10 Visualization of procedural game structure generation projected in the games landscape.** Each figure visualizes the generation of a game of specified size, which targets a pre-defined game (or mixture of games) of a different size. The three payoffs in each respective figure, from left to right, correspond to the initial procedural game parameters, intermediate parameters, and final optimized parameters. **a** 5 × 5 generated game with the target game set to Rock–Paper–Scissors (3 × 3). **b** 13 × 13 generated game with the target game set to Elo (1000 × 1000). **c** 13 × 13 generated game with the target game set to the mixture of Rock–Paper–Scissors (3 × 3) and Elo game (1000 × 1000). **d** 19 × 19 generated game with the target game set to the mixture of AlphaStar League and Go (board size = 3). Strategies are sorted by mean payoffs in **b** and **c** to more easily identify transitive structures expected from an Elo game.

interesting combinations of games of different sizes and rule-sets (e.g., as in Fig. 10d, which generates games targeting a mixture of Go (board size = 3) and the AlphaStar League), and subsequent training of AI agents using such a curriculum of generated games. Moreover, we can observe interesting trends when analyzing the Nash equilibria associated with the class of normal-form games considered here (as detailed in Supplementary Note 2). Overall, these examples illustrate a key benefit of the proposed graph-theoretic measures in that it captures the underlying structure of various classes of games. The characterization of games achieved by our approach directly enables the navigation of the associated games landscape to generate never-before-seen instances of games with fundamentally related structure.

## Discussion

In 1965, mathematician Alexander Kronrod stated that "chess is the Drosophila of artificial intelligence"[8], referring to the genus of flies used extensively for genetics research. This parallel drawn to biology invites the question of whether a family, order, or, more concretely, shared structures linking various games can be identified. Our work demonstrated a means of revealing this topological structure, extending beyond related works investigating this question for small classes of games (e.g., 2 × 2 games[40,85,101]). We believe that such a topological landscape of games can help to identify and generate related games of interest for AI agents to tackle, as targeted by the Problem Problem, hopefully significantly extending the reach of AI system capabilities. As such, this paper

presented a comprehensive study of games under the lens of graph theory and empirical game theory, operating on the response graph of any game of interest. The proposed approach applies to general-sum, many-player games, enabling richer understanding of the inherent relationships between strategies (or agents), contraction to a representative (and smaller) underlying game, and identification of a game's inherent topology. We highlighted insights offered by this approach when applied to a large suite of games, including canonical games, empirical games consisting of trained agent policies, and real-world games consisting of representative sampled policies, extending well beyond typical characterizations of games using raw payoff visualizations, cardinal measures such as strategy or game tree sizes, or strategy rankings. We demonstrated that complexity measures associated with the response graphs analyzed correlate well to the computational complexity of solving these games, and importantly enable the visualization of the landscape of games in relation to one another (as in Fig. 1). The games landscape exposed here was then leveraged to procedurally generate games, providing a principled means of better understanding and facilitating the solution of the so-called Problem Problem.

While the classes of games generated in this paper were restricted to the normal-form (e.g., generalized variants of Rock–Paper–Scissors), they served as an important validation of the proposed approach. Specifically, this work provides a foundational layer for generating games that are of interest in a richer context of domains. In contrast to some of the prior works in taxonomization of games (e.g., that of Bruns[40], Liebrand[39], Rapoport and Guyer[38]), a key strength of our approach is that it does not rely on human expertize, or manual isolation of patterns in payoffs or equilibria to compute a taxonomy over games. We demonstrate an example of this in Supplementary Note 2, by highlighting similarities and differences of our taxonomization with those of Bruns[38] over a set of 144 2 × 2 games. Importantly, these comparisons highlight that our response graph-based approach does not preclude more classical equilibrium-centric analysis from being conducted following clustering, while also avoiding the need for a human-in-the-loop analysis of equilibria for classifying the games themselves.

While our graph-based game analysis approach (i.e., the spectral analysis and clustering technique) applies to general-sum, many-player games, the procedural game structure generation approach used in our experiments is limited to zero-sum games (due to our use of the mElo parameterization). However, any other general-sum payoff parameterization approaches (e.g., even direct generation of the payoff entries) can also be used to avoid the zero-sum constraint. The study of normal-form games continues to play a prominent role in the game theory and machine learning literature[102–108] and as such the procedural generation of normal-form games can play an important role in the research community. An important line of future work will involve investigating means of generalizing this approach to generation of more complex classes of games. Specifically, one way to generate more complex underlying games would be to parameterize core mechanisms of such a games class (either explicitly, or via mechanism discovery, e.g., using a technique such as that described by Charity et al.[88]). Subsequently, one could train AI agents on a population of such games, constructing a corresponding empirical game, and using the response graph-based techniques used here to analyze the space of such games (e.g., in a manner reminiscent of Wang et al.[35], though under our graph-theoretic lens). The connection to the SBPCG literature mentioned in The Problem Problem and Procedural Game Structure Generation section also offers an avenue of alternative investigations into game structure generation, as the variations of fitness

measures and representations previously explored in that literature[34] may be considered in lieu of the approach used here.

Moreover, as the principal contribution of this paper was to establish a graph-theoretic approach for investigating the landscape of games, we focused our investigation on empirical analysis of a large suite of games. As such, for larger games, our analysis relied on sampling of a representative set of policies to characterize them. An important limitation here is that the empirical game-theoretic results are subject to the policies used to generate them. While our sensitivity analysis (presented in Supplementary Note 2) seems to indicate that the combination of our policy sampling scheme and analysis pipeline produce fairly robust results, this is an important factor to revisit in significantly larger games. Specifically, such a policy sampling scheme can be inherently expensive for extremely large games, making it important to further investigate alternative sampling schemes and associated sensitivities. Consideration of an expanded set of such policies (e.g., those that balance the odds for players by ensuring a near-equal win probability) and correlations between the empirical game complexity and the complexity of the underlying policy representations (e.g., deep vs. shallow neural networks or, whenever possible, Boolean measures of strategic complexity[109,110]) also seem interesting to investigate. Moreover, formal study of the propagation of empirical payoff variance to the topological analysis results is another avenue of future interest, potentially using techniques similar to Rowland et al.[51]. Another direction of research for future work is to analyze agent-vs.-task games (e.g., those considered in Balduzzi et al.[41]) from a graph-theoretic lens. Finally, our approach is general enough to be applicable to other areas in social and life sciences[111–115], characterized by complex ecologies often involving a large number of strategies or traits. In particular, these processes may be modeled through the use of large-scale response graphs or invasion diagrams[116–120], whose overall complexity (and how it may vary with the inclusion or removal of species, strategies, and conflicts) is often hard to infer.

Overall, we believe that this work paves the way for related investigations of theoretical properties of graph-based games analysis, for further scientific progress on the Problem Problem and task theory, and further links to related works investigating the geometry and structure of games[37,40,42,85,101,121].

## Methods

**Games.** Our work applies to $K$-player, general-sum games, wherein each player $k \in [K]$ has a finite set $S^k$ of pure strategies. The space of pure strategy profiles is denoted $S = \prod_k S^k$, where a specific pure strategy profile instance is denoted $s = (s^1, ..., s^K) \in S$. For a give profile $s \in S$, the payoffs vector is denoted $\mathbf{M}(s) = (\mathbf{M}^1(s), ..., \mathbf{M}^K(s)) \in \mathbb{R}^K$, where $\mathbf{M}^k(s)$ is the payoff for each player $k \in [k]$. We denote by $s^{-k}$ the profile of strategies used by all but the $k$-th player. A game is said to be zero-sum if $\sum_k \mathbf{M}^k(s) = 0$ for all $s \in S$. A game is said to be symmetric if all players have the same strategy set, and $\mathbf{M}^k(s^1, ..., s^K) = \mathbf{M}^{\rho(k)}(s^{\rho(1)}, ..., s^{\rho(K)})$ for all permutations $\rho$, strategy profiles $s$, and player indices $k \in [K]$.

**Empirical games.** For the real-world games considered (e.g., Go, Tic–Tac–Toe, etc.), we conduct our analysis using an empirical game-theoretic approach[67,68,122–124]. Specifically, rather than consider the space of all pure strategies in the game (which can be enormous, even in the case of, e.g., Tic–Tac–Toe), we construct an empirical game over meta-strategies, which can be considered higher-level strategies over atomic actions. In empirical games, a meta-strategy $s^k$ for each player $k$ corresponds to a sampled policy (e.g., in the case of our real-world games examples), or an AI agent (e.g., in our study of AlphaGo, where each meta-strategy was a specific variant of AlphaGo). Empirical game payoffs are calculated according to the win/loss ratio of these meta-strategies against one another, over many trials of the underlying games. From a practical perspective, game-theoretic analysis applies to empirical games (over agents) in the same manner as standard games (over strategies); thus, we consider strategies and agents as synonymous in this work. Overall, empirical games provide a useful abstraction of the underlying game that enables the study of significantly larger and more complex interactions.

**Finite population models and $\alpha$-Rank.** In game theory[125–129], one often seeks algorithms or models for evaluating and training strategies with respect to one another (i.e., models that produce a score or ranking over strategy profiles, or an equilibrium over them). As a specific example, the Double Oracle algorithm[73], which is used to quantify the computational complexity of solving games in some of our experiments, converges to Nash equilibria, albeit only in two-player zero-sum games. More recently, a line of research has introduced and applied the $\alpha$-Rank algorithm[49–51] for evaluation of strategies in general-sum, n-player many-strategy games. $\alpha$-Rank leverages notions from stochastic evolutionary dynamics in finite populations[118,130–134] in the limit of rare mutations[117,120,135], which are subsequently analyzed to produce these scalar ratings (one per strategy or agent). At a high level, $\alpha$-Rank models the probability of a population transitioning from a given strategy to a new strategy, by considering the additional payoff the population would receive via such a deviation. These evolutionary relations are considered between all strategies in the game, and are summarized in its so-called response graph. $\alpha$-Rank then uses the stationary distribution over this response graph to quantify the long-term propensity of playing each of the strategies, assigning a scalar score to each.

Overall, $\alpha$-Rank yields a useful representation of the limiting behaviors of the players, providing a summary of the characteristics of the underlying game–albeit a 1-dimensional one (a scalar rating per strategy profile). In our work, we exploit the higher-dimensional structural properties of the $\alpha$-Rank response graph, to make more informed characterizations of the underlying game, rather than compute scalar rankings.

**Response graphs.** The $\alpha$-Rank response graph[50] provides the mathematical model that underpins our analysis. It constitutes an analog (yet, not equivalent) model of the invasion graphs used to describe the evolution dynamics in finite populations in the limit when mutations are rare (see, e.g., Fudenberg and Imhof[117], Hauert et al.[136], Van Segbroeck et al.[120], Vasconcelos et al.[119]). In this small-mutation approximation, directed edges stand for the fixation probability[131] of a single mutant in a monomorphic population of resident individuals (the vertices), such that all transitions are computed through a processes involving only two strategies at a time. Here, we use a similar approach. Let us consider a pure strategy profile $s = (s^1, \ldots, s^K)$. Consider a unilateral deviation (corresponding to a mutation) of player $k$ from playing $s^k \in S^k$ to a new strategy $\sigma^k \in \tilde{S}^k$, thus resulting in a new profile $\sigma = (\sigma^k, s^{-k})$. The response graph associated with the game considers all such deviations, defining transition probabilities between all pairs of strategy profiles involving a unilateral deviation. Specifically, let $\mathbf{E}_{s,\sigma}$ denote the transition probability from $s$ to $\sigma$ (where the latter involves a unilateral deviation), defined as

$$\mathbf{E}_{s,\sigma} = \begin{cases} \eta \frac{1-\exp\left(-\alpha\left(\mathbf{M}^k(\sigma)-\mathbf{M}^k(s)\right)\right)}{1-\exp\left(-\alpha m\left(\mathbf{M}^k(\sigma)-\mathbf{M}^k(s)\right)\right)} & \text{if } \mathbf{M}^k(\sigma) \neq \mathbf{M}^k(s) \\ \frac{\eta}{m} & \text{otherwise,} \end{cases} \tag{1}$$

where $\eta$ is a normalizing factor denoting the reciprocal of the total number of unilateral deviations from a given strategy profile, i.e., $\eta = \left(\sum_{l=1}^{K}(|S^l|-1)\right)^{-1}$. Furthermore, $\alpha \geq 0$ and $m \in \mathbb{N}$ are parameters of the underlying evolutionary model considered and denote, respectively, to the so-called selection pressure and population size.

To further simplify the model and avoid sweeps over these parameters, we consider here the limit of infinite-$\alpha$ introduced by Omidshafiei et al.[50], which specifies transitions from lower-payoff profiles to higher-payoff ones with probability $\eta(1-\varepsilon)$, the reverse transition with probability $\eta\varepsilon$, and transition between strategies of equal payoff with probability $\frac{\eta}{2}$, where $0 < \varepsilon \ll 1$ is a small perturbation factor. We use $\varepsilon = 1e-10$ in our experiments, and found low sensitivity of results to this choice given a sufficiently small value. For further theoretical exposition of $\alpha$-Rank under this infinite-$\alpha$ regime, see Rowland et al.[51]. Given the pairwise strategy transitions defined as such, the self-transition probability of $s$ is subsequently defined as,

$$\mathbf{E}_{s,s} = 1 - \sum_{\substack{k \in [K] \\ \sigma | \sigma^k \in S^k \setminus \{s^k\}}} \mathbf{E}_{s,\sigma}. \tag{2}$$

As mentioned earlier, if two strategy profiles $s$ and $\sigma$ do not correspond to a unilateral deviation (i.e., differ in more than one player's strategy), no transition occurs between them under this model (i.e., $\mathbf{E}_{s,\sigma} = 0$).

The transition structure above is informed by particular models in evolutionary dynamics as explained in detail in Omidshafiei et al.[50]. The introduction of the perturbation term $\varepsilon$ effectively ensures the ergodicity of the associated Markov chain with row-stochastic transition matrix $\mathbf{E}$. This transition structure then enables definition of the $\alpha$-Rank response graph of a game.

## Definition 1

(Response graph) The response graph of a game is a weighted directed graph (digraph) $G = (S, \mathbf{E})$ where each node corresponds to a pure strategy profile $s \in S$, and each weighted edge $\mathbf{E}_{s,\sigma}$ quantifies the probability of transitioning from profile $s$ to $\sigma$.

For example, the response graph associated with a transitive game is visualized in Fig. 3b, where each node corresponds to a strategy $s$, and directed edges indicate

transition probabilities between nodes. Omidshafiei et al.[50] define $\alpha$-Rank $\pi \in \Delta^{|S|-1}$ as a probability distribution over the strategy profiles $S$, by ordering the masses of the stationary distribution of $\mathbf{E}$ (i.e., solution of the eigenvalue problem $\pi^T \mathbf{E} = \pi^T$). Effectively, the $\alpha$-Rank distribution quantifies the average amount of time spent by the players in each profile $s \in S$ under the associated discrete-time evolutionary population model[117]. Our proposed methodology uses the $\alpha$-Rank response graphs in a more refined manner, quantifying the structural properties defining the underlying game, as detailed in the workflow outlined in Fig. 3 and, in more detail, below.

**Spectral, clustered, and contracted response graphs.** This section details the workflow used to for spectral analysis of games' response graphs (i.e., the steps visualized in Fig. 3c to e). Response graphs are processed in two stages: (i) symmetrization (i.e., transformation of the directed response graphs to an associated undirected graph), and (ii) subsequent spectral analysis. This two-phase approach is a standard technique for analysis of directed graphs, which has proved effective in a large body of prior works (see Malliaros and Vazirgiannis[137], Van Lierde[138] for comprehensive surveys). In addition, spectral analysis of the response graph is closely-associated with the eigenvalue analysis required when solving for the $\alpha$-Rank distribution, establishing a shared formalism of our techniques with those of prior works.

Let $\mathbf{A}$ denote the adjacency matrix of the response graph $G$, where $\mathbf{A} = \mathbf{E}$ as $G$ is a directed weighted graph. We seek a transformation such that response graph strategies with similar relationships to neighboring strategies tend to have higher adjacency with one another. Bibliometric symmetrization[139] provides a useful means to do so in application to directed graphs, whereby the symmetrized adjacency matrix is defined $\widetilde{\mathbf{A}} = \mathbf{A}\mathbf{A}^T + \mathbf{A}^T\mathbf{A}$. Intuitively, in the first term, $\mathbf{A}\mathbf{A}^T$, the $(s, \sigma)$-th entry captures the weighted number of other strategies that both $s$ and $\sigma$ would deviate to in the response graph $G$; the same entry in the second term, $\mathbf{A}^T\mathbf{A}$, captures the weighted number of other strategies that would deviate to both $s$ and $\sigma$. Hence, this symmetrization captures the relationship of each pair of response graph nodes $(s, \sigma)$ with respect to all other nodes, ensuring high values of weighted adjacency when these strategies have similar relational roles with respect to all other strategies in the game. More intuitively, this ensures that in games such as Redundant Rock–Paper–Scissors (see Fig. 5, first row), sets of redundant strategies are considered to be highly adjacent to each other.

Following bibliometric symmetrization of the response graph, clustering proceeds as follows. Specifically, for any partitioning of the strategy profiles $S$ into sets $S_1 \subset S$ and $\bar{S}_1 = S \setminus S_1$, define $w(S_1, \bar{S}_1) = \sum_{s \in S_1, \sigma \in \bar{S}_1} \mathbf{E}_{s,\sigma}$. Let the sets of disjoint strategy profiles $\{S_k\}_{k \in [K]}$ partition $S$ (i.e., $\bigcup_{k \in [K]} S_k = S$). Define the $K$-cut of graph $G$ under partitions $\{S_k\}_{k \in [K]}$ as

$$cut(\{S_k\}) = \sum_k w(S_k, \bar{S}_k) , \tag{3}$$

which, roughly speaking, measures the connectedness of points in each cluster; i.e., a low cut indicates that points across distinct clusters are not well-connected. A standard technique for cluster analysis of graphs is to choose the set of $K$ partitions, $\{S_k\}_{k \in [K]}$, which minimizes (Eq. 3). In certain situations, balanced clusters (i.e., clusters with similar numbers of nodes) may be desirable; here, a more suitable metric is the so-called normalized $K$-cut, or $Ncut$, of graph $G$ under partitions $\{S_k\}_{k \in [K]}$,

$$Ncut(\{S_k\}) = \sum_k \frac{w(S_k, \bar{S}_k)}{w(S_k, S)} . \tag{4}$$

Unfortunately, the minimization problem associated with Eq. (4) is NP-hard even when $K = 2$ (see Shi and Malik[140]). A typical approach is to consider a spectral relaxation of this minimization problem, which corresponds to a generalized eigenvalue problem (i.e., efficiently solved via standard linear algebra); interested readers are referred to Shi and Malik[140], Van Lierde[138] for further exposition. Define the Laplacian matrix $\mathbf{L} = \mathbf{D} - \widetilde{\mathbf{A}}$ (respectively, $\mathbf{L} = \mathbf{I} - \mathbf{D}^{-\frac{1}{2}}\widetilde{\mathbf{A}}\mathbf{D}^{-\frac{1}{2}}$), where degree matrix $\mathbf{D}$ has diagonal entries $D_{i,i} = \sum_j \widetilde{\mathbf{A}}_{i,j}$, and zeroes elsewhere. Then the eigenvectors associated with the lowest nonzero eigenvalues of $\mathbf{L}$ provide the desired spectral projection of the datapoints (i.e., spectral response graph), with the desired number of projection dimensions corresponding to the number of eigenvectors kept. We found that using the unnormalized graph Laplacian $\mathbf{L} = \mathbf{D} - \widetilde{\mathbf{A}}$ yielded intuitive projections in our experiments, which we visualize 2-dimensionally in our results (e.g., see Fig. 3c).

The relaxed clustering problem detailed above is subsequently solved by application of a standard clustering algorithm to the spectral-projected graph nodes. Specifically, we use agglomerative average-linkage clustering in our experiments (see Rokach and Maimon[141], Chapter 15 for details). For determining the appropriate number of clusters, we use the approach introduced by Pham et al.[142], which we found to yield more intuitive clusterings than the gap statistic[143] for the games considered.

Following computation of clustered response graphs (e.g., Fig. 3d), we contract clustered nodes (summing edge probabilities accordingly), as in Fig. 3e. Note that for clarity, our visualizations only show edges corresponding to transitions from lower-payoff to higher-payoff strategies in the standard, spectral, and clustered

response graphs, as these bear the majority of transition mass between nodes; reverse edges (from higher-payoff to lower-payoff nodes) and self-transitions are not visually indicated, despite being used in the underlying spectral clustering. The exception is for contracted response graphs, where we do visualize weighted edges from higher-payoff to lower-payoff nodes; this is due to the node contraction process potentially yielding edges with non-negligible weight in both directions.

**Low-dimensional landscape and game generation**. We next detail the approach used to compute the games landscape and to procedurally generate games, as, respectively, visualized in Figs. 1 and 10.

To compute the low-dimensional games landscape, we use principal component analysis (PCA) of key features associated with the games' response graphs; we use a collection of features we found to correlate well with the underlying computational complexity of solving these games (as detailed in the Results section).

Specifically, using the response graph $G$ of each game, we compute in-degrees and out-degrees for all nodes, the entropy of the $\alpha$-Rank distribution $\boldsymbol{\pi}$, and the total number of 3-cycles. We normalize each of these measures as follows: dividing node-wise in-degrees and out-degrees via the maximum possible degrees for a response graph of the same size; dividing the $\alpha$-Rank distribution entropy via the entropy of the uniform distribution of the same size; finally, dividing the number of 3-cycles via the same measure for a fully connected directed graph.

We subsequently construct a feature vector consisting of the normalized $\alpha$-Rank distribution entropy, the normalized number of 3-cycles, and statistics related to normalized in-degrees and out-degrees. Specifically, we consider the mean, median, standard deviation, skew, and kurtosis of the in-degrees and out-degrees across all response graph nodes, similar to the NetSimile[84] approach, which characterized undirected graphs. This yields a feature vector of fixed size for all games. We subsequently conduct a PCA analysis of the resulting feature vectors, visualizing the landscape in Fig. 1 via projection of the feature vectors onto the top two principal components, yielding a low-dimensional embedding $\boldsymbol{v}_g$ for each game $g$.

**Procedural game structure generation**. The games generated in Figs. 2 and 10 use the multidimensional Elo (mElo) parametric structure[41], an extension of the classical Elo[144] rating system used in Chess and other games. In mElo games, each strategy $i$ is characterized by two sets of parameters: (i) a scalar rating $r_i \in \mathbb{R}$ capturing the strategy's transitive strength, and (ii) a $2k$-dimensional vector $\boldsymbol{c}_i$ capturing the strategy's intransitive relations to other strategies. The payoff a strategy $i$ receives when played against a strategy $j$ in a mElo game is defined by $M(i,j) = \sigma(r_i - r_j + \boldsymbol{c}_i^T \Omega \boldsymbol{c}_j)$ where $\sigma(z) = (1 + \exp(-z))^{-1}$, $\Omega = \sum_{i=1}^{k} (\boldsymbol{e}_{2i-1}\boldsymbol{e}_{2i}^T - \boldsymbol{e}_{2i}\boldsymbol{e}_{2i-1}^T)$, and $\boldsymbol{e}_i$ is the unit vector with coordinate $i$ equal to 1. This parametric structure is particularly useful as it enables definition of a wide array of games, ranging from those with fully transitive strategic interactions (e.g., those with a single dominant strategy, as visualized in Fig. 2a), to intransitive interactions (e.g., those with cyclical relations, as visualized in Fig. 2d), to a mix thereof.

The procedural game structure generation visualized in Fig. 10 is conducted as follows. First, we compute the low-dimensional game embeddings for the collection of games of interest, as detailed above. Next, for an initially randomly-generated mElo game of the specified size and rank $k$, we concatenate the associated mElo parameters $r_i$ and $\boldsymbol{c}_i$ for all strategies, yielding a vector of length $|S|(1 + 2k)$ fully parameterizing the mElo game, and constituting the decision variables of the optimization problem used to generate new games. We used a rank 5 mElo parameterization for all game generation experiments. For any such setting of mElo parameters, we compute the associated mElo payoff matrix $M$, then the associated response graph and features, and finally project these features onto the principal components previously computed for the collection of games of interest, yielding the projected mElo components $\boldsymbol{v}_{mElo}$.

Subsequently, given a game $g$ of interest that we would like to structurally mimic via our generated mElo game, we use a gradient-free optimizer, CMA-ES[100], to minimize $||\boldsymbol{v}_{mElo} - \boldsymbol{v}_g||_2^2$ by appropriately setting the mElo parameters. For targeting mixtures of games (e.g., as in Fig. 10c, d), we simply use a weighted mixture of their principal components $\boldsymbol{v}_g$ (with equal weights used in our experiments). We found the open-source implementation of CMA-ES[145] to converge to suitable parameters within 20 iterations for all experiments, with the exception of the larger game generation results visualized in Fig. 10d, which required 40 iterations.

**Statistics**. To generate the distributions of Double Oracle iterations needed to solve the motivating examples (Fig. 2), we used 20 generated games per class (transitive, cyclical, random), showing four examples of each in Fig. 2a, d and g. For each of these 20 games, we used 10 random initializations of the Double Oracle algorithm, reporting the full distribution of iterations. To generate the complexity results in Fig. 9, we likewise used 10 random initializations of Double Oracle per game, with standard deviations shown in the scatter plots (which may require zooming in). For the Spearman correlation coefficients shown in each of Fig. 9a to c, the reported $p$-value is two-sided and rounded to two decimals.

**Data availability**
We use OpenSpiel[146] as the backend providing many of the games and associated payoff datasets studied here (see Supplementary Methods 1 for details). Payoff datasets for empirical games in the literature are referenced in the main text.

**Code availability**
We use OpenSpiel[146] for the implementation of $\alpha$-Rank and Double Oracle.

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

## Acknowledgements
The authors gratefully thank Marc Lanctot and Thore Graepel for insightful feedback on the paper. F.C.S. acknowledges the support of FCT-Portugal through grants PTDC/MAT/STA/3358/2014 and UIDB/50021/2020. The authors thank the three anonymous reviewers and editor for their constructive and insightful feedback, which helped to significantly improve the clarity of the proposed method, experimental results, and discussions thereof.

## Author contributions
S.O., K.T., and F.C.S. conceptualized the graph-based framework used in the paper. S.O. implemented and generated the experimental results (spectral analysis of response graphs, clusterings, landscape of games, procedural games generation). W.M.C. devised and implemented the policy sampling scheme for real world games, and generated the payoff tables for the games listed in Fig. 1. S.O., K.T., W.M.C., F.C.S., M.R., J.C., D.H., P.M., J.P., B.D.V., A.G., and R.M. contributed actively to discussions, analysis, writing, and review of the paper.

## Competing interests
The authors declare no competing interests.
