## [Peer Review File · Nature Communications]

Reviewer #1 (Remarks to the Author):

This paper seeks to classify the landscape of games by analyzing relations among strategies of the games using graph analysis. Based on the categorization, the paper also shows preliminary results for generating new games that could be interesting for training AI agents (establish a sort of curriculum).

The topic itself is interesting, novel, and important. By classifying games and understanding what makes them interesting, difficult, or strategically interesting, one can potentially identify how to train and evaluate AI algorithms. There have been attempts to classify simple games, but classifying complex games is difficult.

I certainly learned from reading this paper — I enjoyed reading it. It took me a little bit to figure out the authors' strategy for classifying games, but once it clicked for me I did like it quite a bit. The authors have articulated a clever idea.

However, I'm hung up on a number of different points, each of which keeps me from enthusiastically endorsing the paper in its current form:

1. For the smaller canonical games, it seems like there needs to be greater validation for why describing the landscape of games in this way is useful. Perhaps the authors could compare the resulting similarity of games from their algorithms with those of another game-classification scheme. The work of Bruns (2015) ("Names for Games: Locating 2x2 Games"), which builds on the prior work of others, could be one possibility, albeit it only works for 2x2 games. Still, does the proposed classification scheme produce similar results to Bruns' classification scheme in those simple games? If not, why not? I guess the bottom line is that I'm not certain about the extent of the usefulness of the "classification scheme".

2. For the larger games, I'm really hung up by the fact that you get different results when you use different strategies. The authors state that this attribute means that they have addressed the "Policy Problem" rather than the "Problem Problem." To me, that indicates that the paper doesn't achieve what it set out to achieve (which I thought was classifying games). Is there a way to consider strategies that will provide invariance to the classification scheme. If not, I see that as a severe limitation.

3. The results in classifying games seems biased toward a single issue, which appears to be described in the idea that games are "solved" by finding a Nash equilibrium. Is that the only thing that matters in classifying the "landscape of games." Are the kinds of games the authors are looking at shining too much focus on that idea? For example, does the technique described indicate anything about what is required for cooperation (or the possibility for cooperation) to happen? This is just one example I can think of off the top of my head (there are others). The question is whether or not this analysis really captures the defining characteristic of games? My inclination is that the identified technique could be one way to classify games, but it isn't yet clear what the strengths and weaknesses of it are.

4. It isn't clear to me that the new games generated by the method are all that useful. They may be, it just isn't obvious to me as the reader.

Other minor comments:

Page 1:

- primarily "seeked" -> primarily sought

- "While the importance of games as a natural research platform for development of learning algorithms and the measurement of progress in AI is well-established, ..."

Doesn't seem well-established to me. It's certainly used a lot. But some would argue that they have been a detriment to AI development and evaluation. Chollet certainly argues very convincingly against it (see "On the measure of intelligence").

Page 2:

- Figure 1: When I first read the paper, I had a hard time understanding this figure and what it was saying. After reading the paper, I think I get it, though it seems like quite the simplification (that could be good or bad). Also, the term "real-world games" applied to things like Connect-Four and Tic-Tac-Toe seems strange to me.

- "Hierarchy of complexity" — what does that mean? What makes something complex, in general? Given the large amount of brilliant researchers that have tried and failed to define it (unless the authors are referring to computational complexity), my instant thought was that this problem is not likely to be easily solved.

Page 3:

- Early on, I wasn't sure how to interpret the authors' use of the word "intransitive," until I understood that they were talking about the evaluation of strategies. I probably should have caught on earlier, but I didn't.

- Last paragraph of the introduction: Is neural network + a good curriculum all there is to intelligence? Seems like a vast simplification.

Page 4:

- As I read the paper, I kept getting tripped up by what strategies the authors were referring to. I got it after reading further, but it was confusing to me until I got it several several pages after.

- Equating "solving a game" to "finding a Nash equilibrium" gives me pause. What if the game has many Nash Equilibria? What if other players don't play the equilibria I'm considering?

Page 8:

- Figure 7: I can't see the arrows on the "contracted graphs". Same goes for some other graphs

Page 11:

- I think the complexity results are interesting, albeit perhaps intuitive. I'm not sure about the normalization though. Seems like that could bias the results. I think I don't sufficiently understand the implications of the normalization.

Reviewer #2 (Remarks to the Author):

This paper presents a new pipeline for analyzing general-sum games, starting from their payoff matrices, as obtained from the set of strategies for canonical games or a sample of policies (AI agents) for other richer ("empirical") games. The technique is illustrated with a few examples of

canonical and empirical games.

If AI systems are evaluated by the problems they can and cannot solve we have to understand these problems better. K-player games represent a particular kind of problem that has received enormous attention in AI, reinforced by a strong connection with game theory. Then, it is very important to produce novel and insightful ways of analyzing the so-called landscape of games. This has been vindicated in the past as well.

However, this paper falls short in this aspiration, and even the expectations set in the abstract and the introduction. The motivation seems to be supported by understanding how "interesting a game is for an artificial agent". The concept of being "interesting" is diluted in the introduction and only recovered at the end. No definition or metric of interestingness is introduced in the paper, not even the most classical ones, at the purely cognitive level, such as those related to ability and difficulty (a problem is interesting for an agent if it is slightly around its ability, i.e., not too easy not too difficult). Even if we understand interestingness in other ways, for instance in terms of symmetries or cycles in the payoff matrix, it's unclear whether the techniques in this paper help answer the interestingness question, either for educational purposes (training ML agents) or for the "progress" of AI.

Also, the kind of games explored in this paper is somewhat limited. Most challenges in AI today are about perceptual, cognitive and generalization issues. Whether games are transitive or cyclic produce insights for a very narrow take about games (and only some of the games). Many other games and tasks that are not K-player are left out from this study, simply because the payoff matrix is different or captures very little about the task dynamics. For instance, a one-player video game benchmark (e.g., ALE) is represented by the scores produced on these games, and then the analysis of transitivity is different, only non-trivial when analyzing several instances of the same game. As there are several instances of the same pairing (player - game, or player - player), we should also analyze agent reliability. This is ignored altogether in this paper by choosing the mean of the number of times A beats B but not the variance, but it may go beyond winning or losing, but winning or succeeding in a game consistently and with a large margin.

Finally, the analysis is made for the set of basic (non-deterministic) strategies in simple games or experimental strategies (policies as a population of AI agents trained with different algorithms and parameters). This gap is important, as in the second case we derive all the metrics and plots conditioned to the choice of the population of artificial agents. A different choice of agents may lead to a different characterization of the game. This is a very important point, and the paper should analyze how robust the analysis is when the population of agents is derived in a different way. For instance, there are papers analyzing AI (games) competitions, and the results about how games "behave" are subject to the participants of the competition. This subjectivity or relativity is unavoidable, so it is very important to clarify whether the topologies hold if one just changes the population by creating stochastic combinations of existing agents. This not only applies to "experimental" games, but also canonical games. For instance, rock paper scissors, or its simplified version of matching pennies, is cyclic, but it's also peculiar because a fully random agent (choosing between the two or three strategies randomly) cannot be beaten on expectation. Between any two agents one can construct any point in between. The key issue is that one can define infinitely many distributions of stochastic policies from a finite set of strategies and derive the whole pipeline presented in this paper using this distribution. The authors present some guidelines (and algorithm) to generate a 'representative' population in experimental games, but this seems to be devised to get what they want to get rather than a population that is truly representative of some canonical reference that would be achieved in large numbers. Actually, in the landscape of agents (which is dual to the landscape of games) one has to determine a way of considering some populations more meaningful than others, and this may be related to the resources or complexities of the policies.

The generation of games is restricted to a particular class and it is not easy to extrapolate to a much larger landscape of games or tasks that we need to explore, cluster and understand for AI progress.

I'm not sure that they generate really "new and interesting games", as they claim. In which way are they new and interesting?

About the particular methodology, it looks unnecessary complicated at places, with many choices about each step, with the overall feeling that all these steps, transformation algorithms, normalizations and parameters are chosen to get what the authors want to obtain, a picture such as the one in Fig. 1., with no ablation procedure or what could have been different. The authors talk about "findings" but I don't see any surprising thing in Fig. 1.

The paper misses some important connections in the analysis of the "problem problem" (the task theory or game characterization problem using previous terminologies), the analysis of AI benchmarks and competitions, and the notion of interestingness (and associated concepts such as game difficulty, and its relation to agent ability). See below for some of these references.

Because of all this, I think the paper doesn't portray a sufficiently novel, robust, general and insightful procedure for analyzing the landscape of games, fails to connect with AI more generally and doesn't make a convincing case that this would be more useful for its progress than existing, perhaps simpler, approaches.

***** DETAILED COMMENTS *****

In the introduction, it is clear that AI has mostly focused on solving tasks (what the authors awkwardly refer to as the "policy problem"), while the analysis of tasks and problems in AI has only been brought to mainstream attention more recently [1], in the context of the problem of AI evaluation. Here they seem to suggest that the critical question is to define interesting games for progress in AI, with the even more awkward term of "problem problem". This has been referred to by different other less awkward terms in the past, such as the problem of AI needing a "task theory" [2]. This may have the perspective of finding challenging (but not extremely challenging) problems for AI to be solved independently, but also to find curricula or other sequences of tasks that help with the training (or even education) of AI agents [3]. This is no different from psychological measurement and the choice of tasks: how they can be analyzed, arranged, created and chosen, and most especially, how latent factors of these tasks, extracted from the behavior of populations of humans, lead to hierarchies of tasks and clusters thereof using the extracted latent features. In the end, what the authors do in this paper is extracting some latent features that they plot using the top two principal components after PCA.

The relevant literature about the analysis of tasks, games or problems for AI is either not covered or just swiftly mentioned at the bottom of page 2. This includes a very wide range of papers that are cited but not discussed (where the relation to this work is clear, with some of them also exploring the space of tasks) and even some papers from the authors that also develop ways of extracting some topological structure. While I agree that graph theory can give very insightful transformations and representations, it is just a possible way of analyzing the payoff matrices. Many other ways abound in game theory, and we would need to find a comparison or a clear explanation of why this one in particular is advantageous or parsimonious.

Even in the domain of games, and board games more specifically, there is a long tradition of analyzing the structure of games or creating new ones, as done in general game playing [4] for board games, and related approaches. In many of these cases, the "artificial agents" were not learning agents originally (but most are nowadays). There are also several studies of the landscape of games in competitions (e.g., [5]), and even if they are not K-games, they can use pairwise matrices between agents and perform an analysis of transitivity [6]).

Then again, the final paragraph of the intro says that the paper "culminates in a demonstration of how

the topological structure over games can be used to tackle the interestingness question of the Problem Problem". I don't think this is the case. Also, I'm not sure that finding hybrids between two games in terms of the characteristics extracted in this paper is actually the key aspect for "curriculum learning". Rather, it is about the reuse of partial strategies and knowledge as the agent goes and learns through the curriculum. The generation of games as done here does not really solve the problem of finding increasing levels of perceptual or cognitive difficulty (beyond the naive versions of this problem, by adding noise or modifying payoffs and rewards), or at least why this is better than easier ways of analyzing difficulty in games using classical theories such as Item Return Theory [7].

The procedure concatenates a series of transformations to cluster the transitions that emerge from the payoff matrix. There are many choices there, such as the Double Oracle algorithm. For instance, this algorithm is based on planning, but is it really a good proxy for how hard it is to find the optimal strategy? And is this really meaningful? "The complexity of solving" these games happens at a very game-theoretical level, but this may not correspond to the complexity of solving the games that several AI techniques find with these games. My point is precisely that there are many ways to measure this complexity, if we go beyond scores. In AI agents using machine learning, this can be related to the number of iterations, complexity of the trained networks, and many other metrics. A similar thing happens with the spectral analysis. How would the result of many figures be affected by other techniques and transformations? And what would the effect be for Figure 1? Overall, we would need more justification of these choices as proxies of complexity or ways for summarizing/clustering the graphs. I'm not an expert in graph theory, but I can imagine that there are plenty of options, from the analysis of social networks to bibliometric analysis. They pick techniques and transformations from these areas, but the justification of the choices is missing, especially what the impact would have been if another choice had been made.

Figure 3 could include the colors in the payoff matrix as all the other figures. Also, these colors could be better explained the first time.

For the analysis of the canonical games, how does it affect if we include stochastic strategies? Would all the clustering be broken because we can always find a strategy that is anywhere between two strategies? This is partially seen for the empirical games, but there is a particular procedure for choosing the populations of agents (policies or strategies), explained on page 10 (2nd paragraph). It seems that the method works well with redundant agents, but what about continuums of agents between two existing agents? Actually, this is something that could be exploited rather than avoided by the agent sampling algorithm.

What I mean is that the characterization of a game should not depend on the choice of the population of strategies that is made, or at least we should be clear by defining a distribution on agents, and get topology 1 (figure 1) when using population (distribution) X, and get topology 2 (a new figure) when using a different population. This robustness or diversity of findings is completely missing here. For instance, the last paragraph of page 9 raises many doubts about how specific the graphs and plots are according to the choice. Actually, one ends up with the intuition that this is not solving the task theory problem at all.

In any case, a critical element of the whole paper is the algorithm for sampling the "set of representative policies", which is introduced in another paper. Why is it representative? What is the full space of all policies? If finite, why is this more representative than a uniform sampling? If infinite, how do we know the original distribution? I don't think that as more policies are sampled (independently of the distribution) all would converge to the same graphs and plots. If that's the case, the authors should prove it. If that's not the case, then the implication should be upfront (the procedure is subjective to this choice, and the notion of "representative" needs to be clarified).

The normalizations at the bottom of page 10 explain something that is narrated in a surprising tone in Figure 1 ("despite their payoff table sizes ranging..."). Of course, if one normalizes by these sizes

(which is okay), we could still get that games only differing on these size variations could cluster together. You get what you expect. But in the end, is the number of games in Figure 1 sufficient to determine that the whole procedure is not overfitted to get the nice picture?

Figure 9 gives a somewhat troubling picture. Of course, there is correlation, but why is it rock-paper-scissors (no. 15) the most complex game? Well, this is according to Double Oracle, but still, is this a challenging or interesting game for AI? It basically lacks an optimal policy. It is when the other agents do not behave stochastically in an iterated version (with memory about the opponent) that it may become somewhat interesting and complex (but this is not the way it is analyzed here). This is a very particular game, because a player can only exploit the patterns of other agents, but figure 9, in my opinion, shows that the notion of complexity here is not very useful. At least the response graph complexity measures give more information and one of them puts rock-paper-scissors at the bottom. But this figure 9 asks more questions than answers! Tts smávegis

On page 11 we enter the "landscape of games". One would expect that navigating this landscape would be playing with the rules of the games, as the GGP setting does, but it plays with the payoff matrices. This is a restrictive view of the landscape of games and only captures a part of their behavior. Games are generated at this abstract and limited level (normal-form), but I don't see that really *new* and *interesting* games are created. They are just mixtures. (BTW, the authors consider mixtures of games, but not mixtures of agents, as mentioned above, which could have an effect on the characterization of games)

At the level of writing and structure, the paper is very well written, with only a few parts that are cryptic (especially in the methods) and no typos (I only found one at the end: "opens-source").

References:

- [1] A New AI Evaluation Cosmos: Ready to Play the Game? (<https://www.aaai.org/ojs/index.php/aimagazine/article/view/2748>)
- [2] Why Artificial Intelligence Needs a Task Theory (https://link.springer.com/chapter/10.1007/978-3-319-41649-6_12)
- [3] Task Analysis for Teaching Cumulative Learners (https://link.springer.com/chapter/10.1007/978-3-319-97676-1_3)
- [4] Genesereth, Michael; Love, Nathaniel; Pell, Barney (15 June 2005). "General Game Playing: Overview of the AAAI Competition". *AI Magazine*. 26 (2): 62. doi:10.1609/aimag.v26i2.1813.
- [5] D Perez-Liebana, J Liu, A Khalifa, RD Gaina, J Togelius, SM Lucas "General Video Game AI: A Multitrack Framework for Evaluating Agents, Games, and Content Generation Algorithms" *IEEE Transactions on Games* 11 (3), 195-214, 2019.
- [6] Nielsen, T. S., Barros, G. A., Togelius, J., & Nelson, M. J. (2015). Towards generating arcade game rules with VGD. In *Computational Intelligence and Games (CIG), 2015 IEEE Conference on*, pp. 185-192. IEEE.
- [7] Plumed et al. Dual Indicators to Analyse AI Benchmarks: Difficulty, Discrimination, Ability and

Reviewer #3 (Remarks to the Author):

This paper demonstrates a graph-theoretic method (through α -Rank, spectral analysis and clustering) that can simplify the representation of games and reveal aspects of their topological structure. Such structures can, in turn, be used as proxies/correlates of the computational complexity of a game. A number of these features (including α -Rank entropy, number of 3-cycles and so on) has been used by the authors as a feature space of a PCA that projects games onto a 2d (navigation) map.

Once such a game map is available one can use (and combine) the graph features of each game as fitness functions in order to generate new parametric structures of games.

The paper is well written and structured and offers an important and novel perspective for modelling and generating games. The experiments and the general methodology followed appear to be solid.

I have, however, a number of concerns that I feel they need to be addressed before the paper can be accepted (see detailed list below)

** Situating the work in the literature **

"also known as the 'Problem Problem' [37] - -> what you are referring to as "Problem Problem" (ref in 2019) is known as automated game design, or procedural content generation (which makes it to the title of a section later in the paper). While the "problem problem" (better put procedural content generation/procedural generation) exists for more than 30 years while a community within AI and Games had dedicated more than a decade of research efforts on automating the game design process and making games more appealing for players, and more interesting for AI/humans to play - e.g. see the following indicative work:

- AI and Games book (Yannakakis and Togelius) (chapter 4) and references – in particular, the section under game design/rules
- Search-based PCG paper
- Towards automatic personalized content generation for platform games, by Shaker et al

Moreover, automated level design processes have been used to design environments so that agents in games look more believable - e.g. see the work of Camilleri et al:

- Platformer level design for player believability.

In addition, a number of authors have focused recently on creating of adaptive environments (within games) to boost AI research: e.g. see:

- Obstacle tower: A generalization challenge in vision, control, and planning
- Procedural Content Generation: From Automatically Generating Game Levels to Increasing Generality in Machine Learning

The work of Cameron Browne on Yavalath and general board game design (e.g. see his Evolutionary game design paper) is also absent from your review and critical for two reasons

- First, it tackles automated board game design (rules and levels) that is interesting for humans to play (and AI presumably)
- Second, it offers a set of interesting heuristics (including depth) that could influence/inspire the

criteria you consider for the games you examine.

Early work by Nelson/Mateas and Togelius/Schmidhuber is also highly relevant:

- Nelson, Mark J., and Michael Mateas. "Towards automated game design." 2007.
- Togelius, Julian, and Jurgen Schmidhuber. "An experiment in automatic game design." 2008 IEEE Symposium On Computational Intelligence and Games. IEEE, 2008.

Given the above (nb. this is a non-inclusive list), the claim "the significance of the inverse problem remains largely underexposed" is not accurate. Please revise the whole rhetoric of these introductory paragraphs to better reflect the current state of the art in PCG and AI and Games.

**** Definition of Interestingness *****

"what it means for a game to be 'interesting' in the first place, or more fundamentally, how to characterize the topological landscape of games." This sentence views games as "interesting" from an AI perspective and a game-rule perspective; but games are primarily "interesting" for players/humans and have been investigated from several other perspectives beyond game rules. Interest in games, beyond its rules and environments - can be well attributed to its aesthetics, its story, its social impact, the opponent behaviours (e.g. see <https://core.ac.uk/download/pdf/132619524.pdf>) and so on. Please clarify and define interestingness from an AI lens here and state what are the primary factors you consider for defining AI-interestingness. In the next page, it is obvious that by "topographical landscape" you are referring to the landscape of game rules. It would be beneficial however if "interest" is defined properly here.

***** PCG *****

In section "The Problem Problem Revisited: Procedural Game Generation" the authors apply a search-based PCG algorithm (Togelius et al.) for generating new parametric structures (not game rules!) of mElo games based on an evolutionary process and driven by a number of selected objectives. I have two issues in this section

- 1) Authors need to acknowledge they run an SBPCG algorithm and place their approach within the SBPCG taxonomy (e.g. type or representation, type of quality evaluation etc).
- 2) From my understanding of the paper and the PCG method described the generated games are parametric structures (payoff maps) within the space of mElo games and not game rulesets per se. While the generation of parametric structures that resemble a particular game (or a mix of games) is an important step towards game generation, the paper needs to be very specific that the generated games presented here are indirect representations of games that could feature diverse sets of game rules. In that regard, Fig. 3 should also be adapted (game generation -- > parametric structure generation?)

***** Visualisation of the landscape and expressivity analysis in games ******

Earlier work that needs to be discussed in association to your 2D mapping includes the expressivity analysis studies by Smith and Whitehead (<https://dl.acm.org/doi/pdf/10.1145/1814256.1814260>) or the recent work on map-elites-based level/game generation e.g. by Charity et al: <https://arxiv.org/abs/2002.04733>

*****Title *****

I feel that the title is largely misleading as the proposed mapping/generation approach is limited to a particular set of games (zero-sum, multi-player adversarial games). The title should specify which type of games the approach can navigate (and potentially generate).

Minor

Fig 2: please define what S1-S9 are in the caption of the figure

Overall, characterization of the topological -- > Overall, the characterization of the topological

"Specifically, we use a collection of features we found to correlate well with the underlying computational complexity of solving these games (as detailed in the Results section). Specifically,..." --
> Specifically is used twice in subsequent sentences. Consider revising

Navigating the Landscape of Multiplayer Games to Probe the Drosophila of AI Author rebuttal

We thank the reviewers and editor for the opportunity to revise and resubmit our contribution, and for providing insightful and constructive comments. To better contextualize the overall feedback and minimize repetition, we first collect and respond to shared / high-level reviewer comments below. Subsequent sections provide point-by-point rebuttals to remaining individual concerns, with reviewer points numbered and hyperlinked for easier navigation.

All changes have been tracked in a different color in the revised manuscript for comparison of changes made (see `revision_tracked_changes.pdf` under ‘Article File’ in the online submission system, which includes *both* the main paper and supplementary information). A ‘clean’ version of the revised manuscript and supplementary information with tracked changes disabled is also provided as `revision_clean.pdf` under ‘Related Manuscript Files’ in the online submission system (recommended for easier reading).

General Reviewer Remarks (GR) and Responses

The following concerns points raised by at least 2 out of 3 reviewers, which we collectively address:

GR.1 – Focus on ‘solving’ games using Nash concept, and use of the Double Oracle algorithm. (Reviewers 1 & 2)

(Point R1.4): “The results in classifying games seems biased toward a single issue, which appears to be described in the idea that games are “solved” by finding a Nash equilibrium. Is that the only thing that matters in classifying the “landscape of games.” Are the kinds of games the authors are looking at shining too much focus on that idea? For example, does the technique described indicate anything about what is required for cooperation (or the possibility for cooperation) to happen? This is just one example I can think of off the top of my head (there are others). The question is whether or not this analysis really captures the defining characteristic of games? My inclination is that the identified technique could be one way to classify games, but it isn’t yet clear what the strengths and weaknesses of it are.”

(Point R1.13): “Equating “solving a game” to “finding a Nash equilibrium” gives me pause. What if the game has many Nash Equilibria? What if other players don’t play the equilibria I’m considering?”

(Point R2.13): “The procedure concatenates a series of transformations to cluster the transitions that emerge from the payoff matrix. There are many choices there, such as the Double Oracle algorithm. For instance, this algorithm is based on planning, but is it really a good proxy for how hard it is to find the optimal strategy? And is this really meaningful? “The complexity of solving” these games happens at a very game-theoretical level, but this may not correspond to the complexity of solving the games that several AI techniques find with these games. My point is precisely that there are many ways to measure this complexity, if we go beyond scores.”

Author response – We thank the reviewers for the feedback regarding clarifying the role these concepts and experiments play with respect to our overall contribution.

Our work is rooted in game theory, which is a mathematical theory of interactive decision-making that defines core concepts (e.g., Nash equilibria) and techniques to study both classical benchmark games (e.g., Rock–Paper–Scissors), and larger-scale games (e.g., Poker).

Regarding the focus on Nash equilibria, we first emphasize that the primary contribution of our paper is an approach for analyzing the topological properties (of the response graph) of multiplayer games. Having said this, a large body of literature in game theory and multiagent learning has focused on analyzing the computational complexity of searching for and/or learning solutions for these games (Daskalakis et al., 2009, 2010, Savani and von Stengel, 2004, Shoham et al., 2007, von Stengel, 2007). As such, a natural line of investigation in our work was to study the potential connections between the structure of a game (the primary topic studied in this paper) to the complexity of finding such solutions. In response to **R1.4**, we clarify that the motivation behind introducing the Nash equilibrium/double oracle-focused studies was to conduct such an example investigation, which we believe to be relevant for readers interested in these aspects. Overall, this line of experimentation was not intended to propose a specific definition of computational complexity (e.g., with respect to Nash equilibria) as being explicitly useful for defining a topology / classification over games. We also highlight that the Nash equilibrium concept and associated solvers are not used in the core contributions of the paper (i.e., the spectral analysis of individual games, and graph statistics-based analysis of collections of games as in Figure 1 of the submission).

Reviewer 1 rightly points out that an over-emphasis on Nash equilibria may raise further questions (e.g., regarding possibility of cooperation vs. competition). This is a valid concern. We now make sure to highlight that one of the major strengths of our approach (in contrast to prior works on games topology, e.g., Bruns (2015)) is that it avoids the need for a human-in-the-loop analysis of equilibria (e.g., win-win vs. win-loss equilibria) for classifying games, all while being scalable to large-scale many-player games. This is in contrast to prior works in games topology, e.g., Bruns (2015) “Names for Games: Locating 2×2 games”, which we now compare against (as later detailed in **R1.2**). Importantly, this does not preclude this type of equilibrium-centric analysis from being conducted in post-clustering analysis (see **R1.2**, where we show Bruns’ approaches coarsely cluster win-win games into one set, whereas ours offers a refinement). We agree with the reviewer that the strengths and weaknesses of the approach can be made clearer for readers. As such, we have expanded this discussion in the paper, and have explicitly added a ‘Strengths & Limitations’ subsection in the Discussions section.

In response to **R1.13**, please note that we focus on the notion of ‘solving a game’ as being synonymous with ‘finding a Nash equilibrium’, following the precedent set by works including:

- Heads-up limit hold’em poker is solved (Bowling et al., 2015)
- Solving Games with Functional Regret Estimation (Waugh et al., 2015)
- Solving Imperfect Information Games Using Decomposition (Burch et al., 2014)
- An iterative method for solving a game (Robinson, 1951)

We agree that this may be confusing for readers who associate ‘solving a game’ with alternative definitions (e.g., being able to predict the outcome in all game states). As such, we have added clarification of this to the text.

Similar to the focus on Nash equilibria above, we use the double oracle as one example solver to investigate the question of computational complexity. Double oracle has been well-established as a Nash equilibrium solver in multiagent and game theory literature (Bosansky et al., 2015, Jain et al., 2011, Lanctot et al., 2017, McMahan et al., 2003, Regan and Boutilier, 2009); it has also been essential for driving the notion of ‘solving’ multiplayer games via AI/learning algorithms, specifically through the Policy-Space Response Oracles algorithm (PSRO), which generalizes double oracle (Lanctot et al., 2017). PSRO has seen subsequent application to Poker, games involving more than 2 players (Muller et al., 2020), and more recently, a training approach related to PSRO applied to StarCraft II (Vinyals et al., 2019). We agree with Reviewer 2 (**R2.13**) that there are alternative means of measuring the ‘complexity’ of a game (e.g., through use of different solvers, alternative solution concepts, or by conducting theoretical complexity analysis in the same vein as Daskalakis et al. (2009)). However, we re-iterate that this is not the primary contribution of the paper, and double oracle simply provides us a useful (and popular) means of investigating computational questions, which we anticipate will arise from readers interested in solving and analyzing multiplayer games.

GR.2 – Sensitivity to the choice of empirical strategies (Reviewers 1 & 2)

(**Point R1.3**): “For the larger games, I’m really hung up by the fact that you get different results when you use different strategies. The authors state that this attribute means that they have addressed the “Policy Problem” rather than the “Problem Problem.” To me, that indicates that the paper doesn’t achieve what it set out to achieve (which I thought was classifying games). Is there a way to consider strategies that will provide invariance to the classification scheme. If not, I see that as a severe limitation.”

(**Point R2.17**): “What I mean is that the characterization of a game should not depend on the choice of the population of strategies that is made, or at least we should be clear by defining a distribution on agents, and get topology 1 (figure 1) when using population (distribution) X, and get topology 2 (a new figure) when using a different population. This robustness or diversity of findings is completely missing here. For instance, the last paragraph of page 9 raises many doubts about how specific the graphs and plots are according to the choice. Actually, one ends up with the intuition that this is not solving the task theory problem at all.”

(**Point R2.18**): “In any case, a critical element of the whole paper is the algorithm for sampling the “set of representative policies”, which is introduced in another paper. Why is it representative? What is the full space of all policies? If finite, why is this more representative than a uniform sampling? If infinite, how do we know the original distribution? I don’t think that as more policies are sampled (independently of the distribution) all would converge to the same graphs and plots. If that’s the case, the authors should prove it. If that’s not the case, then the implication should be upfront (the procedure is subjective to this choice, and the notion of “representative” needs to be clarified).”

Author response – We thank the reviewers for pointing out the need for additional analysis and discussion regarding sensitivity to the choice of policies used in the empirical games.

We first make some general remarks regarding the motivation behind the policy sampling scheme used. Note that the primary contributions of this paper are the topological analysis and classification of a) individual games, and b) collections of games. While the pure policy spaces are finite in the games considered here, full enumeration of the policies is practically infeasible. For example, even in games such as Tic-Tac-Toe, while the number of unique board configurations is reasonably ‘small’ ($9! = 362880$), the number of pure, behaviorally unique policies is enormous ($\approx 10^{567}$, see Czarnecki et al. (2020, Section J) for details). Moreover, for these large games, empirical payoff matrices constructed solely from uniformly random policies are unlikely to capture intricate and competitive behavioral policies that may be played by, e.g., humans or competent agents; this is primarily due to the strategy support of strong policies constituting a small fraction of the full set of pure strategies in these games (shown to be logarithmically-small in the strategy space by Lipton et al. (2003)). Thus, a principled scheme for sampling a relevant set of policies summarizing the strategic interactions possible within large games remains an important open problem.

The policy sampling approach we use is motivated with the above discussions and open question in mind, in that it samples a set of policies with varying skill levels, leading to a diverse set of potential interactions between them. Moreover, regarding the point on uniform policies raised in **R2.18**, note that these are actually captured as a subset of policies in our sampling scheme. Specifically, as detailed in “Supplementary Information” → “Methods: Additional details” → “Policy sampling scheme”, the set of policies sampled with tree search depth d set to 0 correspond to random policies. Overall, the notion of ‘representative’ targeted here essentially corresponds to ‘a set of policies of varying skill levels, which approximately capture variations of strategic interactions possible in a game’. We agree with **R2.18** that this should be clearly specified in the paper, and we have updated the text to do so. Per Reviewer 2’s suggestion, we now make it clear upfront (in the introduction) that the empirical game theoretic results are subject to the policies actually used to generate them. We also include a ‘Strengths & Limitations’ subsection in the Discussion section to more clearly highlight this.

Having said this, both reviewers raise a valid point regarding the need for sensitivity analysis of the experimental results with respect to the choice of sampled strategies. Based on this feedback, we have conducted a new suite of experiments focused on this topic to test the robustness of our strategy sampling approach. For each game we consider (with the exception of Rock-Paper-Scissors, which is a canonical game and only 3×3 in size), we randomly subsample 50% of the policies. All other policies are discarded,

(a) Distribution of games' first two principal components under 10 trials of policy subsampling, showing sensitivity to policy space changes.

Figure R1: Sensitivity to choice of empirical strategies, via subsampling of policy space. For each empirical game, we randomly subsample 50% of the policies (discarding the rest). We subsequently run the full analysis pipeline. 10 trials of subsampling are conducted per game (with the exception of Rock–Paper–Scissors, where we do not subsample strategies/policies due to it being a small canonical game). (a) indicates the quartiles of the first two principal components per game, over all trials. (b)–(e) shows the landscape of games for 4 example trials. Note that game colors are kept the same as the original landscape of games visual (Figure R7a) for easier comparison.

Figure R2: Iterations to solve procedurally-generated game structures of varying sizes.

thus yielding an empirical payoff matrix 4 times smaller in overall size. We subsequently run the full analysis pipeline over 10 trials of subsampling per game (see Figure R1 for results). Figure R1a indicates the quartiles of the first two principal components per game, summarized over all trials. Despite 50% of the policies being randomly discarded, the overall statistics are relatively robust across the different trials. There is some sensitivity of the principal components to the empirical policies sampled in the process, which on closer inspection seems to occur for highly cyclical games such as Blotto variations and the Disc game. We additionally recompute the ‘Landscape of Games’ figure, independently for several of the subsampling trials, with 4 examples shown in Figures R1b to R1e. Note that while there are some quantitative differences (in terms of relative positioning of these games in the projected space), the overall trends and clusters are quite robust to the subsampling (compared to Figure 1 of the main paper, which shows the landscape generated using all policies).

Overall, we are grateful to the reviewers for raising this point, and believe the revisions, including the robustness of strategy sampling experiment, should make these aspects now clearer for readers.

GR.3 – The usefulness of automatic generation of empirical games (Reviewers 1 & 2)

(Point R1.5): “It isn’t clear to me that the new games generated by the method are all that useful. They may be, it just isn’t obvious to me as the reader.”

(Point R2.5): “The generation of games is restricted to a particular class and it is not easy to extrapolate to a much larger landscape of games or tasks that we need to explore, cluster and understand for AI progress. I’m not sure that they generate really “new and interesting games”, as they claim. In which way are they new and interesting?”

(Point R2.12): Then again, the final paragraph of the intro says that the paper “culminates in a demonstration of how the topological structure over games can be used to tackle the interestingness question of the Problem Problem”. I don’t think this is the case. Also, I’m not sure that finding hybrids between two games in terms of the characteristics extracted in this paper is actually the key aspect for “curriculum learning”. Rather, it is about the reuse of partial strategies and knowledge as the agent goes and learns through the curriculum. The generation of games as done here does not really solve the problem of finding increasing levels of perceptual or cognitive difficulty (beyond the naive versions of this problem, by adding noise or modifying payoffs and rewards), or at least why this is better than easier ways of analyzing difficulty in games using classical theories such as Item Return Theory [7].

Author response – We thank the reviewers for the constructive feedback. We would like to first emphasize that the objective of this paper is not to target the solution of the task generation problem / Problem Problem / curriculum learning problem as such. Our primary objective is to characterize the topology of multiplayer games, which we do believe is useful for better understanding these games, and an important step *towards* subsequently generating new games variants.

The procedure outlined in the ‘Procedural Game Structure Generation’ section of our manuscript is one example of how one can generate a particular form of multiplayer games (i.e., normal-form games). We’d like to emphasize that the study of normal-form games continues to play a prominent role in the game theory and machine learning literature (Bloembergen et al., 2015, de Witt et al., 2019, Durugkar et al., 2020, Leibo et al., 2017, Li and Wellman, 2020, Spooner and Savani, 2020, Wright et al., 2019) and as such the procedural generation of normal-form games can play an important role in the research community. In the context of multiagent reinforcement learning, specifically, progress has been made in recent years in developing AI agents capable of notable performance in highly complex games (Baker et al., 2020, Berner et al., 2019, Silver et al., 2016, 2018). Despite this, understanding the *dynamics* of multiagent learning remains an active research topic, wherein normal-form games continue to provide new insights for the design and evaluation of state-of-the-art algorithms (e.g., see Barfuss et al. (2019), Bloembergen et al. (2015), Hennes et al. (2020), Hu et al. (2019)). Moreover, normal-form games themselves can be considered a simplification or alternative representations of more complex underlying games, capturing key principles such as cooperation (e.g. in coordination games), competition (e.g., in zero-sum games such as Kuhn Poker), and social equality (e.g. in social dilemmas (Leibo et al., 2017)). The normal-form games analyzed, however, are typically those canonical in the game theory literature, or are hand-crafted to illustrate that an algorithm can, e.g., overcome coordination problems, or learn an optimal solution (with respect to some solution concept), or achieve some other desired outcome.

We agree with Reviewer 2 on the point that raw ‘hybrids’ of games alone are unlikely to play a key role in establishing useful curricula. However, we note that discovery of core features or mechanisms in games has, indeed, been a key driver of generation of new games (see, e.g., Charity et al. (2020), Khalifa et al. (2019), Shaker et al. (2013)). As such, we can observe interesting trends even in the specific class of normal-form games that we focus on here.

To make this point clearer, we conduct new experiments that procedurally generate and subsequently train policies on these games via the double oracle algorithm. Specifically, we create a large class of generated games as follows. We first randomly sample 10 pairs of target games from the base games shown in the ‘landscape of games’ (Figure 1, of the main paper). We subsequently procedurally generate games of sizes 2×2 , 3×3 , 5×5 , 7×7 , and 9×9 targeting each sampled pair. This yields a total of 50 generated games of varying characteristics. We subsequently evaluate the complexity of solving these games via double oracle (of course, these can also be used for analysis of the dynamics of multiagent learning algorithms, as done in recent works such as Bloembergen et al. (2015), Hennes et al. (2020)). Figure R2 summarizes the results of this experiment, with each generated instance evaluated using 10 randomly-seeded trials of the double oracle algorithm.

We notice several trends in these generated games. First, the generated games targeting the three pairs (Blotto(10,3) \times Blotto(10,4)), (Blotto(10,4) \times Blotto(5,5)), and (Rock–Paper–Scissors \times Disc game) consistently require the largest number of iterations to solve, across all game sizes. This result is explained by the fact that each of these games targets a mixture over pairs of highly-cyclical underlying games; thus, each generated game likely has a large Nash support (despite the noise in the generation process), making them difficult to solve through iterated strategy discovery. Next, we observe a lower bound roughly established by generated games involving the target pairs (Elo game (noise=0.1) \times Go (board size=4)) and (Elo game (noise=0.5) \times Normal Bernoulli Game). Here, the noisy Elo game plays an important role, as its (roughly) transitive structure tends to reduce the number of ‘strong’ strategies (and Nash support size), thus making them easier to solve. Interestingly, for the largest generated game size (9×9), the pair (Transitive game \times Quoridor (board size=3)) requires the fewest iterations to solve. Here, the lack of noise in the ‘Transitive game’ dominates the structure of these larger generated games, in contrast to the earlier-mentioned noisy Elo games, making this game easier to solve. Finally, the remaining generated games form a cluster with an intermediate number of iterations needed to solve them. This latter class of games primarily targets a mixture of cyclical and transitive games (e.g., (Blotto(10,5) \times Elo game (noise=0.1))), linking to the notion of ‘interestingness’ from an AI perspective, as next discussed in **GR.4**.

Reviewer 2 raises a valid point regarding the extensibility of this approach to more general classes of multiplayer games. We agree that this is an important area for future investigation. Note that while the specific method employed here is not directly applicable to generation of general classes of games, the empirical game-theoretic lens actually offers a means of doing so. Specifically, one way to generate more complex underlying games would be to parameterize core mechanisms of such a games class (either

explicitly, or via mechanism discovery, e.g., using a technique such as that described by Charity et al. (2020)). Subsequently, one could train AI agents on a population of such games, constructing a corresponding empirical game, and using the response graph-based techniques used here to analyze the strategic space of such games (e.g., in a manner similar to Wang et al. (2019), though under our graph-theoretic lens). Moreover, we now make a connection to the Search-based Procedural Content Generation (SBPCG) literature in the ‘Procedural Game Structure Generation’ section, which offers an avenue of alternative investigations into game structure generation. Specifically, the variations of fitness measures and representations previously explored in that literature (Togelius et al., 2011) may be used in lieu of the specific search-based approach used here. Overall, generation of even the ‘simple’ classes of games considered here exemplifies how one might use such an analysis to highlight relationships between generated games and existing, well-studied ones, to better understand and expand the space of multiplayer games.

To highlight these points, we now make clear the limitations of this game generation scheme in the ‘Strengths & Limitations’ section of the Discussions, and additionally include the above experimental results and discussions in the Supplementary Information.

GR.4 – Providing a definition for the ‘interestingness’ of a game (Reviewers 2 & 3)

(Point R2.2): “The motivation seems to be supported by understanding how “interesting a game is for an artificial agent”. The concept of being “interesting” is diluted in the introduction and only recovered at the end. No definition or metric of interestingness is introduced in the paper, not even the most classical ones, at the purely cognitive level, such as those related to ability and difficulty (a problem is interesting for an agent if it is slightly around its ability, i.e., not too easy not too difficult). Even if we understand interestingness in other ways, for instance in terms of symmetries or cycles in the payoff matrix, it’s unclear whether the techniques in this paper help answer the interestingness question, either for educational purposes (training ML agents) or for the “progress” of AI.”

(Point R3.3): “what it means for a game to be ‘interesting’ in the first place, or more fundamentally, how to characterize the topological landscape of games.” This sentence views games as “interesting” from an AI perspective and a game-rule perspective; but games are primarily “interesting” for players/humans and have been investigated from several other perspectives beyond game rules. Interest in games, beyond its rules and environments - can be well attributed to its aesthetics, its story, its social impact, the opponent behaviours (e.g. see <https://core.ac.uk/download/pdf/132619524.pdf>) and so on. Please clarify and define interestingness from an AI lens here and state what are the primary factors you consider for defining AI-interestingness. In the next page, it is obvious that by “topographical landscape” you are referring to the landscape of game rules. It would be beneficial however if “interest” is defined properly here.

Author response – We agree with both reviewers that we should have been clearer regarding the notion of ‘interestingness’ considered in the paper.

We first note that the primary objective of this paper is to analyze the strategic interactions possible within games to topologically characterize them, thereby enabling measurement of similarities between games of varying sizes and characteristics. While we are motivated by the question of ‘what makes a game interesting?’, stemming from the ‘Problem Problem’ and closely-related works on task theory and procedural content generation, we primarily seek to establish tools that enable discovery of a topology/taxonomy of games (regardless of whether they are interesting or not). We have updated the introduction to clarify this.

Of course, such a taxonomy of games can be beneficial for downstream consideration of the ‘interestingness’ question. As pointed out by both reviewers, there exist numerous perspectives on what may make a game ‘interesting’, which varies as a function of the field of study or problem being solved. These include (overlapping) paradigms that consider interestingness from: a human-centric perspective (e.g., level of social interactivity, cognitive learning and problem solving, enjoyment, adrenaline, inherent challenge, aesthetics, story-telling in the game, etc.) (Deterding, 2015, Hao and Chuen-Tsai, 2011, Lazzaro, 2009, Prensky, 2001, Vygotsky, 1978); a curriculum learning perspective (e.g., games or tasks that provide enough learning signal to the human or artificial learner) (Baker et al., 2020); a procedural content generation or optimization perspective (in some instances with a focus on General Game Playing (Genesereth et al., 2005)) that use a variety of fitness measures to generate new game instances (e.g., either direct measures related to the game

structure, or indirect measures such as player win-rates) (Browne and Maire, 2010, Nielsen et al., 2015, Risi and Togelius, 2020, Togelius et al., 2011, Wang et al., 2019); and game-theoretic, multiplayer, or player-vs.-task notions (e.g., game balance, level of competition, social equality or welfare of the optimal game solution, etc.) (Byde, 2003, Hom and Marks, 2007, Yannakakis and Hallam, 2004). Overall, it is difficult (and, arguably, not very useful) to introduce a singular definition of ‘interestingness’ that covers all of the above perspectives.

Having said this, we agree that it would be useful to make more precise the notion of ‘interestingness’ that is most closely linked to the analysis we conduct in our work. As pointed out by Reviewer 3, the specific notion of ‘interestingness’ we focus on here is linked to AI training and evaluation. We now make explicit in the introduction that we consider ‘interestingness’ as defined by the recent work of Czarnecki et al. (2020), which investigates this question from an AI training perspective. Specifically, Czarnecki et al. (2020) defines the notion of ‘Games of Skill’, which are games that are interesting or engaging, in that sense, due to two core features:

1. A notion of progress for players (i.e., transitive improvement of strength);
2. The availability of diverse play styles that perform similarly well (i.e., intransitivity between play styles of similar strengths).

In the ‘Landscape of Games’ section of the revision, we now include a subsection highlighting the various definitions of ‘interestingness’ discussed above. We now also make clear the link between the clusters of games discovered by our taxonomization and the AI-centric notion of ‘interestingness’ that we focus on above. Finally, we note that our topological analysis can be used to identify smaller classes of games with interesting relationships, similar to prior works using hand-specified games taxonomies (please refer to **R1.2** for details). Overall, we thank the reviewers for the feedback, and believe these changes should make the notion of ‘interestingness’ targeted here significantly clearer.

GR.5 – Regarding the classes of games considered and title change (Reviewers 2 & 3)

(Point R2.3): “Also, the kind of games explored in this paper is somewhat limited. Most challenges in AI today are about perceptual, cognitive and generalization issues. Whether games are transitive or cyclic produce insights for a very narrow take about games (and only some of the games). Many other games and tasks that are not K-player are left out from this study, simply because the payoff matrix is different or captures very little about the task dynamics. For instance, a one-player video game benchmark (e.g., ALE) is represented by the scores produced on these games, and then the analysis of transitivity is different, only non-trivial when analyzing several instances of the same game. As there are several instances of the same pairing (player - game, or player - player), we should also analyze agent reliability. This is ignored altogether in this paper by choosing the mean of the number of times A beats B but not the variance, but it may go beyond winning or losing, but winning or succeeding in a game consistently and with a large margin.”

(Point R3.6): “I feel that the title is largely misleading as the proposed mapping/generation approach is limited to a particular set of games (zero-sum, multi-player adversarial games). The title should specify which type of games the approach can navigate (and potentially generate).”

Author response – As noted in the response to **GR.1**, the focus of this paper is on multiplayer games (as in those studied via game theory), as opposed to single-player games (as in environments that a sole agent/human interacts with). We agree with the reviewers’ remarks that there is potential for misunderstanding here, and per Reviewer 3’s suggestion have changed the manuscript title from “Navigating the Landscape of Games” to “Navigating the Landscape of Multiplayer Games to Probe the Drosophila of AI” to better reflect our specific focus on multiplayer games and AI. We have additionally made changes to the introduction of the paper to make the focus on multiplayer games much clearer.

Please note that in the context of multiplayer games (and associated learning problems that can be cast as games, such as the training of Generative Adversarial Networks), the study of transitivity / intransitivity of strategic interactions and learning dynamics remains a fundamental research topic used to drive many recent algorithmic advances, including Bailey et al. (2020), Boone and Piliouras (2019), Hennes et al. (2020),

Mertikopoulos et al. (2018), Palaiopoulos et al. (2017), Vlatakis-Gkaragkounis et al. (2019). We agree with Reviewer 2 that a class of agent vs. task games is interesting to analyze (e.g., in a similar manner as Balduzzi et al. (2018)); while this falls outside the scope of this paper, we have updated the Discussions section to highlight it as a potential avenue for future work. We also fully agree regarding the role that variance of outcomes should play when evaluating agents. We investigated this theoretically and empirically in some of our recent work focused on agent evaluation (Rowland et al., 2019), and note its integration into topological analysis of games as future work in the revision.

An important clarification, in response to **R3.6**, is that our graph-based game *analysis* approach (i.e., the spectral analysis and clustering technique) is not limited to zero-sum games. Specifically, one of the main motivations behind using the α -Rank response graph is its applicability to many-player, general-sum games; this is exemplified, for instance, in the analysis of the game ‘11-20’, which is not zero-sum. However, as the reviewer correctly points out, the procedural game *generation* approach used in later experiments is limited to zero-sum games. This constraint is due to our use of the multidimensional Elo parameterization for generating payoffs in our specific examples. However, any other general-sum payoff parameterization approaches (e.g., even direct generation of the payoff entries) can also be used to avoid the zero-sum constraint.

Overall, we agree that there is room for misinterpretation of these nuances in the paper, and have accordingly revised the Introduction and Discussions sections.

GR.6 – Linking to related fields and additional references to related work (Reviewers 2 & 3)

(Point R2.3): “The paper misses some important connections in the analysis of the “problem problem” (the task theory or game characterization problem using previous terminologies), the analysis of AI benchmarks and competitions, and the notion of interestingness (and associated concepts such as game difficulty, and its relation to agent ability). See below for some of these references.”

(Point R2.9): “In the introduction, it is clear that AI has mostly focused on solving tasks (what the authors awkwardly refer to as the “policy problem”), while the analysis of tasks and problems in AI has only been brought to mainstream attention more recently [1], in the context of the problem of AI evaluation. Here they seem to suggest that the critical question is to define interesting games for progress in AI, with the even more awkward term of “problem problem”. This has been referred to by different other less awkward terms in the past, such as the problem of AI needing a “task theory” [2]. This may have the perspective of finding challenging (but not extremely challenging) problems for AI to be solved independently, but also to find curricula or other sequences of tasks that help with the training (or even education) of AI agents [3].”

(Point R2.10): “The relevant literature about the analysis of tasks, games or problems for AI is either not covered or just swiftly mentioned at the bottom of page 2. This includes a very wide range of papers that are cited but not discussed (where the relation to this work is clear, with some of them also exploring the space of tasks) and even some papers from the authors that also develop ways of extracting some topological structure. While I agree that graph theory can give very insightful transformations and representations, it is just a possible way of analyzing the payoff matrices. Many other ways abound in game theory, and we would need to find a comparison or a clear explanation of why this one in particular is advantageous or parsimonious.”

(Point R2.11): “Even in the domain of games, and board games more specifically, there is a long tradition of analyzing the structure of games or creating new ones, as done in general game playing [4] for board games, and related approaches. In many of these cases, the “artificial agents” were not learning agents originally (but most are nowadays). There are also several studies of the landscape of games in competitions (e.g., [5]), and even if they are not K-games, they can use pairwise matrices between agents and perform an analysis of transitivity [6].”

(Point R3.2): “also known as the ‘Problem Problem’ [37] what you are referring to as “Problem Problem” (ref in 2019) is known as automated game design, or procedural content generation (which makes it to the title of a section later in the paper). While the “problem problem” (better put procedural content generation/procedural generation) exists for more than 30 years while a community within AI and Games had

dedicated more than a decade of research efforts on automating the game design process and making games more appealing for players, and more interesting for AI/humans to play - e.g. see the following indicative work: ...”

Author response – We thank the reviewers for pointing out the important connections to prior literature. Regarding terminology of the ‘Problem Problem’, we follow that used in the reinforcement learning community as discussed in recent work of Leibo et al. (2019) in the context of multiagent reinforcement learning. The term ‘Policy Problem’ directly stems from reinforcement learning as well, referring to the core object an agent aims to learn (i.e. a ‘Policy’ dictating the actions taken by agents in different states). However, we agree with the reviewers’ feedback, and have significantly revised the rhetoric of the introduction to make the connection between the ‘Problem Problem’ to the fields of task theory, procedural content generation, curriculum learning, and automated game design significantly clearer. We also make clear the rich history of these fields, which span over 30 years, and also provide a more detailed breakdown of the suggested papers in the ‘Related Works’ section of the Supplementary Information.

Additionally, we have updated the section ‘The Landscape of Games’ to include numerous references to prior works investigating the question of ‘interestingness’ of games (from various perspectives). Moreover, in response to the request for comparisons made in **R2.10**, we now provide such comparisons against the game-theory based methodology of Bruns (2015) in the Supplementary Information (in addition to the ablative studies and sensitivity analysis detailed in the below responses).

Overall, all of the suggested references (and many others) have been added to the revised paper, which we summarize below:

- “Artificial intelligence and games” (Yannakakis and Togelius, 2018)
- “Rogue” (Toy and Wichman, 1980)
- “Elite” (Braben and Bell, 1984)
- “No Man’s Sky” (Games, 2016)
- “Procedural content generation in games” (Shaker et al., 2016)
- “Search-based procedural content generation: A taxonomy and survey” (Togelius et al., 2011)
- “Towards automatic personalized content generation for platform games” (Shaker et al., 2010)
- “Platformer level design for player believability” (Camilleri et al., 2016)
- “Automatic design of balanced board games” (Hom and Marks, 2007)
- “Obstacle tower: A generalization challenge in vision, control, and planning” (Juliani et al., 2019)
- “Increasing generality in machine learning through procedural content generation” (Risi and Togelius, 2020). Note that this is the updated version of “Procedural content generation: from automatically generating game levels to increasing generality in machine learning” (Risi and Togelius, 2019) suggested by Reviewer 3.
- “Answer set programming for procedural content generation: A design space approach” (Smith and Mateas, 2011)
- “Towards automated game design” (Nelson and Mateas, 2007)
- “Multi-faceted evolution of simple arcade games” (Cook and Colton, 2011)
- “The ANGELINA videogame design system—Part I” (Cook et al., 2016)
- “An experiment in automatic game design” (Togelius and Schmidhuber, 2008)
- “Evolutionary game design” (Browne and Maire, 2010)

- ““Characteristics of generatable games”” (Togelius et al., 2014)
- “Evolving chess-like games using relative algorithm performance profiles” (Kowalski and Szykuła, 2016)
- “Rules and mechanics” (Nelson et al., 2016)
- “Paired open-ended trailblazer (POET): Endlessly generating increasingly complex and diverse learning environments and their solutions” (Wang et al., 2019)
- “Enhanced POET: Open-Ended Reinforcement Learning through Unbounded Invention of Learning Challenges and their Solutions” (Wang et al., 2020)
- “General game playing: Overview of the AAAI competition” (Genesereth et al., 2005)
- “General video game AI: A multitrack framework for evaluating agents, games, and content generation algorithms” (Perez-Liebana et al., 2019)
- “Towards generating arcade game rules with VGDL” (Nielsen et al., 2015)
- “A new AI evaluation cosmos: Ready to play the game?” (Hernández-Orallo et al., 2017)
- “Why artificial intelligence needs a task theory” (Thórisson et al., 2016)
- “Task analysis for teaching cumulative learners” (Bieger and Thórisson, 2018)
- “Interaction between learning and development” (Vygotsky, 1978)
- “The lens of intrinsic skill atoms: A method for gameful design” (Deterding, 2015)
- “Why we play: affect and the fun of games” (Lazzaro, 2009)
- “Fun, play and games: What makes games engaging” (Prensky, 2001)
- “Theory of fun for game design” (Koster, 2013)
- “Game reward systems: Gaming experiences and social meanings” (Hao and Chuen-Tsai, 2011)
- “Emergent tool use from multi-agent autotutorials” (Baker et al., 2020)
- “Increasing generality in machine learning through procedural content generation” (Risi and Togelius, 2020)
- “Applying evolutionary game theory to auction mechanism design” (Byde, 2003)
- “Evolving opponents for interesting interactive computer games” (Yannakakis and Hallam, 2004)
- “Analyzing the expressive range of a level generator” (Smith and Whitehead, 2010)
- “Mech-Elites: Illuminating the Mechanic Space of GVGAI” (Charity et al., 2020)
- “Automatic generation and analysis of physics-based puzzle games” (Shaker et al., 2013)

Reviewer 1

R1.1 – This paper seeks to classify the landscape of games by analyzing relations among strategies of the games using graph analysis. Based on the categorization, the paper also shows preliminary results for generating new games that could be interesting for training AI agents (establish a sort of curriculum).

The topic itself is interesting, novel, and important. By classifying games and understanding what makes them interesting, difficult, or strategically interesting, one can potentially identify how to train and evaluate AI algorithms. There have been attempts to classify simple games, but classifying complex games is difficult.

I certainly learned from reading this paper — I enjoyed reading it. It took me a little bit to figure out the authors’ strategy for classifying games, but once it clicked for me I did like it quite a bit. The authors have articulated a clever idea.

Author response – We thank the reviewer for the accurate summary, constructive suggestions, and positive remarks regarding the main idea of our work. Thanks to the reviewer’s feedback, we have made substantial improvements to the paper, which we hope also improves the clarity of the general idea and associated strengths & limitations. Below please find a point-by-point reply to all your comments and suggestions.

R1.2 – However, I’m hung up on a number of different points, each of which keeps me from enthusiastically endorsing the paper in its current form:

For the smaller canonical games, it seems like there needs to be greater validation for why describing the landscape of games in this way is useful. Perhaps the authors could compare the resulting similarity of games from their algorithms with those of another game-classification scheme. The work of Bruns (2015) (“Names for Games: Locating 2x2 Games”), which builds on the prior work of others, could be one possibility, albeit it only works for 2x2 games. Still, does the proposed classification scheme produce similar results to Bruns’ classification scheme in those simple games? If not, why not? I guess the bottom line is that I’m not certain about the extent of the usefulness of the “classification scheme”.

Author response – We appreciate the reviewer’s suggestion for running validation studies. While we originally omitted this due to the lack of comparable methods that scale to the larger games we consider here, we find the suggestion of comparing against existing smaller-scale baselines a valid point. As a result, we have conducted a new set of experiments comparing our approach against that of Bruns (2015) (focusing on the ordinal games results presented in Figure 1 of their paper), per the reviewer’s suggestion.

Let us first provide a summary of the approach of Bruns (2015) (also, please see our revised supplementary material), prior to comparing our method against it. The taxonomy of 2×2 games proposed by Bruns (2015) relies on two elements: 1) a topological ordering of said games, according to the patterns of payoffs received by each player; 2) classification of payoff families, based on the respective payoffs received by each player at the Nash equilibria. Based on the earlier work of Robinson and Goforth (2005), Bruns (2015) identifies 12 core payoff patterns in ordinal 2×2 games (which we summarize for the row player in Figure R3a). Column player payoffs are defined in Bruns (2015) by transposing the row player payoffs along the *anti*-diagonal (i.e., not the conventional transpose). The combinations of row and column player payoffs yields a total of 144 2×2 games of interest. These games are subsequently categorized according to the relative payoffs received by the players in the Nash equilibria (e.g., win-win, cyclic, unfair outcomes, etc.), yielding a total of 7 classes (reproduced here in Figure R3b). Note that the 4 quadrants indicated by the thick blue lines in Figure R3b indicate so-called ‘layers’ of 36 games each, where games in each layer have the same highest-payoff relationship between the two players. For example, games in the lower-left layer are all win-win (thus have the highest payoff for both players in the same payoff cell), whereas games in the top-right layer have the highest payoffs in diagonally-opposite payoff cells for the players, and so on.

To generate a clustering of these games using our approach, we run the graph-based analysis detailed in the paper (i.e., computing response graphs, collecting graph measures, and running PCA on the collection of 144 games). We subsequently cluster these games using their top-2 principal components, using 7 clusters (as in Bruns (2015)). The resulting clusters are visualized in Figure R3c. Note that as our clusters are automatically discovered, and not hand-specified using Nash equilibrium payoffs, only their structure (and not color) should be compared against Bruns’ clusters. While some similarities exist between these and Bruns’ clusters (e.g., a visually-apparent division into the same 4 layers is present in our clustering, and

Chicken (Ch)	Battle (Ba)	Hero (Hr)	Compromise (Cm)	Deadlock (DI)	Prisoner's dilemma (Pd)	Stag hunt (Sh)	Assurance (As)	Coordination (Co)	Peace (Pc)	Harmony (Ha)	Concord (Nc)
$\begin{bmatrix} 2 & 3 \\ 1 & 4 \end{bmatrix}$	$\begin{bmatrix} 3 & 2 \\ 1 & 4 \end{bmatrix}$	$\begin{bmatrix} 3 & 1 \\ 2 & 4 \end{bmatrix}$	$\begin{bmatrix} 2 & 1 \\ 3 & 4 \end{bmatrix}$	$\begin{bmatrix} 1 & 2 \\ 3 & 4 \end{bmatrix}$	$\begin{bmatrix} 1 & 3 \\ 2 & 4 \end{bmatrix}$	$\begin{bmatrix} 1 & 4 \\ 2 & 3 \end{bmatrix}$	$\begin{bmatrix} 1 & 4 \\ 3 & 2 \end{bmatrix}$	$\begin{bmatrix} 2 & 4 \\ 3 & 1 \end{bmatrix}$	$\begin{bmatrix} 3 & 4 \\ 2 & 1 \end{bmatrix}$	$\begin{bmatrix} 3 & 4 \\ 1 & 2 \end{bmatrix}$	$\begin{bmatrix} 2 & 4 \\ 1 & 3 \end{bmatrix}$

(a) 2×2 games and row player payoffs from Bruns (2015). Important note: corresponding column player payoffs are defined as a transpose along the *anti*-diagonal in Bruns (2015) (i.e., not according to the typical convention of using standard transposes for symmetric games). This latter point has no impact on quantitative results, but is important for readers qualitatively analyzing the payoff structures and corresponding results below.

(b) Clusters (Bruns).

(c) Clusters (ours, high α -Rank α , where $\alpha = 0.2$).

(d) Clusters (ours, lower α -Rank α , where $\alpha = 0.01$).

(e) Game distances (ours, to $Co \times Co$).

(f) Game distances (ours, to $Pd \times Pd$).

(g) Game distances (ours, to $Ch \times Ch$).

Figure R3: Comparison of our approach to that of Bruns (2015) for 2×2 games.

Figure R4: Dendrogram of 2×2 games, computed using our approach. All 144 games are visualized here, with classes colored in accordance to Figure R3d.

certain games clusters such as $(Ch \times Co, Ch \times As, Ba \times Co, Ba \times Sh, Hr \times As, Hr \times Sh)$ are shared between the two methods), notable differences are present.

To better understand these differences, consider again the means by which the Bruns clusters are derived: manual identification of patterns over the players’ payoffs in the Nash equilibria of each game. Our approach, by contrast, relies on aggregation of statistics over the response graph of the games. As 2×2 games consist of 4 strategy profiles in total, the response graphs correspondingly have 4 nodes, implying that few variations are possible in their structure (in contrast to the much larger games targeted in our main results). Despite this, we can better tune our approach for these extremely small games. Specifically, one of the statistics used for computing our clustering is the α -Rank distribution entropy associated with the response graph. Typically, α -Rank is used with a high value of selection-intensity parameter, α , to increase the distribution mass over strong strategies. In these small 2×2 games, decreasing the value of α -Rank’s selection-intensity parameter, α , causes the α -Rank distribution to increase in entropy, and subsequently better capture the distribution of payoffs received over the profiles (akin to the Nash-based payoff comparisons used by Bruns to derive their clusters). The result of this modification, which brings our approach closer to that of Bruns, is visualized in Figure R3d. Notable similarities between our clusters and Bruns’ are evident in this updated approach. Namely, variations of prisoner’s dilemma (Pd) are clustered in our approach and theirs: $(Pd \times \{Co, As, Sh, Pd\})$ (and their transposed analogs). The cyclic game cluster of $\{Ch, Ba, Hr\} \times \{Co, As, Sh\}$ (and their transposed analogs) are identically clustered in the approaches.

Note also that due to the Nash-payoff based classification scheme of Bruns, variations in the lower-left quadrant (where all Nash equilibria are win-win) are not evident in their approach (Figure R3b). By contrast, our approach highlights differences between these games. For example, whereas Bruns classifies the symmetric Coordination Game $(Co \times Co)$ in the same class as Stag Hunt $(Sh \times Sh)$, ours places it in its own class altogether (better matching intuition, due to the anti-coordination outcomes of Stag Hunt not being present in the Coordination Game).

With these comparisons in place, we further note several benefits of our approach compared to that of Bruns (2015) and related works on classification of 2×2 games. First, our approach does not rely on a hand-crafted taxonomy or classification of games; our games classes are identified automatically, and without need for human expertise in determination of patterns in payoffs. Second, as the taxonomy introduced by Bruns’ relies on enumeration of payoff orderings, it faces significant scalability issues if to be extended to larger games (involving more strategies and/or players). By contrast, our approach applies directly to all normal-form games, and we are not aware of similar automated classification schemes for larger, many-player games. Moreover, whereas Bruns’ approach provides a ‘hard’ classification based on hard-coded structural patterns in payoffs, ours provides a ‘soft’ classification based on a similarity metric (specifically, based on the games’ principal components). Thus, our approach can be used to compute *distances* between games, in contrast to Bruns’ and related approaches. We visualize several examples of such game distances in Figures R3e to R3g. A subsequent benefit of being able to compute distances between games is that it enables computation of a dendrogram over a collection of games. We use our approach to visualize this for the considered 2×2 games in Figure R4. Finally, the downstream usefulness of our approach is that such a similarity metric can be optimized to generate new games (as illustrated in the main text), in contrast to the ‘hard’ classification scheme of Bruns.

Overall, we are grateful to the reviewer for making the suggestion for these additional comparisons. We believe these findings to be interesting and likely relevant for readers, while shedding light on major benefits of our proposed approach. As a result, we have included these comparisons in the supplementary materials of the paper.

R1.3 – For the larger games, I’m really hung up by the fact that you get different results when you use different strategies. The authors state that this attribute means that they have addressed the “Policy Problem” rather than the “Problem Problem.” To me, that indicates that the paper doesn’t achieve what it set out to achieve (which I thought was classifying games). Is there a way to consider strategies that will provide invariance to the classification scheme. If not, I see that as a severe limitation.

Author response – Please refer to our response to **GR.2**.

R1.4 – The results in classifying games seems biased toward a single issue, which appears to be described in the idea that games are “solved” by finding a Nash equilibrium. Is that the only thing that matters in

classifying the “landscape of games.” Are the kinds of games the authors are looking at shining too much focus on that idea? For example, does the technique described indicate anything about what is required for cooperation (or the possibility for cooperation) to happen? This is just one example I can think of off the top of my head (there are others). The question is whether or not this analysis really captures the defining characteristic of games? My inclination is that the identified technique could be one way to classify games, but it isn’t yet clear what the strengths and weaknesses of it are.

Author response – Please refer to our response to **GR.1**.

R1.5 – It isn’t clear to me that the new games generated by the method are all that useful. They may be, it just isn’t obvious to me as the reader.

Author response – Please refer to our response to **GR.3**.

R1.6 – Other minor comments: Page 1: primarily “seeked” → primarily sought.

Author response – Thanks, the sentence has been changed altogether in the revision.

R1.7 – “While the importance of games as a natural research platform for development of learning algorithms and the measurement of progress in AI is well-established, . . .”

Doesn’t seem well-established to me. It’s certainly used a lot. But some would argue that they have been a detriment to AI development and evaluation. Chollet certainly argues very convincingly against it (see “On the measure of intelligence”).

Author response – We thank the reviewer for highlighting the competing viewpoint here. Our original intent was to make clear the prominent role that multiplayer games have played throughout the history of AI, specific within the reinforcement learning and machine learning literature. However, we agree that rewording (to avoid the statement being interpreted as a unanimous viewpoint by AI researchers) is a good idea, and have substantially revised the introduction of the revision to change the rhetoric.

R1.8 – Page 2: Figure 1: When I first read the paper, I had a hard time understanding this figure and what it was saying. After reading the paper, I think I get it, though it seems like quite the simplification (that could be good or bad). Also, the term “real-world games” applied to things like Connect-Four and Tic-Tac-Toe seems strange to me.

Author response – Apologies for the initial confusion. We placed the figure rather early in the text as it provides a useful visual overview of the results, though understand why that can be potentially confusing without additional context. As such, we have updated the figure caption to provide context in terms of how it was generated (via principal component analysis of games’ response graph features), and how one can interpret it.

Regarding the ‘real-world games’ terminology, we borrow this short-hand from Czarnecki et al. (2020) to convey games that are played (typically for pleasure) by humans in the ‘real world’. As not all of these games are of the same class (e.g., ‘board games’ or ‘paper and pencil games’), we use ‘real-world games’ to collectively refer to them. We now provide an explanation of what we mean by this term both in Figure 1’s caption, and in the main text.

R1.9 – “Hierarchy of complexity” — what does that mean? What makes something complex, in general? Given the large amount of brilliant researchers that have tried and failed to define it (unless the authors are referring to computational complexity), my instant thought was that this problem is not likely to be easily solved.

Author response – Good point. We originally intended this phrase to convey that the notions of ‘complexity’ and ‘interestingness’ in multiplayer games are quite nuanced, and that competing/overlapping definitions exist across many fields (i.e., can be taxonomized into a hierarchy of definitions themselves). We agree that providing a unifying definition is not easy. Per the response to **GR.4**, we now introduce a clearer discussion

on the topic of ‘interestingness’ in the ‘Landscape of Games’ section, highlighting this issue, and providing explicit links to several related definitions/investigations in the literature. As such, we have removed this phrase altogether from the introduction.

R1.10 – Page 3: Early on, I wasn’t sure how to interpret the authors’ use of the word “intransitive,” until I understood that they were talking about the evaluation of strategies. I probably should have caught on earlier, but I didn’t.

Author response – We apologize for the initially-confusing wording. We have now added the following clarification where ‘intransitive game’ appears for the first time: “here, a ‘transitive’ game is defined as one where strategies can be ordered in terms of strength, whereas an ‘intransitive’ game may involve cyclical relationships between strategies (e.g., Rock–Paper–Scissors).”

R1.11 – Last paragraph of the introduction: Is neural network + a good curriculum all there is to intelligence? Seems like a vast simplification.

Author response – We agree with the reviewer that this would be a great oversimplification (please note that we did not mention ‘neural networks’ in the noted passage or surrounding paragraphs). Overall, we have substantially revised the introduction to change the rhetoric and provide a broader perspective over the various fields that have tried to tackle the problem of game understanding and content generation.

R1.12 – Page 4: As I read the paper, I kept getting tripped up by what strategies the authors were referring to. I got it after reading further, but it was confusing to me until I got it several several pages after.

Author response – We apologize for the confusion. We have now updated the caption of Figure 2 (and text surrounding) to make it clearer that the strategies correspond to the rows and columns of the payoff tables.

R1.13 – Equating “solving a game” to “finding a Nash equilibrium” gives me pause. What if the game has many Nash Equilibria? What if other players don’t play the equilibria I’m considering?

Author response – Please refer to our response to **GR.1**.

R1.14 – Page 8: Figure 7: I can’t see the arrows on the “contracted graphs”. Same goes for some other graphs.

Author response – Thanks for the suggestion, we have increased the arrow sizes on all contracted graphs. Please note that arrows for some graphs (e.g., spectral graphs for larger games involving many nodes) are left as-is, as enlarging them makes the nodes themselves difficult to discern.

R1.15 – Page 11: I think the complexity results are interesting, albeit perhaps intuitive. I’m not sure about the normalization though. Seems like that could bias the results. I think I don’t sufficiently understand the implications of the normalization.

Author response – We thank the reviewer for the kind remarks. Regarding the effects of normalization, we have now conducted ablative analysis of the effects on both the complexity results, and the overall landscape of games. Overall, the main motivation behind the normalization used was to make games of different sizes directly comparable. Please refer to **R2.14** for experiments comparing the effects of normalization, and additional justification.

Once again, we would like to thank the Reviewer the time spent in reading the manuscript, for their kind comments and detailed suggestions, and for sharing our enthusiasm. Thank you very much!

Reviewer 2

R2.1 – This paper presents a new pipeline for analyzing general-sum games, starting from their payoff matrices, as obtained from the set of strategies for canonical games or a sample of policies (AI agents) for other richer (“empirical”) games. The technique is illustrated with a few examples of canonical and empirical games.

If AI systems are evaluated by the problems they can and cannot solve we have to understand these problems better. K-player games represent a particular kind of problem that has received enormous attention in AI, reinforced by a strong connection with game theory. Then, it is very important to produce novel and insightful ways of analyzing the so-called landscape of games. This has been vindicated in the past as well. However, this paper falls short in this aspiration, and even the expectations set in the abstract and the introduction.

Author response – We thank the reviewer for the constructive feedback, as well as positive remarks regarding the importance of the problem being tackled here (the generation of a landscape of games). Based on your and the other reviewers’ feedback, we have substantially updated the paper to:

- Clarify our objective and contribution upfront (in the Introduction section).
- Reference and discuss missing related works to existing literature and closely-related fields (per **GR.6**).
- Clarify misunderstandings stemming from conflation of terms such as ‘games’ with ‘multiplayer games’ (the latter being the focus here), and specify clearly in the paper the notion of ‘interesting’ games we consider here.
- Include ablative studies (**R2.14**), sensitivity analysis (**GR.2**), and comparison to related works taxonomizing small normal-form games (**R1.2**)
- Clarify the strengths and limitations of the approach (in the Discussion section)

Overall, we believe these changes significantly improve the clarity of the paper, more clearly substantiate the novelty of the approach, and provide better scoping of results and positioning with respect to the prior literature. Thank you again for your helpful comments and suggestions.

R2.2 – The motivation seems to be supported by understanding how “interesting a game is for an artificial agent”. The concept of being “interesting” is diluted in the introduction and only recovered at the end. No definition or metric of interestingness is introduced in the paper, not even the most classical ones, at the purely cognitive level, such as those related to ability and difficulty (a problem is interesting for an agent if it is slightly around its ability, i.e., not too easy not too difficult). Even if we understand interestingness in other ways, for instance in terms of symmetries or cycles in the payoff matrix, it’s unclear whether the techniques in this paper help answer the interestingness question, either for educational purposes (training ML agents) or for the “progress” of AI.

Author response – Please refer to our response to **GR.4**.

R2.3 – Also, the kind of games explored in this paper is somewhat limited. Most challenges in AI today are about perceptual, cognitive and generalization issues. Whether games are transitive or cyclic produce insights for a very narrow take about games (and only some of the games). Many other games and tasks that are not K-player are left out from this study, simply because the payoff matrix is different or captures very little about the task dynamics. For instance, a one-player video game benchmark (e.g., ALE) is represented by the scores produced on these games, and then the analysis of transitivity is different, only non-trivial when analyzing several instances of the same game. As there are several instances of the same pairing (player - game, or player - player), we should also analyze agent reliability. This is ignored altogether in this paper by choosing the mean of the number of times A beats B but not the variance, but it may go beyond winning or losing, but winning or succeeding in a game consistently and with a large margin.

Author response – Please refer to our response to **GR.5**.

R2.4 – Finally, the analysis is made for the set of basic (non-deterministic) strategies in simple games or experimental strategies (policies as a population of AI agents trained with different algorithms and parameters). This gap is important, as in the second case we derive all the metrics and plots conditioned to the choice of the population of artificial agents. A different choice of agents may lead to a different characterization of the game. This is a very important point, and the paper should analyze how robust the analysis is when the population of agents is derived in a different way. For instance, there are papers analyzing AI (games) competitions, and the results about how games “behave” are subject to the participants of the competition. This subjectivity or relativity is unavoidable, so it is very important to clarify whether the topologies hold if one just changes the population by creating stochastic combinations of existing agents. This not only applies to “experimental” games, but also canonical games. For instance, rock paper scissors, or its simplified version of matching pennies, is cyclic, but it’s also peculiar because a fully random agent (choosing between the two or three strategies randomly) cannot be beaten on expectation. Between any two agents one can construct any point in between. The key issue is that one can define infinitely many distributions of stochastic policies from a finite set of strategies and derive the whole pipeline presented in this paper using this distribution. The authors present some guidelines (and algorithm) to generate a ‘representative’ population in experimental games, but this seems to be devised to get what they want to get rather than a population that is truly representative of some canonical reference that would be achieved in large numbers. Actually, in the landscape of agents (which is dual to the landscape of games) one has to determine a way of considering some populations more meaningful than others, and this may be related to the resources or complexities of the policies.

Author response – The reviewer raises a good point regarding sensitivity analysis with respect to sampled policies, which we have now conducted in **GR.2**. We respectfully disagree with the notion that the representative policy sampling is devised to ‘get what we want’, and detail the challenges associated in appropriately sampling such policies in **GR.2**.

The reviewer raises an interesting point regarding the effect that stochastic mixtures of these strategies could have on the results. To a limited extent, the effect of mixing between strategies is captured in the approach for the strongest strategies (i.e., those whose mixtures yield the highest payoffs); specifically, one of the measures used in our analysis is the entropy of the α -Rank distribution over the strategies (which captures, roughly speaking, the mixture over the evolutionarily ‘strongest’ strategies in each game). For example, in Rock–Paper–Scissors, the α -Rank distribution is uniform over the three pure strategies, and this information is used in the method for generating the landscape of games. For the larger empirical games that rely on the strategy sampling scheme detailed in the paper, the sampling scheme used is designed to capture strategies of varying strengths, thus capturing stochasticity of behaviors through the sweeps of the tree depth and random seed parameters in the sampling scheme.

Having said this, we agree that it is interesting to consider the effect that additional mixing of strategies would have on the results, and have now conducted a suite of experiments focused on this. Specifically, for all games in the landscape, we expand each payoff table by adding mixtures of strategies. For each game, we uniformly sample a mixture over a random selection of half of the base strategies; specifically, we sample the mixture coefficients for these base strategies from a $\text{Dir}(1, \dots, 1)$ distribution, with mixture coefficients set to 0 for all other strategies. We repeat this process 100 times, expanding the payoff table accordingly (i.e., with 100 additional mixed strategies). For Rock–Paper–Scissors (due to the small size of the strategy space), we sample these 100 additional mixed policies uniformly over the full support of all 3 base strategies. Figure R5 shows the results for 4 independent trials. Some observations can be made here, in comparing the trials to one another (and to the original landscape, visualized in this rebuttal in Figure R7a). At a high level, most of the prominent clusters found in the original landscape are also present here, with some specific clusters highlighted as follows: 1) Rock–Paper–Scissors and the Disc game (closely clustered in both the original and new landscapes, despite the size of Rock–Paper–Scissors increasing from 3×3 in the original landscape to 103×103 in the mixed strategy landscapes) 2) Elo game (noise=0.0) and Transitive game; 3) Elo game (noise=0.1) and the Normal Bernoulli game; 4) Random Game of Skill, Elo game (noise=0.5), and the AlphaStar League; 5) Real-world games (e.g., Connect Four and Quoridor (board size=4)); 6) The cluster of ‘Blotto’ games have somewhat shifted towards that of Rock–Paper–Scissors and the Disc Game. Notably,

Figure R5: Sensitivity to random mixtures of policies, with 100 additional policies included per game. Note that game colors are kept the same as the original landscape of games visual (Figure R7a) for easier comparison.

all of these games are highly cyclical, and Blotto requires players to play uniformly across all permutations of the token-selection strategies (due to the game rules itself being permutation-invariant), a characteristic shared by Rock-Paper-Scissors.

Overall, we appreciate the reviewer’s suggestion for investigating the effects of mixtures of strategies on the results, and have appended these and related discussions to the Supplementary Information.

R2.5 – The generation of games is restricted to a particular class and it is not easy to extrapolate to a much larger landscape of games or tasks that we need to explore, cluster and understand for AI progress. I’m not sure that they generate really “new and interesting games”, as they claim. In which way are they new and interesting?

Author response – Please refer to our response to **GR.3**.

R2.6 – About the particular methodology, it looks unnecessary complicated at places, with many choices about each step, with the overall feeling that all these steps, transformation algorithms, normalizations and parameters are chosen to get what the authors want to obtain, a picture such as the one in Fig. 1., with no ablation procedure or what could have been different. The authors talk about “findings” but I don’t see any surprising thing in Fig. 1.

Author response – We appreciate the reviewer’s feedback, though respectfully disagree that the method is unnecessarily complex. While there are several steps involved here, these are necessary due to the transformation of a game to a corresponding directed graph, which we can subsequently analyze. Note that we

now also conduct a comparative evaluation of the transformation of graph complexity measures in **R2.20**. Moreover, the experiments related to Nash equilibria and Double Oracle (which may be some of the choices highlighted by the reviewer here) are not factored into the game-specific spectral graphs, nor Figure 1 (see **GR.1** for details). As indicated by these experiments (and those in **GR.2**), these transformations yield meaningful and robust similarity measures over the games (e.g., in Figure 1). To more clearly substantiate our proposed approach, we now go further in highlighting similarities and differences to works on ‘interesting’ games in the ‘Landscape of Games’ section, along with a comparison to the prior work of (Bruns, 2015) in taxonomizing smaller games in the Supplementary Materials (where we discuss several notable differences and strengths of our method).

R2.7 – The paper misses some important connections in the analysis of the “problem problem” (the task theory or game characterization problem using previous terminologies), the analysis of AI benchmarks and competitions, and the notion of interestingness (and associated concepts such as game difficulty, and its relation to agent ability). See below for some of these references.

Author response – Please refer to our response to **GR.6** for related works and discussions. We thank the reviewer for pointing out these additional works, which are all now added to the paper and/or Related Works section of the Supplementary Materials, where applicable.

R2.8 – Because of all this, I think the paper doesn’t portray a sufficiently novel, robust, general and insightful procedure for analyzing the landscape of games, fails to connect with AI more generally and doesn’t make a convincing case that this would be more useful for its progress than existing, perhaps simpler, approaches.

Author response – We appreciate the reviewer’s detailed feedback and constructive suggestions. We have made significant changes to the paper to highlight:

- Novelty, insightfulness, and positioning with respect to closely-related work (see **R1.2** and updated related works in **GR.6**).
- Robustness (by running additional ablative studies and sensitivity analysis, see **R2.14** and **GR.2**)
- Generality (see clarifications regarding the paper scope in **GR.5**, and generality of the analysis techniques in contrast to the state-of-the-art in terms of games classification and topological analysis in **R1.2**).

While the topological analysis of games has a rich history, we note that the prior works, albeit simpler, are limited to a significantly more restrictive class of multiplayer games. As part of the revision, we have conducted a new suite of experiments comparing our approach against that of recent work on classification of 2×2 games, where we highlight significant advantages compared to these works (please refer to **R1.2** for details).

R2.9 – ***** DETAILED COMMENTS *****

In the introduction, it is clear that AI has mostly focused on solving tasks (what the authors awkwardly refer to as the “policy problem”), while the analysis of tasks and problems in AI has only been brought to mainstream attention more recently [1], in the context of the problem of AI evaluation. Here they seem to suggest that the critical question is to define interesting games for progress in AI, with the even more awkward term of “problem problem”. This has been referred to by different other less awkward terms in the past, such as the problem of AI needing a “task theory” [2]. This may have the perspective of finding challenging (but not extremely challenging) problems for AI to be solved independently, but also to find curricula or other sequences of tasks that help with the training (or even education) of AI agents [3]. This is no different from psychological measurement and the choice of tasks: how they can be analyzed, arranged, created and chosen, and most especially, how latent factors of these tasks, extracted from the behavior of populations of humans, lead to hierarchies of tasks and clusters thereof using the extracted latent features. In the end, what the authors do in this paper is extracting some latent features that they plot using the top two principal components after PCA.

Author response – Please refer to our response to GR.6.

R2.10 – The relevant literature about the analysis of tasks, games or problems for AI is either not covered or just swiftly mentioned at the bottom of page 2. This includes a very wide range of papers that are cited but not discussed (where the relation to this work is clear, with some of them also exploring the space of tasks) and even some papers from the authors that also develop ways of extracting some topological structure. While I agree that graph theory can give very insightful transformations and representations, it is just a possible way of analyzing the payoff matrices. Many other ways abound in game theory, and we would need to find a comparison or a clear explanation of why this one in particular is advantageous or parsimonious.

Author response – Please refer to our response to GR.6.

R2.11 – Even in the domain of games, and board games more specifically, there is a long tradition of analyzing the structure of games or creating new ones, as done in general game playing [4] for board games, and related approaches. In many of these cases, the “artificial agents” were not learning agents originally (but most are nowadays). There are also several studies of the landscape of games in competitions (e.g., [5]), and even if they are not K-games, they can use pairwise matrices between agents and perform an analysis of transitivity [6]).

Author response – Please refer to our response to GR.6.

R2.12 – Then again, the final paragraph of the intro says that the paper “culminates in a demonstration of how the topological structure over games can be used to tackle the interestingness question of the Problem Problem”. I don’t think this is the case. Also, I’m not sure that finding hybrids between two games in terms of the characteristics extracted in this paper is actually the key aspect for “curriculum learning”. Rather, it is about the reuse of partial strategies and knowledge as the agent goes and learns through the curriculum. The generation of games as done here does not really solve the problem of finding increasing levels of perceptual or cognitive difficulty (beyond the naive versions of this problem, by adding noise or modifying payoffs and rewards), or at least why this is better than easier ways of analyzing difficulty in games using classical theories such as Item Return Theory [7].

Author response – Please refer to our response to GR.3.

R2.13 – The procedure concatenates a series of transformations to cluster the transitions that emerge from the payoff matrix. There are many choices there, such as the Double Oracle algorithm. For instance, this algorithm is based on planning, but is it really a good proxy for how hard it is to find the optimal strategy? And is this really meaningful? “The complexity of solving” these games happens at a very game-theoretical level, but this may not correspond to the complexity of solving the games that several AI techniques find with these games. My point is precisely that there are many ways to measure this complexity, if we go beyond scores. In AI agents using machine learning, this can be related to the number of iterations, complexity of the trained networks, and many other metrics.

Author response – Please refer to our response to GR.1.

R2.14 – A similar thing happens with the spectral analysis. How would the result of many figures be affected by other techniques and transformations? And what would the effect be for Figure 1? Overall, we would need more justification of these choices as proxies of complexity or ways for summarizing/clustering the graphs. I’m not an expert in graph theory, but I can imagine that there are plenty of options, from the analysis of social networks to bibliometric analysis. They pick techniques and transformations from these areas, but the justification of the choices is missing, especially what the impact would have been if another choice had been made.

Author response – We thank the reviewer for the feedback. Overall, while there are indeed various means of analyzing directed graphs in the literature, the approach we use for the spectral analysis of graphs relies on

(a) Normalized iterations (via payoff sizes): True, Normalized graph measures: True

(b) Normalized iterations (via payoff sizes): False, Normalized graph measures: True

(c) Normalized iterations (via payoff sizes): True, Normalized graph measures: False

(d) Normalized iterations (via payoff sizes): False, Normalized graph measures: False

Figure R6: Ablative studies of response graph complexity vs. computational complexity of solving associated games. Each row plots a respective measure of graph complexity against the number of iterations needed to solve the associated game via the Double Oracle algorithm. Each row visualizes a different combination of normalization applied to the double oracle iterations and/or graph complexity measures.

(a) Normalized graph measures.

(b) Unnormalized graph measures.

Figure R7: Comparison of landscape of games generated via normalized and unnormalized response graph measures. Note that game colors are kept consistent between the two plots for easier comparison.

fairly standard tools. The results intuited in the case studies of smaller games serves as empirical justification of these techniques, and as detailed earlier, we have conducted a sensitivity analysis of the spectral analysis technique (with respect to the policy sampling scheme used) as further justification of its robustness. Here, we go further and conduct additional ablative analysis of the transformations of graph complexity measures. In Figure R6, we plot all complexity results, over all combinations of normalizing/not normalizing the graph measures (y-axes) and/or the number of ‘iterations to solve’ (x-axes). Note that the case corresponding to the main paper is captured by Figure R6a, where both axes are normalized. The primary motivation for conducting this normalization (across both axes) is that the measures considered here would otherwise vary with increasing game size, which would make it easy to artificially inflate a game’s complexity via arbitrary increase of the strategy space size via ‘filler’ strategies (see **R2.20** for details). Moreover, in Figure R6, we find that normalizing both types of measures consistently maximizes their Spearman correlation (with low p-value, see top-right of each subfigure).

We also consider now the impact that this normalization has on the ‘landscape of games’ figure (see Figure R7). Here, note that there are only 2 ablations involving only normalization over the graph measures (as double oracle/iterations to solve are not used in the spectral analysis). The normalized landscape (Figure R7a), used in the main paper, reveals coherent clusters of related games. By contrast, the variety of underlying payoff table sizes and associated graph measures implies that the unnormalized landscape (Figure R7b) is notably less structured. For instance, while games such as Rock–Paper–Scissors and the Disc Game are conceptually very similar, their distance is relatively larger in the unnormalized landscape due to their widely different payoff sizes (3×3 vs. 1000×1000 , respectively). Overall, we appreciate the reviewer pointing out the need for additional studies justifying the choices made, which have been added to the revision.

R2.15 – Figure 3 could include the colors in the payoff matrix as all the other figures. Also, these colors could be better explained the first time.

Author response – Thanks for the suggestion. We now include the node cluster colors for Figure 3, and expanded the explanation of these colorings. We have also added colorbars to all payoff figures to better indicate the payoff scales.

R2.16 – For the analysis of the canonical games, how does it affect if we include stochastic strategies? Would all the clustering be broken because we can always find a strategy that is anywhere between two strategies? This is partially seen for the empirical games, but there is a particular procedure for choosing the populations of agents (policies or strategies), explained on page 10 (2nd paragraph). It seems that the method works well with redundant agents, but what about continuums of agents between two existing agents? Actually, this is something that could be exploited rather than avoided by the agent sampling algorithm.

Author response – We refer the reviewer to the related point **R2.4**, wherein we conduct additional experiments related to inclusion of mixtures of base strategies.

R2.17 – What I mean is that the characterization of a game should not depend on the choice of the population of strategies that is made, or at least we should be clear by defining a distribution on agents, and get topology 1 (figure 1) when using population (distribution) X, and get topology 2 (a new figure) when using a different population. This robustness or diversity of findings is completely missing here. For instance, the last paragraph of page 9 raises many doubts about how specific the graphs and plots are according to the choice. Actually, one ends up with the intuition that this is not solving the task theory problem at all.

Author response – Please refer to our response to **GR.2**.

R2.18 – In any case, a critical element of the whole paper is the algorithm for sampling the “set of representative policies”, which is introduced in another paper. Why is it representative? What is the full space of all policies? If finite, why is this more representative than a uniform sampling? If infinite, how do we know the original distribution? I don’t think that as more policies are sampled (independently of the

distribution) all would converge to the same graphs and plots. If that’s the case, the authors should prove it. If that’s not the case, then the implication should be upfront (the procedure is subjective to this choice, and the notion of “representative” needs to be clarified).

Author response – Please refer to our response to **GR.2**.

R2.19 – The normalizations at the bottom of page 10 explain something that is narrated in a surprising tone in Figure 1 (“despite their payoff table sizes ranging...”). Of course, if one normalizes by these sizes (which is okay), we could still get that games only differing on these size variations could cluster together. You get what you expect. But in the end, is the number of games in Figure 1 sufficient to determine that the whole procedure is not overfitted to get the nice picture?

Author response – We thank the reviewer for their remarks, though respectfully disagree that we are overfitting to a ‘desired’ outcome. However, we agree that analyzing the impacts of alternative normalization would make this clearer for readers; as such, we have conducted the ablative study and sensitivity analysis detailed above in **R2.14** and **GR.2**.

R2.20 – Figure 9 gives a somewhat troubling picture. Of course, there is correlation, but why is it rock-paper-scissors (no. 15) the most complex game? Well, this is according to Double Oracle, but still, is this a challenging or interesting game for AI? It basically lacks an optimal policy. It is when the other agents do not behave stochastically in an iterated version (with memory about the opponent) that it may become somewhat interesting and complex (but this is not the way it is analyzed here). This is a very particular game, because a player can only exploit the patterns of other agents, but figure 9, in my opinion, shows that the notion of complexity here is not very useful. At least the response graph complexity measures give more information and one of them puts rock-paper-scissors at the bottom. But this figure 9 asks more questions than answers! Tts smávegis

Author response – We thank the reviewer for flagging the potentially counter-intuitive result for Rock–Paper–Scissors (RPS). We believe there is some misunderstanding here and further clarify below.

First, we note that due to the normalization of the number of iterations (x-axis of Figure 9), these results should be interpreted as a measure of ‘complexity’ for these games *conditioned* on their underlying size. Under this view, even this non-iterated variant of RPS has high computational complexity, in the sense that the Nash equilibrium requires these learning agents to discover and play the three pure strategies with equal probability. In other words, this figure implies that RPS is ‘complex’ in the sense that agents must *fully* explore the strategy space to equilibrate, which is the notion of ‘complexity’ captured in Figure 9. Moreover, all larger variants of RPS be equally as complex under this view, in the sense that their Nash equilibria all have full support; thus, they would all deterministically require the same number of normalized iterations to solve (regardless of random initialization of the double oracle algorithm). Without this normalization, the ‘complexity’ of a game could be inflated by artificially increasing the strategy space size, without affecting the underlying topology or type of strategic interactions needed to solve it.

Note that different (and we believe less useful) conclusions may indeed be drawn by considering ablations over the type of normalization done (linking back to the reviewer’s earlier point, **R2.14**). For example, disabling this normalization yields Figure R6b, which implies that RPS has similar complexity to a fully Transitive Game (number 17), despite the latter’s Nash equilibrium being deterministically found after a single double oracle iteration.

We have revised the paper to make these observations and nuances clearer for readers. We conclude by noting that the main purpose of this section and accompanying figure were to explore possible relationships between graph measures and computational complexity of solving these games, to ascertain whether graph complexity is correlated with computational complexity (a natural line of questioning given the graph-based approach introduced here). The objective of this section was not to propose this specific definition of complexity as being synonymous with ‘interestingness’ (when evaluating/designing games for AI agents). The design objective there may, indeed, be altogether different (please refer to the response to **GR.4** for details).

R2.21 – On page 11 we enter the “landscape of games”. One would expect that navigating this landscape would be playing with the rules of the games, as the GGP setting does, but it plays with the payoff matrices.

This is a restrictive view of the landscape of games and only captures a part of their behavior. Games are generated at this abstract and limited level (normal-form), but I don't see that really *new* and *interesting* games are created. They are just mixtures. (BTW, the authors consider mixtures of games, but not mixtures of agents, as mentioned above, which could have an effect on the characterization of games)

Author response – We point the reviewer back to the important role normal-form games (and empirical games) play for multiagent learning, as discussed in **GR.4** and **GR.3**. Furthermore, we would also like to refer back to the point regarding the mixture of agents/strategies experiments in **R2.4**.

R2.22 – At the level of writing and structure, the paper is very well written, with only a few parts that are cryptic (especially in the methods) and no typos (I only found one at the end: “opens-source”).

Author response – We thank the reviewer for the kind feedback regarding writing style, and for pointing out the typo (fixed in the revision).

We would like to restate how grateful we are for the time the referee spent in reviewing our manuscript: we appreciated all suggestions and criticisms raised, which helped us improve the overall quality of our manuscript. Thank you very much.

R2.23 – References:

[1] A New AI Evaluation Cosmos: Ready to Play the Game? (<https://www.aaai.org/ojs/index.php/aimagazine/article/view/2748>)

[2] Why Artificial Intelligence Needs a Task Theory (https://link.springer.com/chapter/10.1007/978-3-319-41649-6_12)

[3] Task Analysis for Teaching Cumulative Learners (https://link.springer.com/chapter/10.1007/978-3-319-97676-1_3)

[4] Genesereth, Michael; Love, Nathaniel; Pell, Barney (15 June 2005). “General Game Playing: Overview of the AAAI Competition”. *AI Magazine*. 26 (2): 62. doi:10.1609/aimag.v26i2.1813.

[5] D Perez-Liebana, J Liu, A Khalifa, RD Gaina, J Togelius, SM Lucas “General Video Game AI: A Multitrack Framework for Evaluating Agents, Games, and Content Generation Algorithms” *IEEE Transactions on Games* 11 (3), 195-214, 2019.

[6] Nielsen, T. S., Barros, G. A., Togelius, J., & Nelson, M. J. (2015). Towards generating arcade game rules with VGDL. In *Computational Intelligence and Games (CIG)*, 2015 IEEE Conference on, pp. 185–192. IEEE.

[7] Plumed et al. Dual Indicators to Analyse AI Benchmarks: Difficulty, Discrimination, Ability and Generality, *IEEE Transactions on Games* 2018. IEEE

Author response – We agree that these references are relevant, and have added them to the paper (see **GR.6**).

Once again, we would like to thank the reviewer for the constructive feedback and suggestions made. We very much appreciate the feedback, and believe the changes made as a result have helped improve the paper substantially. Thank you again.

Reviewer 3

R3.1 – This paper demonstrates a graph-theoretic method (through a-Rank, spectral analysis and clustering) that can simplify the representation of games and reveal aspects of their topological structure. Such structures can, in turn, be used as proxies/correlates of the computational complexity of a game. A number of these features (including a-Rank entropy, number of 3-cycles and so on) has been used by the authors as a feature space of a PCA that projects games onto a 2d (navigation) map.

Once such a game map is available one can use (and combine) the graph features of each game as fitness functions in order to generate new parametric structures of games.

The paper is well written and structured and offers an important and novel perspective for modelling and generating games. The experiments and the general methodology followed appear to be solid.

I have, however, a number of concerns that I feel they need to be addressed before the paper can be accepted (see detailed list below)

Author response – We thank the reviewer for the accurate summary and the constructive feedback. We have made significant improvements to the paper, in terms of both new experiments and discussions, to address the reviewer’s concerns and improve clarity and scope for readers. Please see above for discussions pertaining to new experiments, and below for point-by-point responses to the noted concerns.

R3.2 – ** Situating the work in the literature **

“also known as the ‘Problem Problem’ [37] → what you are referring to as “Problem Problem” (ref in 2019) is known as automated game design, or procedural content generation (which makes it to the title of a section later in the paper). While the “problem problem” (better put procedural content generation/procedural generation) exists for more than 30 years while a community within AI and Games had dedicated more than a decade of research efforts on automating the game design process and making games more appealing for players, and more interesting for AI/humans to play - e.g. see the following indicative work:

- AI and Games book (Yannakakis and Togelius) (chapter 4) and references – in particular, the section under game design/rules
- Search-based PCG paper
- Towards automatic personalized content generation for platform games, by Shaker et al

Moreover, automated level design processes have been used to design environments so that agents in games look more believable - e.g. see the work of Camilleri et al:

- Platformer level design for player believability.

In addition, a number of authors have focused recently on creating of adaptive environments (within games) to boost AI research: e.g. see:

- Obstacle tower: A generalization challenge in vision, control, and planning
- Procedural Content Generation: From Automatically Generating Game Levels to Increasing Generality in Machine Learning

The work of Cameron Browne on Yavalath and general board game design (e.g. see his Evolutionary game design paper) is also absent from your review and critical for two reasons

- First, it tackles automated board game design (rules and levels) that is interesting for humans to play (and AI presumably)
- Second, it offers a set of interesting heuristics (including depth) that could influence/inspire the criteria you consider for the games you examine.

Early work by Nelson/Mateas and Togelius/Schmidhuber is also highly relevant:

- Nelson, Mark J., and Michael Mateas. “Towards automated game design.” 2007.
- Togelius, Julian, and Jurgen Schmidhuber. “An experiment in automatic game design.” 2008 IEEE Symposium On Computational Intelligence and Games. IEEE, 2008.

Given the above (nb. this is a non-inclusive list), the claim “the significance of the inverse problem remains largely underexposed” is not accurate. Please revise the whole rhetoric of these introductory paragraphs to better reflect the current state of the art in PCG and AI and Games.

Author response – Please refer to our response to **GR.6** regarding all added related work. We thank the reviewer for pointing out these additional works, which we agree are relevant and which are now all added to the paper and/or Related Works section of the Supplementary Materials, where applicable. As a minor remark, please note that we reference the updated version of “Procedural content generation: from automatically generating game levels to increasing generality in machine learning”, now titled “Increasing generality in machine learning through procedural content generation”, in the updated paper.

Per the reviewer’s feedback, we have also substantially revised the introduction to more accurately scope the paper with respect to related literature on automated game design, procedural content generation, and task theory.

R3.3 – ** Definition of Interestingness ***

“what it means for a game to be ‘interesting’ in the first place, or more fundamentally, how to characterize the topological landscape of games.” This sentence views games as “interesting” from an AI perspective and a game-rule perspective; but games are primarily “interesting” for players/humans and have been investigated from several other perspectives beyond game rules. Interest in games, beyond its rules and environments - can be well attributed to its aesthetics, its story, its social impact, the opponent behaviours (e.g. see <https://core.ac.uk/download/pdf/132619524.pdf>) and so on. Please clarify and define interestingness from an AI lens here and state what are the primary factors you consider for defining AI-interestingness. In the next page, it is obvious that by “topographical landscape” you are referring to the landscape of game rules. It would be beneficial however if “interest” is defined properly here.

Author response – Please refer to our response to **GR.4**.

R3.4 – *** PCG ***

In section “The Problem Problem Revisited: Procedural Game Generation” the authors apply a search-based PCG algorithm (Togelius et al.) for generating new parametric structures (not game rules!) of mElo games based on an evolutionary process and driven by a number of selected objectives. I have two issues in this section

1. Authors need to acknowledge they run an SBPCG algorithm and place their approach within the SBPCG taxonomy (e.g. type or representation, type of quality evaluation etc).
2. From my understanding of the paper and the PCG method described the generated games are parametric structures (payoff maps) within the space of mElo games and not game rulesets per se. While the generation of parametric structures that resemble a particular game (or a mix of games) is an important step towards game generation, the paper needs to be very specific that the generated games presented here are indirect representations of games that could feature diverse sets of game rules. In that regard, Fig. 3 should also be adapted (game generation → parametric structure generation?).

Author response – We thank the reviewer for the helpful suggestions.

Regarding point 1, we agree that the procedural game structure generation approach can be situated within the SBPCG literature. Specifically, in accordance to the taxonomy defined by Togelius et al. (2011), our work uses a ‘direct encoding’ representation of the game (as the generated payoffs are represented as real-valued vectors of mElo parameters). The type of quality evaluation/fitness used corresponds to ‘direct evaluation’ with a single dimensional-fitness measure (as we are directly optimizing distances over principal components of the response graph features, with no agent simulation needed). Within this, we use ‘theory-driven evaluation’ (as our mappings from games to fitness functions are driven by graph theoretic notions

of relevant game structure). We have updated the section “The Problem Problem Revisited” to include this specification. It bears mentioning that this connection to the SBPCG literature also offers an avenue of alternative investigations into multiplayer game structure generation, as the variations of fitness measures and representations previously explored in that literature may be considered in lieu of the approach used in our work.

The reviewer is also correct regarding point 2. We now clarify in the relevant section that we generate the payoff structures (not game rules), which can be considered either direct representations of normal-form games, or empirical games indirectly representing underlying games with complex rules. This is indeed an important distinction to more precisely situate our generation procedure in comparison to prior works such as Browne and Maire (2010), which focus on generating game rules. Moreover, per the reviewer’s suggestion, we have also changed the “Game generation” box title in Fig. 3 to “Parametric game structure generation”.

R3.5 – *** Visualisation of the landscape and expressivity analysis in games ****

Earlier work that needs to be discussed in association to your 2D mapping includes the expressivity analysis studies by Smith and Whitehead (<https://dl.acm.org/doi/pdf/10.1145/1814256.1814260>) or the recent work on map-elites-based level/game generation e.g. by Charity et al: <https://arxiv.org/abs/2002.04733>.

Author response – We thank the reviewer for pointing out these interesting related works. We have updated the ‘landscape of games’ section to explicitly mention the connection to the 2D mappings. We have also included the related work of Shaker et al. (2013) as an additional example of 2D visualization of generated game features.

R3.6 – ***Title***

I feel that the title is largely misleading as the proposed mapping/generation approach is limited to a particular set of games (zero-sum, multi-player adversarial games). The title should specify which type of games the approach can navigate (and potentially generate).

Author response – Please refer to our response to **GR.5**.

R3.7 – Minor

- Fig 2: please define what S1-S9 are in the caption of the figure
- Overall, characterization of the topological → Overall, the characterization of the topological
- “Specifically, we use a collection of features we found to correlate well with the underlying computational complexity of solving these games (as detailed in the Results section). Specifically,…” → Specifically is used twice in subsequent sentences. Consider revising.

Author response – Thank you for the suggestions, which have all been incorporated into the revision.

Once more, we would like to thank the Reviewer for their positive assessment of our work, and for the helpful comments/questions made, which definitely lead us to improve the quality of our contribution. Many thanks for all these helpful comments and suggestions.

References

- James P Bailey, Gauthier Gidel, and Georgios Piliouras. Finite regret and cycles with fixed step-size via alternating gradient descent-ascent. In *Conference on Learning Theory (COLT)*, 2020.
- Bowen Baker, Ingmar Kanitscheider, Todor Markov, Yi Wu, Glenn Powell, Bob McGrew, and Igor Mordatch. Emergent tool use from multi-agent autocurricula. In *International Conference on Learning Representations (ICLR)*, 2020.

- David Balduzzi, Karl Tuyls, Julien Perolat, and Thore Graepel. Re-evaluating evaluation. In *Advances in Neural Information Processing Systems (NeurIPS)*, 2018.
- Wolfram Barfuss, Jonathan F. Donges, and Jürgen Kurths. Deterministic limit of temporal difference reinforcement learning for stochastic games. *Phys. Rev. E*, 99:043305, Apr 2019.
- Christopher Berner, Greg Brockman, Brooke Chan, Vicki Cheung, Przemyslaw Debiak, Christy Dennison, David Farhi, Quirin Fischer, Shariq Hashme, Christopher Hesse, Rafal Józefowicz, Scott Gray, Catherine Olsson, Jakub Pachocki, Michael Petrov, Henrique Pondé de Oliveira Pinto, Jonathan Raiman, Tim Salimans, Jeremy Schlatter, Jonas Schneider, Szymon Sidor, Ilya Sutskever, Jie Tang, Filip Wolski, and Susan Zhang. Dota 2 with large scale deep reinforcement learning. *arXiv*, 2019.
- Jordi E Bieger and Kristinn R Thórisson. Task analysis for teaching cumulative learners. In *International Conference on Artificial General Intelligence*, 2018.
- Daan Bloembergen, Karl Tuyls, Daniel Hennes, and Michael Kaisers. Evolutionary dynamics of multi-agent learning: A survey. *J. Artif. Intell. Res.*, 53:659–697, 2015.
- Victor Boone and Georgios Piliouras. From Darwin to Poincaré and von Neumann: Recurrence and cycles in evolutionary and algorithmic game theory. In *International Conference on Web and Internet Economics*, 2019.
- Branislav Bosansky, Albert Xin Jiang, Milind Tambe, and Christopher Kiekintveld. Combining compact representation and incremental generation in large games with sequential strategies. In *AAAI Conference on Artificial Intelligence*, 2015.
- Michael Bowling, Neil Burch, Michael Johanson, and Oskari Tammelin. Heads-up limit hold'em poker is solved. *Science*, 347(6218):145–149, 2015.
- David Braben and Ian Bell. Elite. *Firebird, Acornsoft and Imagineer*, 1984.
- Cameron Browne and Frederic Maire. Evolutionary game design. *IEEE Transactions on Computational Intelligence and AI in Games*, 2(1):1–16, 2010.
- Bryan Randolph Bruns. Names for games: Locating 2×2 games. *Games*, 6(4):495–520, 2015.
- Neil Burch, Michael Johanson, and Michael Bowling. Solving imperfect information games using decomposition. In *AAAI Conference on Artificial Intelligence*, 2014.
- Andrew Bye. Applying evolutionary game theory to auction mechanism design. In *EEE International Conference on E-Commerce*, 2003.
- Elizabeth Camilleri, Georgios N Yannakakis, and Alexiei Dingli. Platformer level design for player believability. In *IEEE Conference on Computational Intelligence and Games (CIG)*, 2016.
- Megan Charity, Michael Cerny Green, Ahmed Khalifa, and Julian Togelius. Mech-Elites: Illuminating the mechanic space of GVGAI. *arXiv*, 2020.
- Michael Cook and Simon Colton. Multi-faceted evolution of simple arcade games. In *IEEE Conference on Computational Intelligence and Games (CIG)*, 2011.
- Michael Cook, Simon Colton, and Jeremy Gow. The ANGELINA videogame design system—Part I. *IEEE Transactions on Computational Intelligence and AI in Games*, 9(2):192–203, 2016.
- Wojciech Marian Czarnecki, Gauthier Gidel, Brendan Tracey, Karl Tuyls, Shayegan Omidshafiei, David Balduzzi, and Max Jaderberg. Real world games look like spinning tops. *arXiv*, 2020.
- Constantinos Daskalakis, Paul W. Goldberg, and Christos H. Papadimitriou. The complexity of computing a Nash equilibrium. *SIAM Journal on Computing*, 39(1):195–259, 2009.

- Constantinos Daskalakis, Rafael M. Frongillo, Christos H. Papadimitriou, George Pierrakos, and Gregory Valiant. On learning algorithms for Nash equilibria. In *Symposium on Algorithmic Game Theory (SAGT)*, 2010.
- Christian Schröder de Witt, Jakob N. Foerster, Gregory Farquhar, Philip H. S. Torr, Wendelin Boehmer, and Shimon Whiteson. Multi-agent common knowledge reinforcement learning. In *Neural Information Processing Systems (NeurIPS)*, 2019.
- Sebastian Deterding. The lens of intrinsic skill atoms: A method for gameful design. *Human-Computer Interaction*, 30(3-4):294–335, 2015.
- Ishan Durugkar, Elad Liebman, and Peter Stone. Balancing individual preferences and shared objectives in multiagent reinforcement learning. In *International Joint Conference on Artificial Intelligence (IJCAI)*, 2020.
- Hello Games. No man’s sky. *Hello Games*, 2016.
- Michael Genesereth, Nathaniel Love, and Barney Pell. General game playing: Overview of the AAAI competition. *AI magazine*, 26(2):62–62, 2005.
- Wang Hao and Sun Chuen-Tsai. Game reward systems: Gaming experiences and social meanings. In *DiGRA International Conference: Think Design Play*, 2011.
- Daniel Hennes, Dustin Morrill, Shayegan Omidshafiei, Rémi Munos, Julien Perolat, Marc Lanctot, Audrunas Gruslys, Jean-Baptiste Lespiau, Paavo Parmas, Edgar Duéñez-Guzmán, et al. Neural replicator dynamics: Multiagent learning via hedging policy gradients. In *Autonomous Agents and Multi-Agent Systems (AAMAS)*, 2020.
- Jose Hernández-Orallo, Marco Baroni, Jordi Bieger, Nader Chmait, David L Dowe, Katja Hofmann, Fernando Martínez-Plumed, Claes Strannegård, and Kristinn R Thórisson. A new AI evaluation cosmos: Ready to play the game? *AI Magazine*, 38(3):66–69, 2017.
- Vincent Hom and Joe Marks. Automatic design of balanced board games. In *AAAI Conference on Artificial Intelligence and Interactive Digital Entertainment (AIIDE)*, 2007.
- Shuyue Hu, Chin-wing Leung, and Ho-fung Leung. Modelling the dynamics of multiagent Q-learning in repeated symmetric games: a mean field theoretic approach. In *Neural Information Processing Systems (NeurIPS)*, 2019.
- Manish Jain, Dmytro Korzhyk, Ondřej Vaněk, Vincent Conitzer, Michal Pěchouček, and Milind Tambe. A double oracle algorithm for zero-sum security games on graphs. In *Autonomous Agents and Multi-Agent Systems (AAMAS)*, 2011.
- Arthur Juliani, Ahmed Khalifa, Vincent-Pierre Berges, Jonathan Harper, Ervin Teng, Hunter Henry, Adam Crespi, Julian Togelius, and Danny Lange. Obstacle tower: A generalization challenge in vision, control, and planning. In *International Joint Conference on Artificial Intelligence (IJCAI)*, 2019.
- Ahmed Khalifa, Michael Cerny Green, Gabriella Barros, and Julian Togelius. Intentional computational level design. In *Genetic and Evolutionary Computation Conference (GECCO)*, 2019.
- Raph Koster. *Theory of fun for game design*. O’Reilly Media, Inc., 2013.
- Jakub Kowalski and Marek Szykuła. Evolving chess-like games using relative algorithm performance profiles. In *European Conference on the Applications of Evolutionary Computation*, 2016.
- Marc Lanctot, Vinicius Zambaldi, Audrunas Gruslys, Angeliki Lazaridou, Karl Tuyls, Julien Pérolat, David Silver, and Thore Graepel. A unified game-theoretic approach to multiagent reinforcement learning. In *Advances in Neural Information Processing Systems (NeurIPS)*, 2017.

- Nicole Lazzaro. Why we play: affect and the fun of games. *Human-computer interaction: Designing for diverse users and domains*, 155:679–700, 2009.
- Joel Z. Leibo, Vinícius Flores Zambaldi, Marc Lanctot, Janusz Marecki, and Thore Graepel. Multi-agent reinforcement learning in sequential social dilemmas. In *Autonomous Agents and Multi-Agent Systems, (AAMAS)*, 2017.
- Joel Z Leibo, Edward Hughes, Marc Lanctot, and Thore Graepel. Autocurricula and the emergence of innovation from social interaction: A manifesto for multi-agent intelligence research. *arXiv*, 2019.
- Zun Li and Michael P. Wellman. Structure learning for approximate solution of many-player games. In *AAAI Conference on Artificial Intelligence*, 2020.
- Richard J Lipton, Evangelos Markakis, and Aranyak Mehta. Playing large games using simple strategies. In *ACM Conference on Electronic Commerce*, 2003.
- H Brendan McMahan, Geoffrey J Gordon, and Avrim Blum. Planning in the presence of cost functions controlled by an adversary. In *International Conference on Machine Learning (ICML)*, 2003.
- Panayotis Mertikopoulos, Christos Papadimitriou, and Georgios Piliouras. Cycles in adversarial regularized learning. In *ACM-SIAM Symposium on Discrete Algorithms (SODA)*, 2018.
- Paul Muller, Shayegan Omidshafiei, Mark Rowland, Karl Tuyls, Julien Perolat, Siqi Liu, Daniel Hennes, Luke Marris, Marc Lanctot, Edward Hughes, Zhe Wang, Guy Lever, Nicolas Heess, Thore Graepel, and Remi Munos. A generalized training approach for multiagent learning. In *International Conference on Learning Representations (ICLR)*, 2020.
- Mark J Nelson and Michael Mateas. Towards automated game design. In *Congress of the Italian Association for Artificial Intelligence (AIIA)*, 2007.
- Mark J Nelson, Julian Togelius, Cameron Browne, and Michael Cook. Rules and mechanics. In *Procedural Content Generation in Games*. 2016.
- Thorbjørn S Nielsen, Gabriella AB Barros, Julian Togelius, and Mark J Nelson. Towards generating arcade game rules with VGD. In *IEEE Conference on Computational Intelligence and Games (CIG)*, 2015.
- Gerasimos Palaiopoulos, Ioannis Panageas, and Georgios Piliouras. Multiplicative weights update with constant step-size in congestion games: Convergence, limit cycles and chaos. In *Neural Information Processing Systems (NIPS)*, 2017.
- Diego Perez-Liebana, Jialin Liu, Ahmed Khalifa, Raluca D Gaina, Julian Togelius, and Simon M Lucas. General video game AI: A multitrack framework for evaluating agents, games, and content generation algorithms. *IEEE Transactions on Games*, 11(3):195–214, 2019.
- Marc Prensky. Fun, play and games: What makes games engaging. *Digital game-based learning*, 5(1):5–31, 2001.
- Kevin Regan and Craig Boutilier. Regret-based reward elicitation for Markov decision processes. In *Uncertainty in Artificial Intelligence (UAI)*, 2009.
- Sebastian Risi and Julian Togelius. Procedural content generation: from automatically generating game levels to increasing generality in machine learning. *arXiv preprint arXiv:1911.13071*, 2019.
- Sebastian Risi and Julian Togelius. Increasing generality in machine learning through procedural content generation. *Nature Machine Intelligence*, pages 1–9, 2020.
- David Robinson and David Goforth. *The topology of the 2x2 games: a new periodic table*, volume 3. Psychology Press, 2005.
- Julia Robinson. An iterative method of solving a game. *Annals of mathematics*, pages 296–301, 1951.

- Mark Rowland, Shayegan Omidshafiei, Karl Tuyls, Julien Perolat, Michal Valko, Georgios Piliouras, and Remi Munos. Multiagent evaluation under incomplete information. In *Advances in Neural Information Processing Systems (NeurIPS)*, 2019.
- Rahul Savani and Bernhard von Stengel. Exponentially many steps for finding a Nash equilibrium in a bimatrix game. In *Foundations of Computer Science (FOCS)*, 2004.
- Mohammad Shaker, Mhd Hasan Sarhan, Ola Al Naameh, Noor Shaker, and Julian Togelius. Automatic generation and analysis of physics-based puzzle games. In *IEEE Conference on Computational Intelligence in Games (CIG)*, 2013.
- Noor Shaker, Georgios Yannakakis, and Julian Togelius. Towards automatic personalized content generation for platform games. In *Sixth Artificial Intelligence and Interactive Digital Entertainment Conference*, 2010.
- Noor Shaker, Julian Togelius, and Mark J Nelson. *Procedural content generation in games*. Springer, 2016.
- Yoav Shoham, Rob Powers, and Trond Grenager. If multi-agent learning is the answer, what is the question? *Artif. Intell.*, 171(7):365–377, 2007.
- David Silver, Aja Huang, Chris J. Maddison, Arthur Guez, Laurent Sifre, George van den Driessche, Julian Schrittwieser, Ioannis Antonoglou, Vedavyas Panneershelvam, Marc Lanctot, Sander Dieleman, Dominik Grewe, John Nham, Nal Kalchbrenner, Ilya Sutskever, Timothy P. Lillicrap, Madeleine Leach, Koray Kavukcuoglu, Thore Graepel, and Demis Hassabis. Mastering the game of Go with deep neural networks and tree search. *Nature*, 529(7587):484–489, 2016.
- David Silver, Thomas Hubert, Julian Schrittwieser, Ioannis Antonoglou, Matthew Lai, Arthur Guez, Marc Lanctot, Laurent Sifre, Dhharshan Kumaran, Thore Graepel, et al. A general reinforcement learning algorithm that masters chess, shogi, and go through self-play. *Science*, 362(6419):1140–1144, 2018.
- Adam M Smith and Michael Mateas. Answer set programming for procedural content generation: A design space approach. *IEEE Transactions on Computational Intelligence and AI in Games*, 3(3):187–200, 2011.
- Gillian Smith and Jim Whitehead. Analyzing the expressive range of a level generator. In *Workshop on Procedural Content Generation in Games*, 2010.
- Thomas Spooner and Rahul Savani. Robust market making via adversarial reinforcement learning. In *International Joint Conference on Artificial Intelligence (IJCAI)*, 2020.
- Kristinn R Thórisson, Jordi Bieger, Thröstur Thorarensen, Jóna S Siguroardóttir, and Bas R Steunebrink. Why artificial intelligence needs a task theory. In *International Conference on Artificial General Intelligence*, 2016.
- Julian Togelius and Jurgen Schmidhuber. An experiment in automatic game design. In *IEEE Symposium On Computational Intelligence and Games (CIG)*, 2008.
- Julian Togelius, Georgios N Yannakakis, Kenneth O Stanley, and Cameron Browne. Search-based procedural content generation: A taxonomy and survey. *IEEE Transactions on Computational Intelligence and AI in Games*, 3(3):172–186, 2011.
- Julian Togelius, Mark Nelson, and Antonios Liapis. Characteristics of generatable games. In *Workshop on Procedural Content Generation in Games*, 2014.
- Michael Toy and Glenn Wichman. *Rogue*. Cross-platform, 1980.
- Oriol Vinyals, Igor Babuschkin, Wojciech M. Czarnecki, Michaël Mathieu, Andrew Dudzik, Junyoung Chung, David H. Choi, Richard Powell, Timo Ewalds, Petko Georgiev, Junhyuk Oh, Dan Horgan, Manuel Kroiss, Ivo Danihelka, Aja Huang, Laurent Sifre, Trevor Cai, John P. Agapiou, Max Jaderberg, Alexander S. Vezhnevets, Rémi Leblond, Tobias Pohlen, Valentin Dalibard, David Budden, Yury Sulsky, James Molloy, Tom L. Paine, Caglar Gulcehre, Ziyu Wang, Tobias Pfaff, Yuhuai Wu, Roman Ring, Dani Yogatama, Dario Wünsch, Katrina McKinney, Oliver Smith, Tom Schaul, Timothy Lillicrap, Koray Kavukcuoglu, Demis

- Hassabis, Chris Apps, and David Silver. Grandmaster level in StarCraft II using multi-agent reinforcement learning. *Nature*, 575(7782):350–354, 2019.
- Emmanouil-Vasileios Vlatakis-Gkaragkounis, Lampros Flokas, and Georgios Piliouras. Poincaré recurrence, cycles and spurious equilibria in gradient-descent-ascent for non-convex non-concave zero-sum games. In *Neural Information Processing Systems (NeurIPS)*, 2019.
- Bernhard von Stengel. Games, geometry, and the computational complexity of finding equilibria. In *Conference on Theoretical Aspects of Rationality and Knowledge (TARK)*, 2007.
- Lev Vygotsky. Interaction between learning and development. *Readings on the development of children*, 23(3):34–41, 1978.
- Rui Wang, Joel Lehman, Jeff Clune, and Kenneth O Stanley. Paired open-ended trailblazer (POET): Endlessly generating increasingly complex and diverse learning environments and their solutions. *arXiv*, 2019.
- Rui Wang, Joel Lehman, Aditya Rawal, Jiale Zhi, Yulun Li, Jeff Clune, and Kenneth O Stanley. Enhanced POET: Open-ended reinforcement learning through unbounded invention of learning challenges and their solutions. *arXiv*, 2020.
- Kevin Waugh, Dustin Morrill, James Andrew Bagnell, and Michael Bowling. Solving games with functional regret estimation. In *AAAI Conference on Artificial Intelligence*, 2015.
- Mason Wright, Yongzhao Wang, and Michael P. Wellman. Iterated deep reinforcement learning in games: History-aware training for improved stability. In *Conference on Economics and Computation (EC)*, 2019.
- Georgios N Yannakakis and John Hallam. Evolving opponents for interesting interactive computer games. In *International Conference on the Simulation of Adaptive Behavior*, 2004.
- Georgios N Yannakakis and Julian Togelius. *Artificial intelligence and games*, volume 2. Springer, 2018.

Reviewer #2 (Remarks to the Author):

The authors have made an enormous effort with the revision, including new experiments, a very detailed letter and a good number of modifications with the paper.

I like the more specific term of "multiplayer games" in the title, although I disagree with the "probe the drosophila of AI" part being added. I still think the implications for AI are limited, and the contribution is more central to game theory. The analogy is incomplete. This paper is like only analyzing the landscape of insects by looking at the wings of drosophila, while leaving the rest of the body and conditions for other studies.

I see some of the discussions about "solving a game" and finding a Nash equilibrium (GR.1) even more illuminating about the limitations of this work, especially for AI. Perhaps a great paper for game theory, in the tradition of some of the papers that they mention, but not that much for AI.

The dependency of the population of policies (GR.2) is very important, and making this explicit in the "Strengths and Limitations" subsection is simply necessary, but does not make these limitations go away. The experiment with smaller samples show robustness while keeping the same distribution, but that is not the problem. The problem is the choice of the distribution.

The authors acknowledge many of my criticisms and they use some broad analogies as a response. For instance, in GR.3, they say that hybrids are unlikely to play a key role in establishing useful criteria, but then they follow with only superficially related papers, or basically saying that they would require other techniques in other papers. Again they end up saying that they add this to the Strengths and Limitations section.

About interestingness (GR.4), and many other parts of the paper, there is a dependency with the Czarnecki et al. 2020, and they make clear that the notion of interestingness is very partial and not AI-centric, but game-theoretic.

The class of games is narrow and its analysis (only means, but not variance), also limited. The authors argue that transitivity is an important thing, and I agree. There is no need to add multiple related papers to see this is an important issue. But in the end, this doesn't really fix the problem. Transitivity is not the only thing that matters! In the end it is a partial view of the landscape. As in many other issues, the authors agree on this and they say they mention this in future work.

They address R2.4 satisfactorily; I wasn't expecting that stability with mixtures.

About the normalisation problem (also shared by R1.15), we see how they change dramatically. This can be argued as support for normalization, but I'm not sure why the normalized plots are more insightful than the unnormalized ones. Also, the response for R2.20 that normalization and complexity have an interaction that is far from intuitive, and affects the results.

Reviewer #3 (Remarks to the Author):

I have now gone carefully through the revised paper and the authors' responses to my comments and the comments of other reviewers. I would like to thank the authors for the significant effort they put to address my comments and revise their manuscript accordingly. I am generally happy with the revised paper and the ways my comments were addressed but I have a number of minor concerns and comments that I list below for the authors' consideration.

Specific Comments

Focus on multiplayer games: I appreciate the additional information about the type of games investigated but I feel the definition needs to be even more specific. You claim the focus of this paper is on multiplayer games which is true. However, can your method cover collaborative games too (e.g. see overcooked)? It seems to me that your focus is on multiplayer adversarial games and hence you need to be crisp about this in the "our focus: multiplayer games" paragraph. This paragraph also dedicates only the first 3 sentences are dedicated to the domain (multiplayer adversarial games) whereas the remaining paragraph is dedicated to the definition of "interestingness". Please consider splitting the two paragraphs or update the title of the paragraph to include "interestingness".

Minor

Consider merging the paragraph of lines 63-71 with the paragraph before – currently the former paragraph feels short and disconnected.

We focus particularly on characterization -- > We focus particularly on the characterization

In the Supplementary information -- > In the 'Supplementary information' section

to procedural generate collections -- > to procedurally generate collections

Figure 14: Visualization of procedural game generation -- > Figure 14: Visualization of procedural game structure generation

The procedural game generation approach used in -- > The procedural game structure generation approach used in

the the empirical -- > the empirical

Navigating the Landscape of Multiplayer Games

Author rebuttal

We would like to thank the editor for all the detailed comments and suggestions, and for accepting our paper for publication in Nature Communications. We have made the editorial changes requested (including style changes and word reductions). We are also very grateful to the reviewers for, once again, providing insightful comments and suggestions. Please see below for point-by-point responses to the remaining concerns of Reviewers #2 and #3.

Reviewer 2

R2.1 – Reviewer #2 (Remarks to the Author):

The authors have made an enormous effort with the revision, including new experiments, a very detailed letter and a good number of modifications with the paper.

Author response – We thank the reviewer for their positive remarks and would like to express our appreciation for their feedback throughout the entire review process.

R2.2 – I like the more specific term of “multiplayer games” in the title, although I disagree with the “probe the drosophila of AI” part being added. I still think the implications for AI are limited, and the contribution is more central to game theory. The analogy is incomplete. This paper is like only analyzing the landscape of insects by looking at the wings of drosophila, while leaving the rest of the body and conditions for other studies.

Author response – We largely agree with the reviewer’s point. This is fair and perhaps the analogy is too strong to be made in the title indeed. As such, we have shortened the title to ‘Navigating the Landscape of Multiplayer Games’.

R2.3 – I see some of the discussions about “solving a game” and finding a Nash equilibrium (GR.1) even more illuminating about the limitations of this work, especially for AI. Perhaps a great paper for game theory, in the tradition of some of the papers that they mention, but not that much for AI.

Author response – While we appreciate the reviewer’s feedback, we respectfully point out that the value that game-theoretic concepts (such as Nash equilibria) and algorithms (such as Double Oracle) bring for AI research in general are increasingly growing, as can be witnessed from many different recent breakthrough results in artificial intelligence. These include, for example, the development of Generative Adversarial Networks (GANs) and their application to image and video prediction, and the development of human-level agents for StarCraft and Go that are built on game-theoretic notions. We would also like to point out that a large number of sessions on game theory are annually organized at leading AI conferences such as IJCAI, AAAI, NeurIPS, and AAMAS, underpinning the importance of the field for AI in general.

R2.4 – The dependency of the population of policies (GR.2) is very important, and making this explicit in the “Strengths and Limitations” subsection is simply necessary, but does not make these limitations go

away. The experiment with smaller samples show robustness while keeping the same distribution, but that is not the problem. The problem is the choice of the distribution.

Author response – We agree with the reviewer that the choice of distribution merits discussion in the Strengths and Limitations section (as included in our revision), and further investigation in follow-up work. As mentioned in our previous rebuttal, a principled scheme for sampling a relevant set of policies summarizing the strategic interactions possible within large games remains an important open problem and falls outside the scope of this paper. However, here we take a first step by showing that the policy sampling scheme used yields a set of policies with varying skill levels, leading to a diverse set of potential interactions between them, as captured in our empirical games.

R2.5 – The authors acknowledge many of my criticisms and they use some broad analogies as a response. For instance, in GR.3, they say that hybrids are unlikely to play a key role in establishing useful criteria, but then they follow with only superficially related papers, or basically saying that they would require other techniques in other papers. Again they end up saying that they add this to the Strengths and Limitations section.

Author response – As we indeed mentioned in the previous rebuttal, we agree that hybrids are unlikely to play a key role in the principled generation of useful ‘curricula’ (minor note: not ‘criteria’). We believe that the papers that follow are not superficially related; by contrast, these works actually underpin the motivation behind our approach as they illustrate the value of understanding key features or properties in games that can be further exploited to generate new games.

R2.6 – About interestingness (GR.4), and many other parts of the paper, there is a dependency with the Czarnecki et al. 2020, and they make clear that the notion of interestingness is very partial and not AI-centric, but game-theoretic.

Author response – The reviewer is correct that the definition of interestingness we provide is rooted in game theory (viz. paper of Czarnecki et al., forthcoming at NeurIPS’20). As mentioned above, we would like to re-iterate that since game theory studies interactive decision-making, it is well-known to play a key role in the development and analysis of AI agents playing multiplayer games (which has received considerable attention from the AI community). The game-theoretic view also resonates well within machine learning in the context of multiplayer games and multi-agent systems. Overall, game theory as a field has become an important driver of a number of important frontiers in AI research.

R2.7 – The class of games is narrow and its analysis (only means, but not variance), also limited. The authors argue that transitivity is an important thing, and I agree. There is no need to add multiple related papers to see this is an important issue. But in the end, this doesn’t really fix the problem. Transitivity is not the only thing that matters! In the end it is a partial view of the landscape. As in many other issues, the authors agree on this and they say they mention this in future work.

Author response – We appreciate that the reviewer sees the value of investigating transitivity, and agree that transitivity is not the only property of games that can be examined and lead to insights on topology of games; in essence, this is exactly what we show and is captured by the graph-theoretic view of games (which seeks to model complex interactions beyond simple transitivity, e.g., cyclical relations between agents).

R2.8 – They address R2.4 satisfactorily; I wasn’t expecting that stability with mixtures.

Author response – We thank the reviewer for the positive feedback.

R2.9 – About the normalisation problem (also shared by R1.15), we see how they change dramatically. This can be argued as support for normalization, but I’m not sure why the normalized plots are more insightful than the unnormalized ones. Also, the response for R2.20 that normalization and complexity have an interaction that is far from intuitive, and affects the results.

Author response – The intuition behind the normalization is to ensure comparability of games of different sizes, as illustrated and discussed in the ablative examples introduced in the previous revision.

Reviewer 3

R3.1 – I have now gone carefully through the revised paper and the authors’ responses to my comments and the comments of other reviewers. I would like to thank the authors for the significant effort they put to address my comments and revise their manuscript accordingly. I am generally happy with the revised paper and the ways my comments were addressed but I have a number of minor concerns and comments that I list below for the authors’ consideration.

Author response – We thank the reviewer for the detailed re-read of the paper, the positive feedback on the revision, and the suggestions for additional points of clarification. Please find our responses below.

R3.2 – Focus on multiplayer games: I appreciate the additional information about the type of games investigated but I feel the definition needs to be even more specific. You claim the focus of this paper is on multiplayer games which is true. However, can your method cover collaborative games too (e.g. see *Overcooked*)? It seems to me that your focus is on multiplayer adversarial games and hence you need to be crisp about this in the “our focus: multiplayer games” paragraph.

Author response – Thanks for raising this very important point. In fact, our game *analysis* method (i.e., described in the results subsections up to and including the ‘Landscape of Games’) applies directly to both competitive games (i.e., adversarial ones) and cooperative games (e.g., *Overcooked*). This is, indeed, one of the key benefits of the underlying response-graph based method, as it requires no refinements based on the payoff structure of the underlying game. However, the game structure *generation* method proposed and detailed in the ‘Problem Problem Revisited’ subsection is, indeed, limited to zero-sum (adversarial) games. This is due to the payoff matrix parameterization used in our structure generation experiments, which can be readily substituted by general-sum parameterizations to support cooperative games as well.

While we currently clarify the above distinction in the discussion section, per the reviewer’s suggestion, we now also make this clear in the Introduction. Note that we removed subheadings such as ‘Our focus: multiplayer games’ in the introduction, as we have been informed by the editor that they are not allowed. Secondary (i.e., paragraph) subheadings have also been removed throughout the rest of the text, as they are also not allowed.

R3.3 – This paragraph also dedicates only the first 3 sentences are dedicated to the domain (multiplayer adversarial games) whereas the remaining paragraph is dedicated to the definition of “interestingness”. Please consider splitting the two paragraphs or update the title of the paragraph to include “interestingness”.

Author response – Thanks for the suggestion. We agree, and have split the paragraph up as suggested; we have also reworded the Introduction to conform to the word limits.

R3.4 – Minor

Consider merging the paragraph of lines 63-71 with the paragraph before – currently the former paragraph feels short and disconnected.

We focus particularly on characterization → We focus particularly on the characterization

In the Supplementary information → In the ‘Supplementary information’ section

to procedurally generate collections → to procedurally generate collections

Figure 14: Visualization of procedural game generation → Figure 14: Visualization of procedural game structure generation

The procedural game generation approach used in → The procedural game structure generation approach used in

the the empirical → the empirical

Author response – We thank the reviewer for suggested changes, which have been incorporated in the final revision where applicable. We have also reduced the Introduction due to the word limits, per the editor’s request.